



# Emulation of long-term changes in global climate: Application
# to the late Pliocene and future
Natalie S. Lord[1,2], Michel Crucifix[3,4], Dan J. Lunt[1,2], Mike C. Thorne[5], Nabila Bounceur[3],
Harry Dowsett[6], Charlotte L. O'Brien[6,7] and Andy Ridgwell[1,2,8]
[1]School of Geographical Sciences, University of Bristol, Bristol, BS8 1SS, UK.
[2]Cabot Institute, University of Bristol, Bristol, UK.
[3]Université catholique de Louvain, Georges Lemaître Centre for Earth and Climate Research, Earth and Life
Institute, 1348 Louvain-la-Neuve, Belgium.
[4]Belgian National Fund of Scientific Research, Brussels, Belgium.
[5]Mike Thorne and Associates Limited, Quarry Cottage, Hamsterley, Bishop Auckland, Co. Durham, DL13 3NJ,
UK.
[6]Eastern Geology and Paleoclimate Science Center, U. S. Geological Survey, Reston, VA 20192, USA.
[7]Department of Geology and Geophysics, Yale University, New Haven, CT 06511, USA.
[8]Department of Earth Sciences, University of California, Riverside, CA 92521, USA.
*Correspondence to:* Natalie S. Lord (Natalie.Lord@bristol.ac.uk)





**Abstract**
Multi-millennial transient simulations of climate changes have a range of important applications, such as for
investigating key geologic events and transitions for which high resolution palaeoenvironmental proxy data are
available, or for projecting the long-term impacts of future climate evolution on the performance of geological
repositories for the disposal of radioactive wastes. However, due to the high computational requirements of current
fully coupled General Circulation Models (GCMs), long-term simulations can generally only be performed with
less complex models and/or at lower spatial resolution. In this study, we present novel long-term "continuous"
projections of climate evolution based on the output from GCMs, via the use of a statistical emulator. The emulator
is calibrated using ensembles of GCM simulations which have varying orbital configurations and atmospheric
$CO_2$ concentrations and enables a variety of investigations of long-term climate change to be conducted which
would not be possible with other modelling techniques at the same temporal and spatial scales. To illustrate the
potential applications, we apply the emulator to the late Pliocene (by modelling SAT), comparing its results with
palaeo-proxy data for a number of global sites, and to the next 200 thousand years (kyr) (by modelling SAT and
precipitation). A range of $CO_2$ scenarios are modelled for each period. During the late Pliocene, we find that
emulated SAT varies on an approximately precessional timescale, with evidence of increased obliquity response
at times. A comparison of atmospheric $CO_2$ concentration for this period, estimated using the proxy data and
emulator results and using proxy $CO_2$ records, finds that relatively similar concentrations are produced at lower
latitudes, although higher latitude sites show larger discrepancies. In our second illustrative application, spanning
the next 200 kyr into the future, we find that SAT oscillations appear to be primarily influenced by obliquity for
the first ~120 kyr, whilst eccentricity is relatively low, after which precession plays a more dominant role.
Conversely, variations in precipitation over the entire period demonstrate a strong precessional signal. Overall,
we find that the emulator provides a useful and powerful tool for rapidly simulating the long-term evolution of
climate, both past and future, due to its relatively high spatial resolution and relatively low computational cost.



## 1 Introduction

Palaeoclimate natural archives reveal how the Earth's past climate has fluctuated between warmer and cooler intervals. Glacial periods, such as the Last Glacial Maximum (e.g. Lambeck et al., 2001; Yokoyama et al., 2000), exhibit relatively lower temperatures associated with extensive ice sheets at high northern latitudes (Herbert et al., 2010; Jouzel et al., 2007; Lisiecki and Raymo, 2005), whilst interglacials are characterized by much milder temperatures in global mean. Even warmer and sometimes transient ("hyperthermal") intervals, such as occurred during the Palaeocene-Eocene Thermal Maximum (e.g. Kennett and Stott, 1991), occur characterized by even higher global mean temperatures. Assuming that on glacial-interglacial timescales and across transient warmings and climatic transitions, tectonic effects can be neglected, the timing and rate of climatic change is at least partly controlled by the three main orbital parameters – precession, obliquity and eccentricity – which have cycle durations of approximately 23, 41, and both 96 and ~400 thousand years (kyr), respectively (Berger, 1978; Hays et al., 1976; Kawamura et al., 2007; Lisiecki and Raymo, 2007; Milankovitch, 1941). Further key drivers of past climate dynamics include changes in atmospheric $CO_2$ concentration and in respect of the glacial-interglacial cycles, changes in the extent and thickness of ice sheets.

In order to investigate the dynamics, impacts and feedbacks associated with the response of the system to orbital forcing and $CO_2$, long-term ($>10^3$ years (yr)) projections of changing climate are required. Transient simulations such as these are useful for investigating key past episodes of extended duration for which detailed palaeoenvironmental proxy data are available, such as through the Quaternary and Pliocene, allowing data-model comparisons. Simulations of long-term future climate change also have a number of applications, such as in assessments of the safety of geological disposal of radioactive wastes. Due to the long half-lives of potentially harmful radionuclides in these wastes, geological disposal facilities must remain functional for up to 100 kyr in the case of low- and intermediate-level wastes (e.g. Low Level Waste Repository, UK (LLWR, 2011)), and up to 1 Ma in the case of high-level wastes and spent nuclear fuel (e.g. proposed KBS-3 facility, Sweden (SKB, 2011)). Projections of possible long-term future climate evolution are therefore required in order for the impact of potential climatic changes on the performance and safety of a repository to be assessed (SKB, 2013; Texier et al., 2003). Indeed, while the glacial-interglacial cycles are expected to continue into the future, the timing of onset of the next glacial episode is currently uncertain and will be fundamentally impacted by the increased radiative forcing from anthropogenic $CO_2$ emissions (Archer and Ganopolski, 2005; Ganopolski et al., 2016; Loutre and Berger, 2000b).

Making spatially-resolved past or future projections of changes in surface climate generally involves the use of fully coupled General Circulation Models (GCMs). However, a consequence of their high spatial and temporal resolution and structural complexity (and attendant computational resources) is that it is not usually practical to run them for simulations of more than a few millennia, and invariably, rather less than a single processional cycle. Even when run for several thousand years, only a limited number of runs can be performed. Previously, therefore, lower complexity models such as Earth system Models of Intermediate Complexity (EMICs) have been used to simulate long-term transient past (e.g. Loutre and Berger, 2000a; Stap et al., 2014) and future (e.g. Archer and Ganopolski, 2005; Eby et al., 2009; Ganopolski et al., 2016; Lenton et al., 2006; Loutre and Berger, 2000b) climate development. Where GCMs have been employed, generally only a small number of snapshot simulations of



particular climate states or time slices of interest have been modelled (Braconnot et al., 2007; Haywood et al.,
2013; Marzocchi et al., 2015; Masson-Delmotte et al., 2011; Prescott et al., 2014).

In this study, we present long-term continuous projections of climate evolution based on the output from a GCM,
via the use of a statistical emulator. Emulators have been utilised in previous studies for a range of applications,
including sensitivity analyses of climate to orbital, atmospheric $CO_2$ and ice sheet configurations (Araya-Melo et
al., 2015; Bounceur et al., 2015) and model parameterizations (Holden et al., 2010). However, to the best of our
knowledge, this is the first time that an emulator has been trained on data from a GCM and then used to simulate
long-term future transient climate change. It should be noted that, whilst other research communities may use
different terms, we refer to the groups of climate model experiments as "ensembles", and we refer directly to the
GCM when discussing calibration of the emulator, rather than using the term "simulator" as has been used in a
number of previous studies.

We calibrated an emulator using SAT data produced using the HadCM3 GCM (Gordon et al., 2000). Two
ensembles of simulations were run, with varying orbital configurations and atmospheric $CO_2$ concentrations. Each
ensemble was run twice, once with modern-day continental ice sheets and once (for a reduced number of
members) with reduced-extent ice sheets. We adopted this approach because in at least two of the intended uses
for the emulator (Pliocene, and long-term future climate for application to performance assessments for potential
radioactive waste repositories), it is thought that the Greenland and West Antarctic ice sheets (GIS, WAIS) could
be reduced relative to their current size. The ensembles thus cover a range of possible future conditions, including
the high atmospheric $CO_2$ concentrations expected in the near-term due to anthropogenic fossil fuel emissions,
and the gradual reduction of this $CO_2$ perturbation over timescales of hundreds of thousands of years by the long-
term carbon cycle (Lord et al., 2015, 2016).

We go on to illustrate a number of different ways in which the emulator can be applied to investigate long-term
climate evolution of hundreds of thousands to millions of years. Firstly, the emulator is used to simulate SAT
changes for the late Pliocene for the period 3300-2800 kyr before present (BP) for a range of $CO_2$ concentrations.
This interval occurs in the middle part of the Piacenzian Age, and was previously referred to as the "mid-
Pliocene". During this time, global temperatures were warmer than pre-industrial (Haywood and Valdes, 2004;
Lunt et al., 2010), before the transition to the intensified glacial-interglacial cycles that are associated with
modern-day climate (Lisiecki and Raymo, 2007). We then apply the emulator to future climate, simulating
temperature and precipitation data for the next 200 kyr (AP – after present) for a range of fossil fuel emissions
scenarios. Regional changes in climate at a number of European sites (grid boxes) are presented, selected either
because they have been identified as adopted or proposed locations for the geological disposal of solid radioactive
wastes, as in the cases of Forsmark, Sweden and El Cabril, Spain, or simply as reference locations where a suitable
site has not yet been identified, as in the cases of Switzerland and the UK.

The paper is structured such that the theoretical basis of the emulator is described in Sect. 2, the GCM model
description and simulations are presented in Sect. 3 and an account of how the emulator is trained and evaluated
is given in Sect. 4. Section 5 presents illustrative examples of a number of potential applications of the emulator

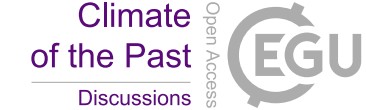



for the late Pliocene. Further examples of the application of the emulator to the next 200 kyr are described in Sect.
6, and the conclusions of this study are presented in Sect. 7.
**2 Theoretical basis of the emulator**
The emulator is a statistical representation of a more complex model, in this case a GCM. It works on the principle
that a relatively small number of experiments are carried out using the GCM, which fill the entire
multidimensional input space (in our case, four dimensions consisting of three orbital dimensions and a $CO_2$
dimension), albeit rather sparsely. The statistical model is calibrated on these experiments, with the aim of being
able to interpolate the GCM results such that it can provide a prediction of the output that the GCM would produce
if it were run using any particular input configuration. If successful (as can be tested by comparing emulator
results with additional GCM results not included in the calibration), no further experiments are required using the
GCM; the emulator can then be used to produce results for any set of conditions or sequence of sets of conditions
within the range of conditions on which it has been calibrated. It cannot, of course, be used to extrapolate to
conditions outside that range.
In this study, we use a principal component analysis (PCA) Gaussian Process (GP) emulator based on Sacks et al.
(1989), with the subsequent Bayesian treatment of Kennedy and O'Hagan (2000) and Oakley and O'Hagan (2002)
and associated with principal component analysis by Wilkinson (2010). All code for the GP package is available
online at https://github.com/mcrucifix/GP. This principal component (PC) emulator is based on climate data for
the entire global grid, as opposed to calibrating separate emulators based on data for individual grid boxes. This
approach is taken because, for past climate, the global response overall is of interest, rather than just the response
at specific locations individually. It also means that the results are consistent across all locations. For future
climate, and in particular for application to nuclear waste, recommendations and results should be consistent
across all sites, which would be especially relevant to a large country such as the US. Alternatively, for some
countries and locations, it may be more appropriate to emulate specific grid boxes. The theoretical basis for the
emulator and its calibration, is as follows.
Let $D$ represent the design matrix of input data with $n$ rows, where $n$ is the total number of experiments performed
with the GCM, here 60. The number of columns, $p$, is defined by the number of dimensions in input parameter
space. In this case, $p = 4$ representing the three orbital parameters and atmospheric $CO_2$ concentration. A more
detailed explanation of the orbital input parameters is included in Sect. 3; however, briefly, they are longitude of
perihelion ($\varpi$), obliquity ($\varepsilon$) and eccentricity ($e$), with longitude of perihelion and eccentricity being combined
under the form $e\sin\varpi$ and $e\cos\varpi$. For a set of $i=1, n$ simulations, each simulation represents a point in input space,
and is characterised by the input vector $x_i$, i.e. a row of $D$.
The corresponding GCM climate data output is denoted $f(x_i)$, where the function $f$ represents the GCM model.
This output for all $n$ experiments is contained in the matrix $Y$. The raw output from the GCM is in the form of
gridded data covering the Earth's surface, with 96 longitude by 73 latitude grid boxes. We perform a principal
component analysis, to reduce the dimension of the output data before it is used to calibrate the emulator. Each
column of $Y$ contains the results for one experiment, i.e. $Y = [y(x_1), ..., y(x_n)]$. Furthermore, the centred matrix





$Y^*$ can be defined as $Y - Y_{mean}$, where $Y_{mean}$ is a matrix in which each row comprises a set of identical elements
that are the row averages of $Y$. The singular value decomposition (SVD) of $Y^*$ is:

$$Y^* = USV^{T*}, \tag{1}$$

where $S$ is the diagonal matrix containing the corresponding eigenvalues of $V$, $V$ is a matrix of the right singular
vectors of $Y$, and $U$ is a matrix of the left singular vectors. $U$ and $V$ are orthonormal, and $V^{T*}$ denotes the conjugate
transpose of the unitary matrix $V$. The columns of $US$ represent the principal components, and the columns of $V$
the principal directions/axes. Each column of $U$ represents an eigenvector, $u_k$, and $VS$ provides the projection
coefficients $\beta_k$. Specifically, for experiment $i$, $a_k(x_i) = \sum_k V_{ik} S_{kk}$ gives the projection coefficient for the $k$th
eigenvector. The eigenvectors are ordered by decreasing eigenvalue, and in practice only a relatively small number
of the eigenvectors will be retained ($n'$), typically selected on the basis of the largest values of $a_k(x)$. Thus:

$$y(x) = \sum_{k=1}^{n'} a_k(x) u_k, \tag{2}$$

We calibrate the emulator using the reduced dimension output data rather than the raw spatial climate data.
However, for simplicity, we will first consider a simple GP emulator. For this, the model output $f(x)$ for the input
conditions $x$ is modelled as a stochastic quantity that is defined by a Gaussian process. Its distribution is fully
specified by its mean function, $m(x)$, and its covariance function, $V(x, x')$, which may be written:

$$f(x) = GP[m(x), V(x, x')], \tag{3}$$

The mean and covariance functions take the form:

$$m(x) = h(x)^{\mathrm{T}} \beta, \tag{4}$$

$$V(x, x') = \sigma^2 [c(x, x')], \tag{5}$$

where $h(x)$ is a vector of known regression functions of the inputs, $\beta$ is a column vector of regression coefficients
corresponding to the mean function, $c(x, x')$ is the GP correlation function and $\sigma^2$ is a scaling value for the
covariance function. $h(x)$ and $\beta$ both have $q$ components and, as before, $^{\mathrm{T}}$ denotes the transpose operation.

A range of options are available for the regression functions $h(x)$ and the GP correlation function $c$, the most
suitable of which depends on the application of the emulator. Any existing knowledge that the user may have
about the expected response of the GCM to the input parameters can be used to inform their function choices.
However, if the emulator performs poorly, an alternative function can be selected which may prove to be more
suitable.

We assume a linear model, $h(x)^T = (1, x^T)$, with any non-linearities in the GCM response being absorbed by the
stochastic component of the GP. The correlation function is exponential decay with a nugget, a detailed discussion
of which can be found in Andrianakis and Challenor (2012). Hence, for the input parameters $a=1, p$, the correlation
function can be written as:

$$c(x, x') = exp\left[-\sum_{a=1}^{p}\left\{\frac{(x_a - x'_a)}{\delta_a}\right\}^2\right] + v I_{x=x'}, \tag{6}$$



where δ is the correlation length hyperparameter for each input, $v$ is the nugget term, and $I$ is an operator which is
equal to 1 when $x = x'$, and 0 otherwise. The nugget term has a number of functions in this application, including
accounting for any non-linearity in the output response to the inputs and for non-explicitly specified inactive
inputs, such as initial conditions and experiment, and averaging length. It also represents the effects of lower-
order PCs that are excluded from the emulator.

Now consider run $i$, which has inputs characterised by $x_i$ and outputs by $y_i$. Let $H$ be the design matrix relating to
the GCM output, where row $i$ represents the regressors $h(x_i)$, making $H$ an $n$ by $q$ matrix. The adopted modelling
approach states that the prior distribution of $y$ is Gaussian, characterised by $y \sim N(H\beta, \sigma^2 A)$, with $A_{ij} =$
$c(x_i, x_j)$.

Following the specification of the prior model above, a Bayesian approach is now used to update the prior
distribution. The posterior estimate of the GCM output is described by:

$$m^*(x) = h(x)^T \widehat{\beta} + t(x)A^{-1}(y - H\widehat{\beta}), \tag{7}$$

$$V^*(x, x') = \sigma^2[c(x, x') - t(x)^T A^{-1} t(x') + P(x)(H^T A^{-1} H)^{-1} P(x')^T], \tag{8}$$

where

$$\sigma^2 = (n - q - 2)^{-1}(y - H\widehat{\beta})^T A^{-1}(y - H\widehat{\beta}), \tag{9}$$

$$\widehat{\beta} = (H^T A^{-1} H)^{-1} H^T A^{-1} y, \tag{10}$$

and $t(x)_i = c(x, x_i)$ and $P(x) = h(x)^T - t(x)^T A^{-1} H$.

We follow the suggestion of Berger et al. (2001) and assume a vague prior $(\beta, \sigma^2)$ which is proportional to $\sigma^2$, an
approach that has been adopted by several other studies, including Oakley and O'Hagan (2002), Bastos and
O'Hagan (2009), Araya-Melo et al. (2015) and Bounceur et al. (2015). The posterior distribution of the GCM
output is a student-t distribution with $n - q$ degrees of freedom, but is sufficiently close to being Gaussian for this
application.

Now, taking the output from the PCA performed earlier, we apply the GP model to each basis vector ($a_k(x)$),
which has been updated according to Eq. 7 and 8, in turn. Thus:

$$a_k(x) = GP[m_k(x), V_k(x, x')], \tag{11}$$

where mean and covariance functions take the form:

$$m(x) = \sum_{k=1}^{n'} m_k(x) u_k, \tag{12}$$

$$V(x, x') = \sum_{k=1}^{n'} V_k(x, x') u_k u_k^T + \sum_{k=n'+1}^{n} \frac{s_{kk}^2}{n} u_k u_k^T, \tag{13}$$

The values of the hyperparameters are chosen by maximising the likelihood of the emulator, following Kennedy
and O'Hagan (2000), and based on the following expression from Andrianakis and Challenor (2012):

$$logL(v, \delta) = -\frac{1}{2}(\log(|A||H^T A^{-1} H|) + (n - q)\log(\hat{\sigma}^2)) + K, \tag{14}$$

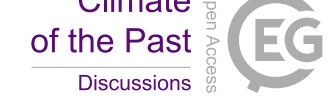



where $K$ is an unspecified constant. On the recommendation of Andrianakis and Challenor (2012), a penalised
likelihood is used, which limits the amplitude of the nugget:

$$logL^P(v, \delta) = logL(v, \delta) - 2\frac{\bar{M}(v,\delta)}{\epsilon\bar{M}(\infty)}, \tag{15}$$

where $\bar{M}(v,\delta)$ is the Mean Squared Error between the GCM's output data and the emulator's posterior mean at the
design points, defined by $\bar{M}(v, \delta) = v^2/n(\boldsymbol{y} - \boldsymbol{H}\boldsymbol{\beta})^T \boldsymbol{A}^{-2}(\boldsymbol{y} - \boldsymbol{H}\boldsymbol{\beta})$. $\bar{M}(\infty)$ is its asymptotic value at $\delta_i \to \infty$, given
by $\bar{M}(\infty) = 1/n(\boldsymbol{y} - \boldsymbol{H}\boldsymbol{\beta})^T(\boldsymbol{y} - \boldsymbol{H}\boldsymbol{\beta})$. $\epsilon$ is assigned a value of 1.

To summarise, in this study $\boldsymbol{D}$ is a 60 x 4 matrix ($n$ x $p$) of input data, consisting of 60 GCM simulations and four
input factors ($\varepsilon$, $e\sin\varpi$, $e\cos\varpi$, and $CO_2$). The matrix $\boldsymbol{Y}$ contains the output data from the GCM, with dimensions
of 96 x 73 x 60 (longitude x latitude x $n$). A PC analysis is performed on this output data, which is then used to
calibrate the emulator. Four hyperparameters ($\delta$) are used, due to there being four input factors, along with a
nugget term ($v$). The optimal values for these hyperparameters and the number of PCs retained are calculated
during calibration and evaluation of the emulator, discussed in Sect. 4. The GCM data used in this study are mean
annual SAT, and mean annual precipitation.
**3 AOGCM simulations**
**3.1 Model description**
To run the GCM simulations, we used the HadCM3 climate model (Gordon et al., 2000; Pope et al., 2000) – a
coupled atmosphere-ocean general circulation model (AOGCM) developed by the UK Met Office. Although
HadCM3 can no longer be considered as state-of-the-art when compared with the latest generation of GCMs, such
as those used in the most recent IPCC Fifth Assessment Report (IPCC, 2013), its relative computational efficiency
makes it ideal for running experiments for comparatively long periods of time (of several centuries) and for
running large ensembles of simulations, as performed in this study. As a result, this model is still widely used in
climate research, both in palaeoclimatic studies (e.g. Prescott et al., 2014) and in projections of future climate
(Armstrong et al., 2016). In addition, it has previously been employed in research into climate sensitivity using a
statistical emulator (Araya-Melo et al., 2015). The horizontal resolution of the atmosphere component is 2.5º
latitude by 3.75º longitude with 19 vertical levels, whilst the ocean has a resolution of 1.25º by 1.25º and 20
vertical levels.

HadCM3 is coupled to the land surface scheme MOSES2.1 (Met Office Surface Exchange Scheme), which was
developed from MOSES1 (Cox et al., 1999). It has been used in a wide range of studies (Cox et al., 2000; Crucifix
et al., 2005), and a comparison to MOSES1 and to observations is provided by Valdes et al. (2017). MOSES2.1
in turn is coupled to the dynamic vegetation model TRIFFID (Top-down Representation of Interactive Foliage
and Flora Including Dynamics) (Cox et al., 2002). TRIFFID calculates the global distribution of vegetation based
on five plant functional types: broadleaf trees, needleleaf trees, C3 grasses, C4 grasses and shrubs. Further details
of the overall model setup, denoted HadCM3M2.1E, can be found in Valdes et al. (2017).



### 3.2 Experimental design


In our simulations, four input parameters are varied: atmospheric $CO_2$ concentration and the three main orbital
forcings of longitude of perihelion ($\varpi$), obliquity ($\varepsilon$) and eccentricity ($e$). The extents of the GIS and WAIS are
also modified, although only between two modes – their present-day configurations and their reduced-extent
Pliocene configurations (Haywood et al., 2016). A more detailed description of the continental ice sheet
configurations is provided in Sect. 3.5.

We combined eccentricity and longitude of perihelion under the forms $e\sin\varpi$ and $e\cos\varpi$ given that, in general at
any point in the year, insolation can be approximated as a linear combination of these terms (Loutre, 1993). The
ranges of orbital and $CO_2$ values considered are appropriate for the next 1 Ma and a range of anthropogenic
emissions scenarios. For the astronomical parameters, calculated using the Laskar et al. (2004) solution, this
essentially equates to their full ranges of -0.055 to 0.055 for $e\sin\varpi$ and $e\cos\varpi$, and 22.2º to 24.4º for $\varepsilon$.

For $CO_2$, an emissions scenario is selected from Lord et al. (2016) in which atmospheric $CO_2$ follows observed
historical concentrations from 1750 AD (Anno Domini) to 2010 AD (Meinshausen et al., 2011), after which
emissions follow a logistic trajectory, resulting in cumulative total emissions of 10,000 Pg C by year ~3200. This
experiment was run for 1 Ma using the cGENIE Earth system model, and aims to represent a maximum total
future $CO_2$ release. To put this into perspective: current estimates of remaining fossil fuel reserves are
approximately 1000 Pg C, with an estimated ~4000 Pg C in fossil fuel resources that may be extractable in the
future (McGlade and Ekins, 2015), and up to 20-25,000 Pg C in nonconventional resources such as methane
clathrates (Rogner, 1997). The evolution of atmospheric $CO_2$ concentration over the next 200 kyr for this
emissions scenario is show in Fig. 1. Although in the *c*GENIE simulation, atmospheric $CO_2$ reaches a maximum
of 3900 parts per million (ppm) within the first few hundred years, this concentration is not at equilibrium and
only lasts for a couple of decades before decreasing. As a result, the concentration at 500 years into the experiment,
3600 ppm, is chosen as the upper $CO_2$ limit, which means that the climatic effects of emissions of more than
10,000 Pg C cannot be estimated with the emulator.

By the end of the 1 Ma emissions scenario, atmospheric $CO_2$ concentrations have nearly declined to pre-industrial
levels, reaching 285 ppm. However, this experiment does not account for natural variations in the carbon cycle,
which resulted in atmospheric $CO_2$ varying between 260 and 280 ppm during the Holocene (11 kyr BP to ~1750
AD) (Monnin et al., 2004). A value of 250 ppm is therefore deemed to be appropriate to account for these natural
variations, in addition to possible uncertainties in the model and hence is assumed as the value of the lower $CO_2$
limit in the ensemble.

The orbital and $CO_2$ parameter ranges that have been selected are also applicable to the late Pliocene, when
atmospheric $CO_2$ was estimated to be higher than pre-industrial values (Raymo et al., 1996). In this study, we do
not consider or attempt to simulate past or future glacial episodes, which may be accompanied by larger
continental ice sheets, although the conditions required to initiate the next glaciation, and extending the ensemble
of GCM simulations to represent glacial states, are being investigated in a separate study. The underlying



assumption of our ensemble is that it is suitable for simulating periods for which the $CO_2$ concentration is high
enough to prevent entry into a glacial state.

Two ensembles were generated, each made up of 40 simulations, meeting the recommended 10 experiments per
input parameter (Loeppky et al., 2009). One ensemble includes orbital values suitable for the next 1 Ma and a
relatively small range of lower $CO_2$ values, whereas the other ensemble represents the shorter-term future with a
reduced range of orbital values and a larger range of higher $CO_2$ concentrations. This approach was adopted
because various studies have shown that on geological timescales of thousands to hundreds of thousands of years,
an emission of fossil fuel $CO_2$ to the atmosphere is removed by natural carbon cycle processes over different
timescales (Archer et al., 1997; Lord et al., 2016). A relatively large fraction of the $CO_2$ perturbation is neutralised
on shorter timescales of $10^3$-$10^4$ years, but it takes $10^5$-$10^6$ years for atmospheric $CO_2$ concentrations to very
slowly return to pre-industrial levels (Colbourn et al., 2015; Lenton and Britton, 2006; Lord et al., 2016). Hence,
only a relatively short portion of the full million years has very high $CO_2$ concentrations under any emissions
scenario, with the major part of the time having a $CO_2$ concentration no more than several hundred ppm above
pre-industrial, as demonstrated in Fig. 1.

The parameter ranges for the two ensembles, which are referred to as "*highCO₂*" and "*lowCO₂*", are given in
Table 1. The cut-off point for the *highCO₂* ensemble is set at 110 kyr AP, as after this time eccentricity, which
remained relatively low prior to this time, starts to increase more rapidly and variability in $e\sin\varpi$ and $e\cos\varpi$
increases. This first ensemble therefore has $CO_2$ sampled up to 3600 ppm, and the orbital parameters are sampled
within the reduced range of values that will occur over the next 110 kyr. The *lowCO₂* ensemble samples the full
range of orbital values and the upper $CO_2$ limit is set to 560 ppm. This upper limit also covers the range of $CO_2$
concentrations that have been estimated for the late Pliocene (e.g. Martinez-Boti et al., 2015; Seki et al., 2010).
At 110 kyr in the 10,000 Pg C emissions scenario, the atmospheric $CO_2$ concentration is 542 ppm, which is
rounded up to twice the pre-industrial atmospheric $CO_2$ concentration (560 ppm = 2*280 ppm), a common
scenario used in future climate-change modelling studies.

The benefits of the approach of having separate ensembles for high and low $CO_2$ mean that both parameter ranges
have sufficient sampling density, whilst also reducing the chance of unrealistic sets of parameters, in particular
for the period of the next 110 kyr. During this time, $CO_2$ is likely to be comparatively high, while eccentricity
remains relatively low, and $e\sin\varpi$ and $e\cos\varpi$ exhibit relatively low variability. Having a separate ensemble in
which $CO_2$ and the orbital parameters are only sampled within the ranges experienced within the next 110 kyr
avoids wasting computing time on parameter combinations that are highly unlikely to occur, such as very high
$CO_2$ and very high eccentricity. This methodology also provides the additional benefit of the low $CO_2$ emulator
being applicable to palaeo-modelling studies, as the ensemble encompasses an appropriate range of $CO_2$ and
orbital values for many past periods of interest, such as the Pliocene.
**3.3 Generation of experiment ensembles**
We used the Latin hypercube sampling function from the MATLAB Statistics and Machine Learning Toolbox
(LHC; (MATLAB, 2012b)) to generate the two ensembles. This is a statistical method that efficiently samples the

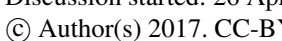



four-dimensional input parameter space (Mckay et al., 1979). Briefly, this method works by dividing the
parameter space within the prescribed ranges into $n$ equally probable intervals, $n$ being the number of experiments
required, which in this case is 40 per ensemble. $n$ points are then selected for each input variable, one from each
interval, without replacement. The sample points for the four variables are then randomly combined. The LHC
sampling function also includes an option to maximize the minimum distance between all pairs of points, which
is utilised here to ensure the set of experiments is optimally space filling. This is called the maxi-min criteria.

For each ensemble, 3000 sample sets were created, with each set consisting of an $n$ by $p$ matrix, $X$, containing the
four sampled input parameter values for each of the 40 experiments, and then the optimal sample set was selected
as the final ensemble based on a number of criteria. Following Joseph and Hung (2008), we seek, in addition to
the maxi-min criteria, to maximise $det(X^TX)$. Here, we will term this determinant the "orthogonality", because the
columns of the design matrix will indeed approach orthogonality as this determinant is maximised (assuming that
input factors are normalised). However, a limitation of the method of sampling the parameters $esin\varpi$ and $ecos\varpi$,
rather than eccentricity and longitude of perihelion directly, is that due to the nature of the $esin\varpi$ and $ecos\varpi$
parameter space, the sampling process favours higher values of eccentricity over lower ones. This is not an issue
for the longitude of perihelion, as when eccentricity is low the value of this parameter has little effect on insolation.
However, the value of obliquity selected for a given eccentricity value could have a significant impact on climate,
meaning that it is desirable to have a relatively large range of obliquity values for low (<0.01) and high (>0.05)
eccentricity values, in order to sample the boundaries sufficiently. It was observed that the sample sets with the
highest orthogonality had comparatively few, if any, values of low eccentricity, also meaning that a very limited
number of obliquity values were sampled for low eccentricity. We therefore adopted the approach whereby all
sample sets that demonstrated normalised orthogonality values that were more than 1 standard deviation above
the mean orthogonality were selected. From these, the single sample set with the greatest range of obliquity values
for low eccentricity, hence with maximal sampling coverage of the low eccentricity boundary, was selected as the
final ensemble design. The input parameter values for the $highCO_2$ and $lowCO_2$ ensembles are given in Table 2,
and the distributions in parameter space illustrated in Fig. 2.
**3.4 AOGCM simulations**
The two $CO_2$ ensembles were initially run with constant modern-day GIS and WAIS configurations (*modice*).
Atmospheric $CO_2$ and the orbital parameters were kept constant throughout each simulation, and each experiment
was run for a total of 500 model years. This run length allows the experiments with lower $CO_2$ to reach near-
equilibrium at the surface. Experiments with higher $CO_2$ have not yet equilibrated by the end of this period; the
significance of this is addressed in Sect. 3.6. A number of the very high $CO_2$ experiments caused the model to
become unstable and the interpretation of these experiments is discussed in Sect. 3.4.1. A control simulation was
also run for 500 years, with the atmospheric $CO_2$ concentration and the orbital parameters set at pre-industrial
values. All climate variable results for the model, unless specified, are an average of the final 50 years of the
simulation. Anomalies compared with the pre-industrial control (i.e. emulated minus pre-industrial) are discussed
and used in the emulator, rather than absolute values, to account for biases in the control climate of the model.



### 3.4.1 Very high $CO_2$ simulations

As mentioned previously, experiments in the *highCO₂* ensemble with $CO_2$ concentrations of greater than 3100 ppm become unstable. These experiments exhibit accelerating warming trends several hundred years into the simulation, which eventually cause the model to crash before completion. This is the result of a runaway positive feedback caused, at least in part, by the vertical distribution of ozone in the model being prescribed, rather than being able to respond to changes in climate, resulting in runaway warming as relatively high concentrations of ozone enter the troposphere.

All other experiments ran for the full 500 years. However, those with a $CO_2$ concentration of 2000 ppm or higher also exhibited accelerating warming trends before the end of the simulation. Consequently, only simulations with $CO_2$ concentrations of less than 2000 ppm (equivalent to a total fossil fuel $CO_2$ release of up to 6000 Pg C) are included in the rest of this study, meaning the methodology is not appropriate for $CO_2$ values greater than this. This equates to 20 experiments in total from the *highCO₂* ensemble, with $CO_2$ concentrations ranging from 303 to 1901 ppm. All 40 of the *lowCO₂* experiments were used.

### 3.5 Sensitivity to ice sheets

In addition to running the two ensembles with modern-day GIS and WAIS configurations, we also investigated the climatic impact of reducing the sizes of the ice sheets. Many of the $CO_2$ values sampled, particularly in the *highCO₂* ensemble, are significantly higher than pre-industrial levels, and if the resulting climate were to persist for long periods of time they could result in significant melting of the continental ice sheets over timescales of $10^3$-$10^4$ years (Charbit et al., 2008; Stone et al., 2010; Winkelmann et al., 2015).

We therefore set up the *highCO₂* and *lowCO₂* ensembles with reduced GIS and WAIS extents (*lowice*), using the PRISM4 Pliocene reconstruction of the ice sheets (Dowsett et al., 2016). In this reconstruction, the GIS is limited to high elevations in the Eastern Greenland Mountains, and no ice is present over Western Antarctica. Similar patterns of ice retreat have been simulated in response to future warming scenarios for the GIS (Greve, 2000; Huybrechts and de Wolde, 1999; Ridley et al., 2005; Stone et al., 2010) and WAIS (Huybrechts and de Wolde, 1999; Winkelmann et al., 2015), equivalent to ~7 m (Ridley et al., 2005) and ~3 m (Bamber et al., 2009; Feldmann and Levermann, 2015) of global sea level rise, respectively. Large regions of the East Antarctic ice sheet (EAIS) show minimal changes or slightly increased surface elevation, although there is substantial loss of ice in the Wilkes and Aurora subglacial basins (Haywood et al., 2016).

The same $CO_2$ and orbital parameter sample sets were used for both ice configuration ensembles to allow the impact of varying the ice-sheet extents on climate to be directly compared. Only the Greenland and Antarctic grid boxes were modified; the boundary conditions for all other grid boxes, as well as the land/sea mask, were the same as in the modern-day ice sheet simulations. For Greenland and Antarctica, the extent and orography of the ice sheets was updated with the PRISM4 data, as well as the orography of any grid boxes that are projected to be ice-free. Soil properties, land surface type and snow cover were also updated for these grid boxes. Figure 3 compares the orography for the *modice* and *lowice* ensembles, clearly showing the reduced extents for the ice sheets.

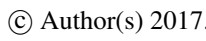



### 3.5.1 Pattern scaling of reduced ice simulations

It was expected that reducing the size of the continental ice sheets would have a relatively localised impact on climate, and that the effect would be of a linear nature. Therefore, a subset of five simulations from the two ensembles were selected as reduced ice-sheet simulations (*lowCO2* – experiments 8, 19 and 29; *highCO2* – experiments 21, and 34; see Table 2), covering a range of orbital and $CO_2$ values.

A comparison of the mean annual SAT anomaly for the five experiments showed that the largest temperature changes occur over Greenland and Antarctica, particularly in regions where there is ice in the *modice* ensemble but that are ice free in *lowice*. The spatial pattern of the change is also fairly similar across the simulations, suggesting that the response of climate to the extents of the ice sheets is largely independent of orbital variations or $CO_2$ concentration. The SAT anomaly for the five *lowice* experiments compared with their *modice* equivalents was calculated, and then averaged across the experiments, shown in Fig. 4. The largest SAT anomalies occur locally to the GIS and Antarctic ice sheet (AIS), accompanied by smaller anomalies in some of the surrounding ocean regions (e.g. Barents and Ross Seas), with no significant changes in SAT elsewhere, in line with the results of Lunt et al. (2004); Toniazzo et al. (2004) and (Ridley et al., 2005). This SAT anomaly, caused by the reduced extents of the GIS and WAIS, was then applied (added) to the mean annual SAT anomaly data for all other *highCO2* and *lowCO2 modice* experiments, to generate the SAT data for two *lowice* ensembles.

### 3.6 Calculation of equilibrated climate

Given the high values of $CO_2$ concentration in many of the experiments, particularly in the *highCO2* ensemble, even by the end of the 500 yr running period the climate has not yet reached steady state. We therefore calculated the fully equilibrated climate response using the methods described below.

### 3.6.1 Gregory plots

In order to estimate the equilibrated response, we applied the method of Gregory et al. (2004) to the model results, regressing the net radiative flux at the top of the atmosphere (TOA) against the global average SAT change, as displayed in figures termed Gregory plots (Andrews et al., 2015; Andrews et al., 2012; Gregory et al., 2015). In this method, for an experiment which has a constant forcing applied (i.e. with no inter-annual variation) it can be assumed that:

$$N = F - \alpha \Delta T, \tag{16}$$

where $N$ is the change in the global mean net TOA radiative flux (W m$^{-2}$), $F$ is the effective radiative forcing (W m$^{-2}$; positive downwards), $\alpha$ is the climate feedback parameter (W m$^{-2}$ °C$^{-1}$), and $\Delta T$ is the global mean annual SAT change compared with the control simulation (°C). This method works on the assumption that if $F$ and $\alpha$ are constant, $N$ is an approximately linear function of $\Delta T$. By linearly regressing $\Delta T$ against $N$, both $F$ (intercept of the line at $\Delta T = 0$) and $-\alpha$ (slope of the line) can be diagnosed. The intercept of the line at $N = 0$ provides an estimate of the equilibrium SAT change (relative to the pre-industrial SAT) for the experiment, denoted $\Delta T_{eq}{}^{g}$ to indicate it was calculated from the Gregory plots, and is equal to $F/\alpha$. This is in contrast to the SAT change calculated directly from the GCM model data by averaging the final 50 years of the experiment ($\Delta T_{500}$).



The Gregory plots for two *modice* experiments, *modice_lowCO2_13* ($CO_2$ 555.6 ppm) and *modice_highCO2_17*
($CO_2$ 1151.6 ppm), are shown in Fig. 5. These experiments were selected as they have $CO_2$ values nearest to the
2x and 4x pre-industrial $CO_2$ scenarios that are commonly used in idealised future climate experiments. For each
experiment, mean annual data are plotted for years 1-20 of the simulation, and mean decadal data for years 21-
500. The regression fits are to mean annual data in each case, and years 1-20 and 21-500 were fitted separately.
The values for $F$ and $\alpha$ estimated from Fig. 5 are presented in Table 3. These values are slightly lower than those
identified in other studies using the same method. For example, Gregory et al. (2004) used HadCM3 to run
experiments with 2x and 4x$CO_2$, obtaining values for years 1-90 of 3.9 ± 0.2 and 7.5 ± 0.3 W m$^{-2}$ for $F$, and -1.26
± 0.09 and -1.19 ± 0.07 W m$^{-2}$ °C$^{-1}$ for $\alpha$, respectively. Andrews et al. (2015) calculated $F$ to be 7.73 ± 0.26 W m$^{-2}$
and $\alpha$ to be -1.25 W m$^{-2}$ °C$^{-1}$ for years 1-20 and -0.74 W m$^{-2}$ °C$^{-1}$ for years 21-100 for 4x$CO_2$ simulations using
HadCM3. The differences between our results and theirs may be due to the fact that we used MOSES2.1 and the
TRIFFID vegetation model, whereas they used MOSES1, which is a different land-surface scheme and does not
account for vegetation feedbacks.

The decrease in the climate response parameter ($\alpha$) as the experiment progresses suggests that the strength of the
climate feedbacks changes as the climate evolves over time. Consequently, the $\Delta T$ intercept ($N = 0$) for the first
20 years of the simulation underestimates the actual warming of the model. Over longer timescales, the slope of
the regression line becomes less negative, implying that the sensitivity of the climate system to the forcing
increases (Andrews et al., 2015; Gregory et al., 2004; Knutti and Rugenstein, 2015). This non-linearity has been
found to be particularly apparent in cloud feedback parameters, in particular shortwave cloud feedback processes
(Andrews et al., 2015; Andrews et al., 2012). A number of studies have attributed this strengthening of the
feedbacks to changes in the pattern of surface warming (Williams et al., 2008), mainly in the eastern tropical
Pacific where an intensification of warming can occur after a few decades, but also in other regions such as the
Southern Ocean (Andrews et al., 2015). The impact of variations in ocean heat uptake has also been suggested to
be a contributing factor (Geoffroy et al., 2013; Held et al., 2010; Winton et al., 2010).

We take the $\Delta T$ intercept ($N = 0$) for years 21-500 to give the equilibrium temperature change ($\Delta T_{eq}{}^g$) for the
experiments, equating to values of 4.3°C and 8.9°C for the 2x and 4x$CO_2$ scenarios in Fig. 5. A limitation of this
approach is that it assumes that the response of climate to a forcing is linear after the first 20 years, which has
been shown to be unlikely in longer simulations of several decades or centuries (Andrews et al., 2015; Armour et
al., 2013; Winton et al., 2010). However, a comparison of the difference in temperature response to upper- and
deep-ocean heat uptake and its contribution to the relationship between net radiative flux change ($N$) and global
temperature change ($\Delta T$) in Geoffroy et al. (2013) indicated that the method of Gregory et al. (2004) of fitting two
separate linear models to the early and subsequent ($N$, $\Delta T$) data gives a good approximation of $\Delta T_{eq}{}^g$, $F$ and $\alpha$ as
they have been calculated here. A study by Li et al. (2013) also found that, using the Gregory plot methodology,
$\Delta T_{eq}{}^g$ was estimated to within 10% of its actual value, obtained by running the simulation very close to equilibrium
(~6000 yr). However, this was using the ECHAM5/MPIOM model, meaning that it is not necessarily also true for
HadCM3.





Given that the slope of the 21-500 yr regression line appears to become shallower with time, the estimates of $\Delta T_{eq}{}^g$
should be taken as a lower limit of the actual equilibrated SAT anomaly. However, this tendency to flatten,
particularly as the $CO_2$ concentration is increased, further justifies our use of the Gregory methodology; by the
end of 500 years the high $CO_2$ experiments have not yet reached steady state, and even in the lower $CO_2$
experiments SAT is increasing very slowly, so will likely take a long time to reach equilibrium. It would therefore
not be feasible to run most of these experiments to steady state using a GCM, due to the associated computational
and time requirements. Furthermore, on longer timescales the boundary conditions (orbital characteristics and,
more importantly, atmospheric $CO_2$ concentrations) would have changed, such that, in reality, equilibrium would
never be attained.
**3.6.2 Equilibrated climate**
The final estimates of $\Delta T_{eq}{}^g$ for the *lowCO₂* and *highCO₂ modice* ensembles range from a minimum of -0.4ºC
($CO_2$ 264.5 ppm) to a maximum of 12.5ºC ($CO_2$ 1900.9 ppm). Figure 6 illustrates the difference between global
mean annual SAT anomaly calculated from the GCM model data ($\Delta T_{500}$) and calculated using the Gregory plot
($\Delta T_{eq}{}^g$). Experiments with $CO_2$ below or near to pre-industrial levels tended to reach equilibrium by the end of the
500 years making a Gregory plot unnecessary, hence $\Delta T_{eq}{}^g$ is taken to be the same as $\Delta T_{500}$ in these cases. As $CO_2$
increases, the data points in Fig. 6 deviate further from the 1:1 line. This is the result of the ratio between $\Delta T_{eq}{}^g$
and $\Delta T_{500}$ increasing, as the experiments grow increasingly far from equilibrium by the end of the GCM run with
increasing $CO_2$.

We next calculated the ratio between $\Delta T_{eq}{}^g$ and $\Delta T_{500}$ for each experiment ($\Delta T_{eq}{}^g/\Delta T_{500}$), which represents the
fractional increase in climate change still due to occur after the end of the 500 year model run in order for steady
state to be reached. To estimate the fully equilibrated climate anomaly, the spatial distribution of mean annual
SAT anomaly was multiplied by the $\Delta T_{eq}{}^g/\Delta T_{500}$ ratio. The ratio identified for each experiment is assumed to be
equally applicable to all grid boxes. The equilibrated global mean annual SAT anomaly ($\Delta T_{eq}$) for the *highCO₂*
and *lowCO₂ modice* ensembles is plotted against $\log(CO_2)$ in Fig. 7, along with $\Delta T_{500}$ for reference. The linear
nature of the plot increases our confidence that the Gregory methodology is suitable for our uses, given the
logarithmic relationship between SAT and $CO_2$ concentration. Also plotted on Fig. 7 are a number of lines
illustrating idealised relationships between $\Delta T_{eq}$ and $CO_2$ based on a range of climate sensitivities. The most recent
IPCC report suggested that the likely range for equilibrium climate sensitivity is 1.5°C to 4.5°C (IPCC, 2013),
hence sensitivities of 1.5°C, 3°C and 4.5°C have been plotted. The size of the correction required to calculate $\Delta T_{eq}$
from $\Delta T_{500}$ increases with increasing $CO_2$, and brings the final temperature estimates in line with the expected
response (red lines), further increasing our confidence. The $\Delta T_{eq}$ estimated for the experiments generally follows
the upper line, equivalent to an equilibrium climate sensitivity of 4.5°C, which is higher than a previous estimate
of 3.3ºC for HadCM3 (Williams et al., 2001). This difference may be due to our simulations being "fully
equilibrated" following the application of the Gregory plot methodology. In addition, Williams et al. (2001) used
an older version of HadCM3 and prescribed vegetation (MOSES1), whilst in this study interactive vegetation is
used (MOSES2.1 with TRIFFID).

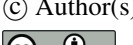



**4 Calibration and evaluation of the emulator**

By considering different contributions of modern and low ice, high and low $CO_2$, different number of PCs, and different values for the correlation length hyperparameters, we generated an ensemble of emulators, in order to test their relative performance. The *modice* and *lowice* ensembles were treated as independent data sets that were used separately when calibrating the emulator, since ice extent is not defined explicitly as an input parameter in the emulator code. $Log(CO_2)$ was used as one of the four input parameters, along with obliquity, *esinϖ* and *ecosϖ*. The performance of each emulator was assessed using a leave-one-out cross-validation approach, where a series of emulators is constructed, and used to predict one left-out experiment each time. For example, for the *lowCO₂ modice* ensemble (40 experiments), 40 emulators were calibrated with one experiment left out of each. This left-out experiment was then reproduced using the corresponding emulator, and the results compared with the actual experiment results. The number of grid boxes for each experiment calculated to lie within different standard deviation bands, and the root mean squared error (RMSE) averaged across all the emulators were used as performance indicators to compare the different input configurations and hyperparameter value selections. The results in this section are applicable to the *modice* emulator, unless otherwise specified, however the calibration and evaluation for the *lowice* emulator yielded similar trends and results.

**4.1 Sensitivity to input data**

We investigated the impact on performance of calibrating the emulator on the *highCO₂* and *lowCO₂ modice* ensembles separately, and combined. The *lowCO₂ modice* emulator generally performs slightly better in the leave-one-out cross-validation exercise than the *highCO₂ modice* version, with a lower RMSE and fewer grid boxes with an error of more than 2 standard deviations. Combining the two ensembles into one emulator results in a similar RMSE to the *lowCO₂*-only *modice* emulator but decreases the RMSE compared with the *highCO₂*-only *modice* emulator. As a consequence, we took the approach of calibrating the emulator on the combined ensembles for the rest of the study. This has the advantage that continuous simulations of climate with $CO_2$ levels that cross the boundary between the high and low $CO_2$ ensembles (~560 ppm), such as may be appropriate for emulation of future climate, can be performed using one emulator, rather than having to calibrate separate emulators for different time periods based on $CO_2$ concentration. There is also no loss of performance in the emulator for either set of $CO_2$ ranges, but rather a slight improvement for the *highCO₂* ensemble.

**4.2 Optimisation of hyperparameters**

We calibrated two separate emulators, the first using the *modice* data and the second using the *lowice* data, both with 60 experiments each (combined *highCO₂* and *lowCO₂*). The input factors ($\varepsilon$, *esinϖ, ecosϖ* and $CO_2$) were standardised prior to the calibration being performed; each was centred in relation to its column mean, and then scaled based on the standard deviation of the column. We tested different emulator configurations by varying the number of principal components retained, ranging from 5 to 20, and for each emulator configuration, the correlation length scales $\delta$ and nugget $\nu$ were optimized by maximization of the penalised likelihood. This optimisation was carried out in log-space, ensuring that the optimised hyperparameters would be positive. A leave-one-out validation was performed each time, and the *modice* and *lowice* configurations that performed best were selected as the final two optimised emulators. We found that a *modice* emulator retaining 13 principal components has the lowest RMSE and a relatively low percentage of grid boxes with errors of more than 2 standard deviations.





The scales $\delta$ for the *modice* emulator are 7.509 ($\varepsilon$), 3.361 ($e\sin\varpi$), 3.799 ($e\cos\varpi$), 0.881 ($CO_2$), and the nugget is
0.0631. In contrast, a *lowice* emulator using 15 principal components exhibits the best performance, with length
scales $\delta$ of 5.597 ($\varepsilon$), 2.887 ($e\sin\varpi$), 3.273 ($e\cos\varpi$), 0.846 ($CO_2$), and a nugget of 0.0925. In both cases, the scales
for the three orbital parameters are larger than the range associated with the input factors, indicating that the
response is relatively linear with respect to these terms.

The *modice* emulator was evaluated using the leave-one-out methodology and results are shown in Fig. 8. The
results suggest that the emulator performs well. Figure 8a shows the percentage of grid boxes for each left-out
experiment estimated by the corresponding emulator within different standard deviation bands, along with the
RMSE. The mean percentage of grid boxes within 1 and 2 standard deviations is 80% and 97%, which roughly
corresponds to the 68% and 95% ratios expected for a normal distribution, suggesting that the uncertainty in the
prediction is being correctly captured.

Several of the experiments performed considerably worse than others, exhibiting below the expected number of
grid boxes with errors within 1 standard deviation (for reference, the mean value for 1 standard deviation across
the left-out experiments is 0.3°C), and/or higher than the expected number of grid boxes with errors of greater
than 2 standard deviations, which is generally accompanied by a higher RMSE. However, the input conditions for
these experiments are not particularly similar or unique. Experiments *modice_highCO2_43*, *modice_highCO2_45*
and *modice_highCO2_46* all have a fairly low eccentricity and obliquity, and a $CO_2$ concentration of ~1000 ppm,
but there are multiple experiments with similar values that have lower RMSE values. A spatial map of the errors
(not shown) indicates that the grid boxes with errors of 3 or more standard deviations are at high northern latitudes
in these experiments. However, the signs of the anomalies are not the same across these experiments, as the
emulator overestimates the Arctic SAT anomaly in *modice_highCO2_43* and underestimates it in
*modice_highCO2_45* and *modice_highCO2_46*. This suggests that the emulator is perhaps not quite capturing the
full model behaviour in high northern latitudes, particularly for low eccentricity values, but this is certainly not
true for all experiments. The errors in the experiments are generally less than ±4°C, and for most of the Arctic
much lower than that. Note that the Arctic is a region in the model with high inter-annual variability, so one factor
may be that the model simulations which are used to calibrate the emulator are not representative of the true
stationary mean. There does not appear to be any obvious systematic error associated with the input parameters,
suggesting that errors are less likely to be an issue resulting from the design of the emulator and more likely to
arise from run-to-run variability in the behaviour of the underlying GCM.

Figure 8b compares the mean annual SAT index for each left-out experiment calculated by the GCM and the
corresponding emulator (Note: this is the mean value for the GCM output data grid assuming all grid boxes are
of equal size, hence not taking into account grid box area). There are no obvious outliers, and the emulated means
are relatively close to their modelled equivalents. There also does not appear to be any significant loss of
performance at very low or very high temperature, and therefore at very low or very high $CO_2$.

In summary, our calibration and evaluation shows that the emulator is able to reproduce the left-out ensemble
simulations reasonably well, with no obvious systematic errors in its predictions. Using the emulator, calibrated



on the full set of 60 simulations (*modice* or *lowice*), we are able to simulate global climate development over long
periods of time (several million years), provided that the atmospheric $CO_2$ levels for the period are known, and
are within the limits of those used to calibrate the emulator, ice sheets do not change outside the range considered
in the two ensembles, and the topography and land-sea mask are unchanged.

In the next two sections, we present illustrative examples of a number of potential applications of the emulator,
by applying it to the late Pliocene in Sect. 5, and the next 200 kyr in Sect. 6.

## 5 Application of the emulator to the late Pliocene

In addition to being able to rapidly project long-term climate evolution, the emulator also allows climatic changes
to be examined and analysed using a range of different methods that may not be possible using other modelling
approaches. To illustrate this, we applied the *lowice* emulator to the late Pliocene and compared the results to
palaeo-proxy data for the period. The *lowice* emulator was used because the ice sheets in this configuration are
the PRISM4 Pliocene ice sheets (Dowsett et al., 2016). We also tested the *modice* emulator which, in agreement
with the findings in Sect. 4, had a limited impact on the long-term evolution of global SSTs outside the immediate
region of the ice sheets themselves. Potential applications of the emulator for palaeoclimate are described below.

### 5.1 Time series data

One application of the emulator is to produce a time series of the continuous evolution of climate for a particular
time period, as is illustrated here where climate is simulated at 1 kyr intervals over the period 3300 – 2800 kyr
BP. This period of the late Pliocene was selected because it has been extensively studied as part of a number of
projects (e.g. PRISM (Dowsett et al., 2016; Dowsett, 2007), PlioMIP (Haywood et al., 2010; Haywood et al.,
2016)), represents the warm phase of climate (interglacial conditions), and does not include major glaciations like
the M2 cooling event, for which the emulator would not be appropriate. Orbital data for each of the time slices
(Laskar et al., 2004) were provided as input to the calibrated emulator, along with three representative $CO_2$
concentrations. Three $CO_2$ reference scenarios were initially emulated, with constant concentrations of 280, 350
and 400 ppm (although note that in reality, $CO_2$ varied during this period on orbital timescales (Martinez-Boti et
al., 2015)).

To illustrate the comparison of the emulator results to palaeo-proxy data, SST data for various locations were
compared with the emulated SAT for the equivalent grid box. Specifically, alkenone-derived palaeo-SST
estimates from four (Integrated) Ocean Drilling Program (IODP/ODP) sites were used: ODP Site 982 (North
Atlantic; (Lawrence et al., 2009)), IODP Site U1313 (North Atlantic; (Naafs et al., 2010)), ODP Site 722 (Arabian
Sea; (Herbert et al., 2010)) and ODP Site 662 (tropical Atlantic; (Herbert et al., 2010)). The locations of the sites
are shown in Fig. 9a and detailed in Table 4. These Pliocene datasets were selected because they are all of
sufficiently high resolution (≤4 kyr) for the impacts of individual orbital cycles on climate to be captured, whilst
covering a range of locations and climatic conditions. Alkenone data are shown converted to SST using two
commonly applied calibrations: Prahl et al. (1988) and Muller et al. (1998). All temperatures are presented as an
anomaly compared with pre-industrial. The emulator results are compared with the SAT for the relevant grid box





in the pre-industrial control experiment, whilst the proxy data are compared with SST observations for the relevant
location taken from the HadISST dataset (Rayner et al., 2003). Observations are annual means and are averaged
over the period 1870-1900.

For the modelled period, the emulator estimates the mean SAT anomaly compared with the pre-industrial control
in the 280 ppm scenario to be 0.6 ± 0.4°C, -0.8 ± 0.3°C, 0 ± 0.2°C, 0.2 ± 0.2°C for the two North Atlantic (982 and
U1313), Arabian Sea, and equatorial Atlantic grid boxes, respectively (Table 4). This mean increases with
increasing $CO_2$, by ~1°C at low latitudes to 2-3°C at high latitudes for atmospheric $CO_2$ of 400 ppm. Figure 10
illustrates the evolution of annual mean temperature variations through the late Pliocene as calculated using the
various methods. For the equatorial and Arabian Sea sites (662 and 722), the SAT and SST estimates are relatively
similar to each other, particularly for the higher $CO_2$ scenarios of 350 and 400 ppm. At the higher latitudes, the
simulated SAT estimate is generally lower than the proxy data SST. This is a common issue in GCM simulations
of the late Pliocene, where temperatures at high latitudes under increased $CO_2$-induced radiative forcing are often
underestimated (Haywood et al., 2013). It could also be that the alkenones are not recording mean annual
temperature, and instead are being produced during peak warmth (e.g. during the summer months), especially at
higher latitudes (Lawrence et al., 2009). This seasonal bias could explain the large offset in temperature at the
northernmost site (982), which exhibits a maximum difference in mean temperature anomaly for the period of
5.1°C between data sets, and possibly also at Site U1313. The emulated uncertainty in SAT is also shown in Fig.
10, and average values for the period given in Table 4. This is slightly higher at the northernmost North Atlantic
site (982) compared to the lower latitude sites, but overall the uncertainty is relatively small when compared with
the effects of variations in the orbital parameters and atmospheric $CO_2$ concentration.
**5.2 Orbital variability and spectral analysis**
The emulator can also be used to identify the influence of orbital variations on long-term climate change. One
approach is to assess the spatial distribution of orbital timescale variability, by plotting the standard deviation for
a climate variable for each grid box, as illustrated for SAT in Fig. 9 for the 400 ppm $CO_2$ scenario (blue lines in
Fig. 10). Figure 9a shows mean annual SAT (compared with pre-industrial) produced by the emulator under
modern-day orbital conditions. Anomalies over the majority of the Earth's surface are positive, due to the
relatively high atmospheric $CO_2$ concentration of 400 ppm. Warming is larger at high latitudes, primarily due to
a number of positive feedbacks operating in these regions (known as polar amplification). The greatest warming
is centred over parts of the GIS and WAIS, showing a similar spatial pattern to that in Fig. 4, and is a result of the
reduced ice sheet extents in the emulated experiments compared with the pre-industrial simulation. Figure 9b
shows the difference between modern-day emulated mean annual SAT (Fig. 9a) and emulated mean annual SAT
(compared with pre-industrial) averaged over the late Pliocene period (late Pliocene minus modern), whilst the
standard deviation of mean annual SAT for the late Pliocene is presented in Fig. 9c. In both Fig. 9b and 9c, spatial
variations primarily illustrate differences in the impact of orbital forcing on climate. For example, the relatively
higher values at high latitudes compared with low latitudes in Fig. 9c suggest that changes in the orbital parameters
have a relatively large impact on SAT in these regions. This is consistent with astronomical theory, as changes in
both obliquity and precession affect the distribution of insolation in space and time, with this effect being
particularly significant at high latitudes. Monsoonal regions also demonstrate relatively large variations (Fig. 9b





and 9c), including Africa, India, and South America, in agreement with previous studies which suggest a link
between orbital changes and monsoon variability (Caley et al., 2011; Prell and Kutzbach, 1987; Tuenter et al.,

671    2003).


In order to visualise the effects of orbital forcing over time, a spectral wavelet analysis was performed on the SAT
time series data produced by the emulator, for the scenario with constant $CO_2$ at 400 ppm, shown in Fig. 10 (blue
line). We used the standard MATLAB wavelet software of Torrence and Compo (1998) (available online
at http://atoc.colorado.edu/research/wavelets). The wavelet power spectra for the four ODP/IODP sites are
presented in Fig. 11, from which the dominant orbital frequencies influencing climate can be identified. For the
late Pliocene up to ~2900 kyr, Fig. 11 suggests that changes in emulated SAT are paced by a combination of
precession (longitude of perihelion) and eccentricity, with periodicities of approximately 21 and 96 kyr,
respectively. The influence of precession is also supported by the frequency of the SAT oscillations for this period
shown in Fig. 10, and it appears to have a larger impact on SAT at higher latitudes (Fig. 10 and 11). After ~2900
kyr, obliquity appears to have an increased impact at the high latitude site 982, superimposing the precession-
driven temperature variations with a periodicity of ~41 kyr (Fig. 10 and 11). This signal is also apparent to a lesser
extent at Site 722, but not at Site U1313. Spectral analysis of palaeo-proxy data and June insolation at 65$^o$ N also
finds a reduction in the influence of precession and an increase in 41 kyr obliquity forcing around this time
(Herbert et al., 2010; Lawrence et al., 2009). SAT changes at the lower latitude sites generally continue to be
dominated by variations in precession and eccentricity, although the relatively low eccentricity during this period
is likely to reduce the impact that precession has on climate. It also significantly reduces the variability in
temperature, which is also observed during the period of low eccentricity between approximately 3240 and 3200
kyr in both Fig. 10 and 11. The slightly higher amplitudes of the peaks in temperature around 3150 kyr, 3050 kyr
and 2950 kyr in Fig. 10 coincide with periods of high eccentricity, when its impact on climate is increased (Fig.
11). It is more difficult to identify orbital trends in the proxy data, particularly in sections with lower resolution.
This is due to there being significantly more variation, both on shorter timescales of several tens of thousands of
years, and longer timescales of hundreds of thousands of years, likely caused in part by changes in atmospheric
$CO_2$. However, the amplitude of variations in the palaeo data at all four sites is generally, though not always,
lower during periods of low eccentricity, particularly for the period ~3225-3200 kyr.
**5.3 Calculation of atmospheric $CO_2$**
We also illustrate the use of the emulator for calculating a simple estimate of atmospheric $CO_2$ concentration
during the late Pliocene, and its comparison to published palaeo $CO_2$ records obtained from proxy data. $CO_2$ is
estimated from the four alkenone SST records presented in Table 4 and Fig. 10: Herbert et al. (2010) (Sites 662
and 722), Naafs et al. (2010) (Site U1313) and Lawrence et al. (2009) (Site 982). A linear regression is performed
on the emulated grid box mean annual SAT data versus prescribed atmospheric $CO_2$ concentration, for the three
constant $CO_2$ scenarios of 280, 350 and 400 ppm. The $CO_2$ concentration is then estimated from the palaeo SST
data based on this linear relationship, and is presented in Fig. 12, along with the uncertainty. A number of $CO_2$
proxy records are also compared, derived from alkenone data at ODP Site 1241 in the east tropical Pacific (Seki
et al., 2010) and Site 999 in the Caribbean (Badger et al., 2013; Seki et al., 2010), and from boron ($\delta^{11}$B) data at
Site 662 (Martinez-Boti et al., 2015) and Site 999 (Bartoli et al., 2011; Martinez-Boti et al., 2015; Seki et al.,





2010). Our model-based $CO_2$ estimates suggest a mean atmospheric $CO_2$ concentration for the period of between
approximately $350 \pm 14$ and $540 \pm 17$ ppm (error represents the uncertainty taking into account the emulated gird
box posterior variance for SAT), indicated at Sites 722 and 982, respectively. Our estimates are generally higher
than the proxy records, particularly at the two North Atlantic sites (982 and U1313), where palaeo SST
temperatures were also estimated to be high, compared with tropical SSTs, by the proxy data (Fig. 10). However,
$CO_2$ concentrations derived from SST data calibrated using the approach of Prahl et al. (1988) at the tropical sites
of 722 and 662 shows greater similarity to the proxy data, both in terms of mean concentration and variance (not
shown). It is difficult to identify temporal similarities between our $CO_2$ estimates and the palaeo records. This is
partly due to the high level of variability in our $CO_2$ time series, resulting from the variability in the SST records
that they were derived from. In addition, the $CO_2$ proxy records have comparatively low resolutions, generally
with intervals of 10 kyr or greater, and there is also considerable variation between them.
**6 Application of the emulator to future climate**
In addition to using the emulator to model past climates, it can also be applied to future climate, and in particular
on the long timescales ($>10^3$ yr) that are of interest for the disposal of solid radioactive wastes. Previous modelling
of long-term future climate has involved the use of lower complexity models such as EMICs for transient
simulations (Archer and Ganopolski, 2005; Eby et al., 2009; Ganopolski et al., 2016; Loutre and Berger, 2000b),
or of GCMs to model a relatively small number of snapshot simulations of particular reference climate states of
interest. The BIOCLIM (Modelling Sequential Biosphere Systems under Climate Change for Radioactive Waste
Disposal) research programme (BIOCLIM, 2001, 2003), for example, utilised both of these approaches to
investigate climatic and vegetation changes for the next 200 kyr, for use in performance assessments for radiative
waste disposal facilities.

Here, for the first time, a GCM has been used to project future long-term transient climate evolution, via use of
the emulator. We provide illustrations of two possible applications of the emulator, including to produce a time
series of climatic data and to assess the impact of orbital variations on climate. This work has input to the
International Atomic Energy Agency (IAEA) MOdelling and DAta for Radiological Impact Assessments
(MODARIA) collaborative research programme (http://www-ns.iaea.org/projects/modaria/default.asp?l=116).
**6.1 Time series data**
Similarly to the late Pliocene, snapshots of SAT and precipitation at 1 kyr intervals were produced using the
*modice* emulator for the next 200 kyr, assuming modern day ice sheet configurations. The projected evolution of
climate is a result of future variations in the orbital parameters and atmospheric $CO_2$ concentrations, which were
provided as input data to the emulator (again, at 1 kyr intervals). Four $CO_2$ emissions scenarios were modelled,
with the response of atmospheric $CO_2$ concentration to emissions and its long-term evolution calculated using the
impulse response function of Lord et al. (2016). The scenarios adopted logistic $CO_2$ emissions of 500, 1000, 2000
and 5000 Pg C released over the first few hundred years, followed by a gradual reduction of atmospheric $CO_2$
concentrations by the long-term carbon cycle. These four scenarios cover the range of emissions that might occur





given currently economic and potentially economic fossil fuel reserves, but not including other potentially
exploitable reserves, such as clathrates.

Four single grid boxes are selected, shown in Fig. 13, which represent example locations that could potentially be
relevant for nuclear waste disposal: Forsmark, Sweden (60.4º N latitude, 18.2º E longitude), Central England, UK
(52.0º N latitude, 0º W longitude), Switzerland (47.6º N latitude, 8.7º E longitude) and El Cabril, Spain (38º N
latitude, 5.4º W longitude). The evolution of SAT at these grid boxes is presented in Fig. 14, along with the
emulated uncertainty (1 standard deviation). Across the four sites, the maximum SAT increase is between 4.1 ±
0.2°C (Switzerland grid box) and 12.3 ± 0.3°C (Spain grid box) in the 500 Pg C and 5000 Pg C scenarios,
respectively. For comparison, when the *lowice* emulator is utilized, these values are reduced slightly to 3.9 ± 0.3°C
(Spain grid box) and 12.2 ± 0.3°C (Spain grid box), respectively. This peak in temperature occurs up to the first
thousand years, when atmospheric $CO_2$ is at its highest following the emissions period, after which it decreases
relatively rapidly with declining atmospheric $CO_2$ until around 20 kyr AP. By 200 kyr AP, SAT at all sites is
within 2.6°C (2.2°C using the *lowice* emulator) of pre-industrial values, calculated by averaging the final 10 kyr
of the 5000 Pg C scenarios. The emulated uncertainty for the next 200 kyr is of a similar magnitude to that for the
late Pliocene and, similarly, is relatively small when compared with the fluctuations in SAT that result from orbital
variations and changing atmospheric $CO_2$ concentration.

Up until ~20 kyr AP, the behaviour of the climate is primarily driven by the high levels of $CO_2$ in the atmosphere
caused by fossil-fuel emissions and other human activities. However, after this time, changes in orbital conditions
begin to exert a relatively greater influence on climate, as the periodic fluctuations in SAT at all locations appear
to be paced by the orbital cycles, which are shown in Fig. 14a.

The timing and relative amplitudes of the oscillations in future SAT are in good agreement with a number of
previous studies. Paillard (2006) applied the conceptual model of Paillard and Parrenin (2004), previously
mentioned in Sect. 5, to the next 1 Ma. The development of atmospheric $CO_2$ over the next 200 kyr, simulated by
the model following emissions of 450 to 5000 Pg C and accounting for natural variations, shows a similar pattern
of response to that of SAT presented here. Estimates of global mean temperature in Archer and Ganopolski (2005),
derived by scaling changes in modelled ice volume to temperature, before applying anthropogenic $CO_2$
temperature forcing for a number of emissions scenarios, also demonstrate fluctuations in global mean annual
SAT (not shown) of a similar timing and relative scale. The influence of declining $CO_2$ is still evident after 20
kyr, particularly for the higher emissions scenarios, in the slightly negative gradient of the general evolution of
SAT. This is due to the long atmospheric lifetime of fossil fuel emissions (Lord et al., 2016), and is also
demonstrated in other studies (Archer and Ganopolski, 2005; Archer et al., 2009; Paillard, 2006). The impact of
excess atmospheric $CO_2$ on the long-term evolution of SAT appears to be fairly linear, with only minor differences
between the scenarios and sites, discounting the overall offset of SAT for different total emissions.

One of the key uncertainties associated with future climate change, which is of particular relevance to radioactive
waste repositories located at high northern latitudes, is the timing of the next glacial inception. This is expected
to occur during a period of relatively low incoming solar radiation at high northern latitudes, which, for the next



100 kyr, occurs at 0 kyr, 54 kyr and 100 kyr. A number of studies have investigated the possible timing of the
next glaciation under pre-industrial atmospheric $CO_2$ concentrations (280 ppm), finding that it is unlikely to occur
until after 50 kyr AP (Archer and Ganopolski, 2005; Berger and Loutre, 2002; Paillard, 2001).

When fossil fuel $CO_2$ emissions are taken into account, the current interglacial is likely to last significantly longer,
until ~130 kyr AP following emissions of 1000 Pg C and beyond 500 kyr AP for emissions of 5000 Pg C (Archer
and Ganopolski, 2005). A recent study by Ganopolski et al. (2016) using the CLIMBER-2 model found that
emissions of 1000 Pg C significantly reduced the probability of a glaciation in the next 100 kyr, and that a glacial
inception within the next 100 kyr is very unlikely for $CO_2$ emissions of 1500 Pg C or higher.

Our $CO_2$ emissions scenarios, modelled using the response function of Lord et al. (2016), suggest that atmospheric
$CO_2$ will not have returned to pre-industrial levels by 100 ka AP, equalling 298 and 400 ppm for the 500 and 5000
Pg C emissions scenarios, respectively. We calculated the critical summer insolation threshold at 65° N using the
logarithmic relationship identified between maximum summer insolation at 65° N and atmospheric $CO_2$ by
Ganopolski et al. (2016). The evolution of atmospheric $CO_2$ concentration over the course of our emissions
scenarios suggests that, for emissions of 1000 Pg C or less, Northern Hemisphere summer insolation will next fall
below the critical insolation threshold in approximately 50 ka, and in ~100 ka for emissions of 2000 Pg C. For the
highest emissions scenario of 5000 Pg C, the threshold is not passed for considerably longer, until ~160 ka.
However, the uncertainty of the critical insolation value is $\pm 4$ W m$^{-2}$ (1 standard deviation), and often the
difference between summer insolation at 65° N and the insolation threshold is less than this, potentially impacting
whether the threshold has in fact been passed and therefore whether glacial inception is likely. For example, for
the 1000 Pg C scenario, whilst insolation first falls below the critical threshold at ~50 ka, it does not fall below
by more than the uncertainty value until ~130 ka.

A limitation of our study relates to the continental ice sheets in HadCM3 being prescribed rather than responsive
to changes in climate. A consequence of this is that an increase in the extent or thickness of the ice sheets, and
hence the onset of glaciation, cannot be explicitly projected, but this also means that a regime shift of the ice
sheets to one of negative mass balance, which may be expected to occur under high $CO_2$ emissions scenarios
(Ridley et al., 2005; Stone et al., 2010; Swingedouw et al., 2008; Winkelmann et al., 2015), cannot be modelled.
However, the results of the sensitivity analysis to ice sheets described in Sect. 3.5., for which a number of
simulations were run again with reduced GIS and WAIS extents, suggest that the reduction in continental ice
results in relatively localised increases in SAT in regions that are ice free, in addition to some regional cooling at
high latitudes. Consequently, this does not act as a significant restriction on the glaciation timings put forward in
this study considering their radioactive waste disposal application; given that the earliest timing of the next
glaciation is of significant interest, smaller continental ice sheets and therefore higher local SATs would likely
inhibit the build-up of snow and ice, delaying glacial inception further. As such, the estimates presented here
should be viewed as conservative.

The emulator can also be used to project the evolution of a range of other climate variables, providing that they
were modelled as part of the initial GCM ensembles. Figure 15 illustrates the development of mean annual





precipitation and emulated uncertainty over the next 200 kyr at the four sites. The maximum increase in
precipitation is between $0.3 \pm 0.1$ mm day$^{-1}$ (Switzerland grid box) and $0.6 \pm 0.1$ mm day$^{-1}$ (Sweden grid box) in
the 500 Pg C and 5000 Pg C scenarios, respectively. Precipitation increases with increasing atmospheric $CO_2$ at
all sites apart from the Spain grid box, where it decreases by up to $0.9 \pm 0.1$ mm day$^{-1}$. Regional differences in the
sign of changes in precipitation, including an increase at high latitudes and a decrease in the Mediterranean, are
consistent with modelling results included in the International Panel on Climate Change (IPCC) Fifth Assessment
Report, for simulations forced with the Representative Concentration Pathway (RCP) 8.5 scenario (Collins et al.,
2013). In contrast to SAT, precipitation appears to be more closely influenced by precession, illustrated by its
periodicity of slightly less than 25 kyr; an increase in the intensity of precipitation fluctuations from approximately
140 kyr onwards suggest that the modulation of precession by eccentricity also has an impact, as expected.
**6.2 Orbital variability and spectral analysis**
The impact of orbital forcing was assessed by performing a spectral wavelet analysis on the SAT and precipitation
time series data produced by the emulator for the Central England grid box for the 5000 Pg C emissions scenario,
represented by blue lines in Fig. 14c and 15c, respectively. As for the late Pliocene, the wavelet software of
Torrence and Compo (1998) was utilized. The analysis was performed on the data for 20-200 kyr AP, because the
climate response up until ~20 kyr AP is dominated by the impact of elevated atmospheric $CO_2$ concentrations,
which masks the orbital signal and affects the results of the wavelet analysis.

For future SAT, Fig. 16a suggests that, up until ~160 kyr, the obliquity cycle acts as the dominant influence,
resulting in temperature oscillations with a periodicity of approximately 41 kyr. This is confirmed by Fig. 14c,
which shows that the major peaks in SAT generally coincide with periods of high obliquity. Over this period,
precession has a far more limited influence, likely due to eccentricity being relatively low until ~110 kyr (Fig.
14a). However, from ~120 kyr AP onwards, concurrently with increasing eccentricity, precession becomes a more
significant forcing on climate, resulting in SAT peaks approximately every 21 kyr. In contrast, precession appears
to be the dominant forcing on precipitation for the Central England grid box for the entire 20-200 kyr AP period
(Fig. 15c and 16b). This signal is particularly strong after ~120 kyr AP, due to higher eccentricity.
**7 Conclusions**
In this study, we present long-term continuous projections of future climate evolution at the spatial resolution of
a GCM, via the use of a statistical emulator. The emulator was calibrated on two ensembles of simulations with
varied orbital and atmospheric $CO_2$ conditions and modern day continental ice sheet extents, produced using the
HadCM3 climate model. The method presented by Gregory et al. (2004) to calculate the steady-state global
temperature change for a simulation, by regressing the net radiative flux at the top of the atmosphere against the
change in global SAT, was utilised to calculate the equilibrated SAT data for these ensembles, as it was not
feasible to run the experiments to equilibrium due to the associated time and computer resources needed. A
number of simulations testing the sensitivity of SAT to the extent of the GIS and WAIS suggest that the response
of SAT is fairly linear regardless of orbit, and that the largest changes are generally local to regions that are ice



free. The mean SAT anomaly identified across these experiments was then applied to the equilibrated SAT results
of the modern-day ice sheet extent ensembles, to generate two equivalent ensembles with reduced ice sheets.

Output data from the modern-day and reduced ice sheet ensembles were then used to calibrate separate emulators,
which were optimised and then validated using a leave-one-out approach, resulting in satisfactory performance
results. We discuss a number of useful applications of the emulator, which may not be possible using other
modelling approaches at the same temporal and spatial resolution. Firstly, a particular benefit of the emulator is
that it can be used to produce time series of climatic variables that cover long periods of time (i.e. several thousand
years or more) at a GCM resolution, accompanied by an estimation of the uncertainty in the form of the posterior
variance. This would not be feasible using GCMs due to the significant time and computational requirements
involved. The global grid coverage of the data also means that the evolution of a climate variable at a particular
grid box can be examined, allowing for comparisons to data at a regional or local scale, such as palaeo-proxy data,
or for the evolution of climate at a specific site to be studied. Secondly, the influence of orbital forcing on climate
can be assessed. This effect may be visualised with a continuous wavelet analysis on the time series data for a
particular $CO_2$ emissions scenario, which will identify the orbital frequencies dominating at different times. The
spatial distribution of orbital timescale variability can also be simulated, by plotting the standard deviation for a
climate variable for each grid box, taking into account the emulator posterior variance. Finally, the emulator can
be used to back-calculate past atmospheric $CO_2$ concentrations based on proxy climate data. Through an inversion,
atmospheric $CO_2$ concentrations can be estimated using SST proxy data, based on a linear relationship between
emulated grid box mean annual SAT and prescribed $CO_2$ concentration. Estimated $CO_2$ can then be compared
with palaeo $CO_2$ concentration proxy records.

To illustrate these potential applications, we applied the emulator at 1 kyr intervals to the late Pliocene (3300-
2800 kyr BP) for atmospheric $CO_2$ concentrations of 280, 350 and 400 ppm, and compared the emulated SATs at
specific grid boxes to SSTs determined from proxy data from a number of ODP/IODP sites. The wavelet power
spectrum for SAT at each site was also produced, and the dominant orbital frequency assessed. In addition, we
used the SST proxy data to estimate atmospheric $CO_2$ concentrations, based on a linear relationship between
emulated grid box mean annual SAT and prescribed $CO_2$ concentration. We find that:

- Temperature estimates from the emulator and proxy data show greater similarity at the equatorial sites
than at the high latitude sites. Discrepancies may be the result of biases in the GCM, errors in the
emulator, seasonal biases in the proxy data, unknown changes in the climate and/or carbon cycle, or
issues with the tuning of parts of the record.
- The response of emulated SAT appears to be dominated by a combination of precessional and
eccentricity forcing from 3300 kyr to approximately 2900 kyr, after which obliquity begins to have an
increased influence.
- Regions with a particularly large response to orbital forcing include the high latitudes and monsoon
regions (Fig. 9b and 9c).



- Our $CO_2$ reconstructions from tropical ODP/IODP sites show relatively similar concentrations to $CO_2$
proxy records for the same period, although for the higher latitude sites concentrations are generally
significantly higher than the proxy data.

The emulator was also applied to the next 200 kyr, as long-term future simulations such as these have relevance
to the geological disposal of solid radioactive wastes. The continuous evolution of mean annual SAT and
precipitation at a number of sites in Europe are presented, for four scenarios with fossil fuel $CO_2$ emissions of
500, 1000, 2000 and 5000 Pg C. A spectral wavelet analysis was also performed on the SAT and precipitation
data for the Central England grid box. The data suggests that:

- SAT and, to a lesser extent, precipitation exhibit a relatively rapid decline back towards pre-industrial
values over the next 20 kyr, as excess atmospheric $CO_2$ is removed by the long-term carbon cycle.
- Following this, SAT fluctuates due to orbital forcing on an approximate 41 kyr obliquity timescale until
~160 kyr AP, before the influence of precession increases with increasing eccentricity from ~120 kyr
AP.
- Conversely, precipitation variations over the entire 200 kyr period demonstrate a strong precessional
signal.

Overall, we find that the emulator provides a useful and powerful tool for rapidly simulating the long-term
evolution of climate, both past and future, due to its relatively high spatial resolution and relatively low
computational cost. We have presented illustrative examples of a number of different possible applications, which
we believe make it suitable for tackling a wide range of climate questions.

**Code availability**
Code for the Latin hypercube sampling function is available from the MATLAB Statistics and Machine Learning
Toolbox. The wavelet software of Torrence and Compo (1998) is available online
at http://atoc.colorado.edu/research/wavelets.
**Data availability**
The data used in this paper are available from Natalie S. Lord (Natalie.Lord@bristol.ac.uk).
**Competing interests**
The authors declare that they have no conflict of interest.



### Acknowledgements

This research is funded by RWM Limited via a framework contract with Amec Foster Wheeler, who are being supported by Quintessa. It contributes to the MODARIA international research programme, sponsored and coordinated by the International Atomic Energy Agency (IAEA). The ensembles of AOGCM simulations were run using the computational facilities of the Advanced Computing Research Centre, University of Bristol – http://www.bris.ac.uk/acrc/. Any use of trade, firm, or product names is for descriptive purposes only and does not imply endorsement by the U. S. Government.

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





**Table 1. Ensembles setup: sampling ranges for input parameters (obliquity, *e*sin$\varpi$, *e*cos$\varpi$ and CO₂) for the *highCO₂***
**and *lowCO₂* ensembles.**

| Ensemble | Time covered from present day (AP) | Parameter | Sampling range | |
|---|---|---|---|---|
| | | | Minimum | Maximum |
| *highCO₂* | 110 kyr | $\varepsilon$ (º) | 22.3 | 24.3 |
| | | $e\sin\varpi$ | -0.016 | 0.016 |
| | | $e\cos\varpi$ | -0.016 | 0.015 |
| | | CO₂ (ppm) | 280 | 3600 |
| *lowCO₂* | 1 Ma | $\varepsilon$ (º) | 22.2 | 24.4 |
| | | $e\sin\varpi$ | -0.055 | 0.055 |
| | | $e\cos\varpi$ | -0.055 | 0.055 |
| | | CO₂ (ppm) | 250 | 560 |

**Table 2. Experiment setup: Orbital parameters (obliquity, eccentricity and longitude of perihelion) and atmospheric**
**CO₂ concentration for simulations in the *highCO₂* and *lowCO₂* ensembles. All experiments in both ensembles were run**
**with modern ice sheet (*modice*) configurations. Experiments shown in bold were also run with reduced ice sheet (*lowice*)**
**configurations. The experiment number is given, and the experiment name is constructed using the ice sheet**
**configuration, the ensemble name and the experiment number, for example: modice_lowCO2_1.**

| Ensemble | # | $\varepsilon$ (º) | $e$ - | $\varpi$ (º) | CO₂ (ppm) | Ensemble | # | $\varepsilon$ (º) | $e$ - | $\varpi$ (º) | CO₂ (ppm) |
|---|---|---|---|---|---|---|---|---|---|---|---|
| *highCO₂* | 1 | 23.53 | 0.0093 | 240.3 | 3348.2 | *lowCO₂* | 1 | 22.99 | 0.0481 | 320.1 | 375.7 |
| | 2 | 24.24 | 0.0135 | 212.6 | 2159.3 | | 2 | 23.02 | 0.0323 | 63.7 | 516.9 |
| | 3 | 22.38 | 0.0110 | 260.0 | 1645.0 | | 3 | 22.81 | 0.0481 | 334.2 | 470.4 |
| | 4 | 24.07 | 0.0044 | 101.8 | 800.8 | | 4 | 24.03 | 0.0537 | 84.9 | 390.3 |
| | 5 | 23.07 | 0.0203 | 313.0 | 1999.9 | | 5 | 23.09 | 0.0294 | 293.8 | 325.3 |
| | 6 | 24.03 | 0.0087 | 184.9 | 3049.0 | | 6 | 23.58 | 0.0098 | 325.1 | 337.5 |
| | 7 | 22.53 | 0.0163 | 162.0 | 900.9 | | 7 | 23.72 | 0.0133 | 74.3 | 489.2 |
| | 8 | 23.57 | 0.0158 | 21.0 | 1746.3 | | **8** | **24.17** | **0.0066** | **174.1** | **346.0** |
| | 9 | 23.34 | 0.0131 | 113.5 | 996.8 | | 9 | 23.82 | 0.0400 | 48.2 | 260.6 |
| | 10 | 23.37 | 0.0198 | 220.2 | 3139.3 | | 10 | 23.39 | 0.0412 | 53.8 | 409.5 |
| | 11 | 22.73 | 0.0187 | 236.1 | 1081.9 | | 11 | 22.89 | 0.0531 | 115.2 | 436.6 |
| | 12 | 22.63 | 0.0121 | 184.8 | 2451.5 | | 12 | 23.34 | 0.0281 | 133.9 | 504.4 |
| | 13 | 22.41 | 0.0131 | 192.8 | 3372.4 | | 13 | 22.65 | 0.0473 | 102.6 | 555.6 |
| | 14 | 22.78 | 0.0137 | 299.3 | 448.2 | | 14 | 23.20 | 0.0368 | 180.9 | 385.1 |
| | 15 | 22.97 | 0.0111 | 14.1 | 1225.7 | | 15 | 23.96 | 0.0232 | 40.0 | 403.4 |
| | 16 | 22.90 | 0.0087 | 62.2 | 1841.9 | | 16 | 24.27 | 0.0460 | 298.1 | 341.1 |
| | 17 | 23.63 | 0.0151 | 200.6 | 1151.6 | | 17 | 22.35 | 0.0391 | 265.9 | 522.1 |
| | 18 | 23.77 | 0.0134 | 78.7 | 2101.7 | | 18 | 23.91 | 0.0361 | 343.2 | 318.6 |
| | 19 | 23.73 | 0.0159 | 323.7 | 1526.6 | | **19** | **22.33** | **0.0484** | **324.2** | **264.5** |
| | 20 | 24.29 | 0.0082 | 164.6 | 2890.4 | | 20 | 22.94 | 0.0350 | 268.7 | 540.8 |



| 21 | **22.31** | **0.0038** | **299.1** | **1389.5** | 21 | 22.68 | 0.0323 | 332.4 | 531.5 |
|----|-----------|------------|-----------|------------|----|-------|--------|-------|-------|
| 22 | 23.42 | 0.0117 | 122.5 | 397.3 | 22 | 24.28 | 0.0387 | 118.7 | 446.7 |
| 23 | 24.00 | 0.0101 | 206.6 | 303.4 | 23 | 23.60 | 0.0484 | 282.0 | 310.5 |
| 24 | 22.48 | 0.0146 | 294.9 | 2845.7 | 24 | 24.19 | 0.0337 | 346.3 | 548.3 |
| 25 | 22.57 | 0.0067 | 81.2 | 1341.2 | 25 | 24.14 | 0.0423 | 11.6 | 425.4 |
| 26 | 22.93 | 0.0171 | 114.4 | 3516.0 | 26 | 22.20 | 0.0035 | 85.2 | 303.0 |
| 27 | 24.13 | 0.0143 | 257.3 | 2951.8 | 27 | 22.78 | 0.0070 | 212.1 | 480.4 |
| 28 | 23.00 | 0.0062 | 272.2 | 2274.6 | 28 | 22.72 | 0.0526 | 239.9 | 280.0 |
| 29 | 23.95 | 0.0103 | 114.7 | 564.7 | 29 | **23.65** | **0.0543** | **30.3** | **362.0** |
| 30 | 23.17 | 0.0169 | 56.7 | 1900.9 | 30 | 23.24 | 0.0351 | 200.4 | 411.9 |
| 31 | 23.70 | 0.0122 | 1.4 | 773.0 | 31 | 23.87 | 0.0276 | 156.5 | 287.5 |
| 32 | 23.24 | 0.0021 | 310.2 | 2582.1 | 32 | 22.25 | 0.0499 | 208.9 | 365.3 |
| 33 | 22.81 | 0.0121 | 66.3 | 2386.5 | 33 | 22.54 | 0.0510 | 103.4 | 471.1 |
| **34** | **24.18** | **0.0145** | **36.6** | **668.2** | 34 | 22.58 | 0.0404 | 292.2 | 544.5 |
| 35 | 23.82 | 0.0075 | 10.8 | 2244.8 | 35 | 22.87 | 0.0530 | 20.9 | 498.2 |
| 36 | 23.14 | 0.0141 | 314.1 | 3588.9 | 36 | 23.53 | 0.0414 | 147.0 | 507.0 |
| 37 | 23.49 | 0.0121 | 101.5 | 2760.4 | 37 | 22.39 | 0.0165 | 149.1 | 393.9 |
| 38 | 22.66 | 0.0162 | 69.5 | 2623.9 | 38 | 22.43 | 0.0537 | 175.0 | 484.8 |
| 39 | 23.28 | 0.0146 | 207.5 | 1484.8 | 39 | 24.38 | 0.0482 | 342.9 | 418.3 |
| 40 | 23.89 | 0.0092 | 21.1 | 3188.8 | 40 | 23.76 | 0.0504 | 127.0 | 528.1 |

**Table 3. Parameter values estimated from Gregory plots for the 2x and 4x pre-industrial $CO_2$ simulations. Shown are the effective radiative forcing ($F$; W m$^{-2}$) and the climate feedback parameter ($\alpha$; W m$^{-2}$ $^\circ$C$^{-1}$) for years 1-20 and years 21-100. The uncertainties are the standard error from the linear regression.**

| Simulation | | $F$ | | $\alpha$ | |
|---|---|---|---|---|---|
| | | (W m$^{-2}$) | | (W m$^{-2}$ $^\circ$C$^{-1}$) | |
| | | yr 1-20 | yr 21-100 | yr 1-20 | yr 21-100 |
| 2xCO$_2$ | *modice_lowCO2_13* | 4.24 ± 0.4 | - | -1.30 ± 0.2 | -0.68 ± 0.05 |
| 4xCO$_2$ | *modice_highCO2_17* | 6.88 ± 0.3 | - | -0.99 ± 0.1 | -0.56 ± 0.02 |





**Table 4. Mean temperature anomalies and uncertainties (1 standard deviation) for the period 3300-2800 kyr BP**
**estimated by the emulator and alkenone proxy data for the four ODP/IODP sites.**

| ODP/IODP Site | Location | | | Emulated SAT anomaly (°C) | | | Proxy data SST anomaly (°C) | |
|---|---|---|---|---|---|---|---|---|
| | | Lat | Lon | 280 ppm | 350 ppm | 400 ppm | Prahl et al. (1988) | Muller et al. (1998) |
| 982[1] | North Atlantic | 57.5° N | 15.9° W | 0.6 ±0.4 | 2.4 ±0.3 | 3.3 ±0.3 | 5.4 | 5.7 |
| U1313[2] | North Atlantic | 41.0° N | 33.0° W | -0.8 ±0.3 | 0.0 ±0.2 | 0.8 ±0.2 | 1.6 | 2.0 |
| 722[3] | Arabian Sea | 16.6° N | 59.8° E | 0.0 ±0.2 | 1.0 ±0.2 | 1.7 ±0.2 | 1.0 | 1.7 |
| 662[3] | Tropical Atlantic | 1.4° S | 11.7° W | 0.2 ±0.2 | 0.9 ±0.2 | 1.3 ±0.2 | 1.3 | 1.9 |

[1]Lawrence et al. (2009); [2]Naafs et al. (2010); [3]Herbert et al. (2010).

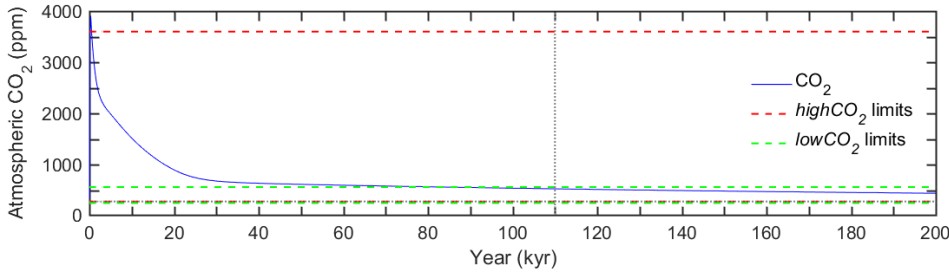

**Figure 1. Time series of atmospheric CO₂ concentration (ppm) for the next 200 kyr following logistic CO₂ emissions of**
**10,000 PgC, run using the cGENIE model (Lord et al., 2016). Also shown are the upper and lower CO₂ limits of the**
**_highCO₂_ (red dashed lines) and _lowCO₂_ (green dashed lines) ensembles. The pre-industrial CO₂ concentration of 280**
**ppm (horizontal grey dotted line), and the 110 kyr cut-off for the highCO₂ ensemble (vertical grey dotted line) are**
**included for reference.**





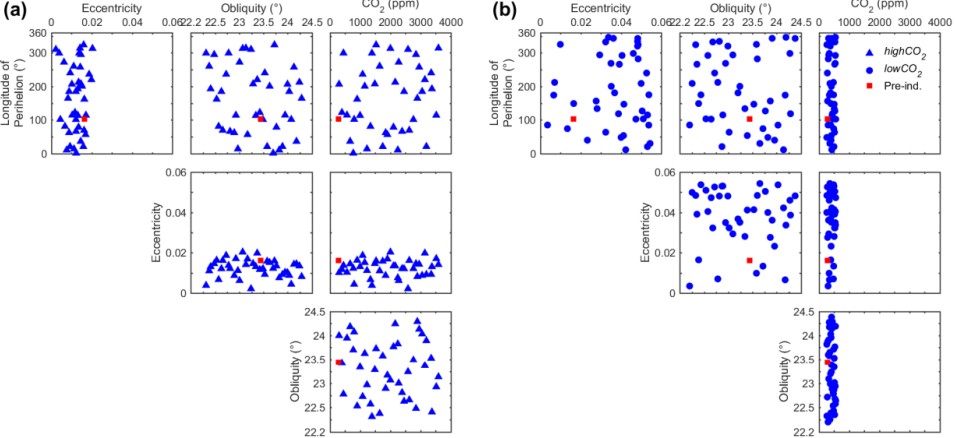

Figure 2. Distribution of 40 experiments produced by Latin hypercube sampling, displayed as two-dimensional slices through four-dimensional space. (a) *highCO₂* ensemble, (b) *lowCO₂* ensemble. The variables are eccentricity (*e*), longitude of perihelion ($\varpi$; degrees), obliquity ($\varepsilon$; degrees), and atmospheric $CO_2$ concentration (ppm). A pre-industrial control simulation is shown in red.

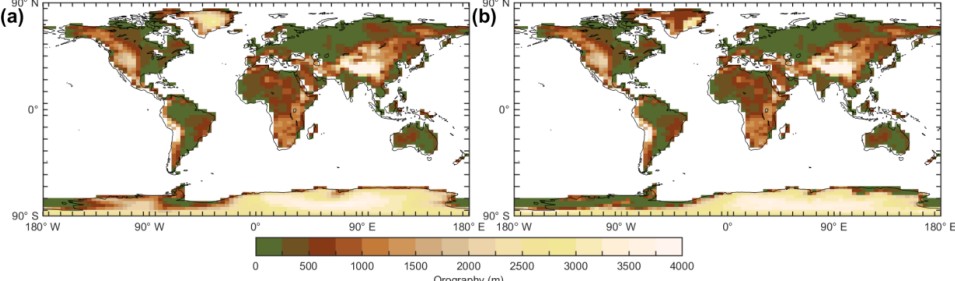

Figure 3. Orography (m) in the two ice sheet configuration ensembles. (a) *modice* ensemble, (b) *lowice* ensemble. Differences only occur over Greenland and Antarctica.

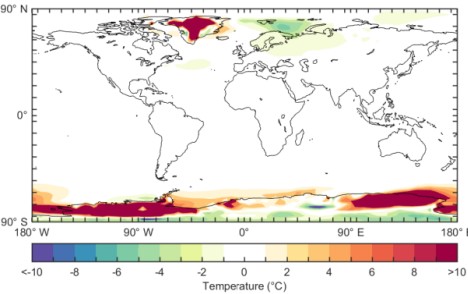

Figure 4. Mean annual SAT (ºC) anomaly for the lowice experiments compared with their *modice* equivalents, averaged across the five *lowice* experiments. All SAT anomalies have been calculated compared with the pre-industrial control simulation.

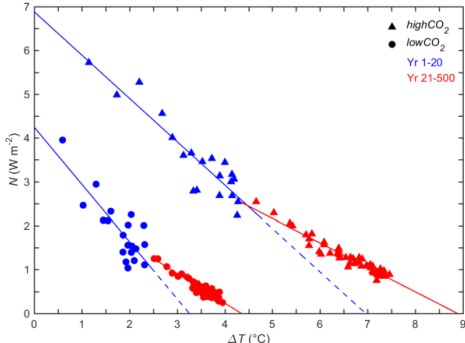

**Figure 5. Gregory plot showing change in TOA net downward radiation flux ($N$; W m$^{-2}$) as a function of change in global mean annual SAT ($\Delta T$; ºC) for approximate 2xCO$_2$ (modice_lowCO2_13; circles) and 4xCO$_2$ (modice_highCO2_17; triangles) experiments. Lines show regression fits to the global mean annual data points for years 1-20 (blue) and years 21-500 (red). Data points are annual data for years 1-20 (blue) and mean decadal data for years 21-500 (red). The $\Delta T$ intercepts ($N$=0) of the red lines give the estimated equilibrated SAT ($\Delta T_{eq}{}^g$) for the two experiments. The $\Delta T$ intercepts of the dashed blue lines represent the equilibrium that the experiment would have reached if the feedback strengths in the first 20 years had been maintained. SAT is shown as an anomaly compared with the pre-industrial control simulation.**

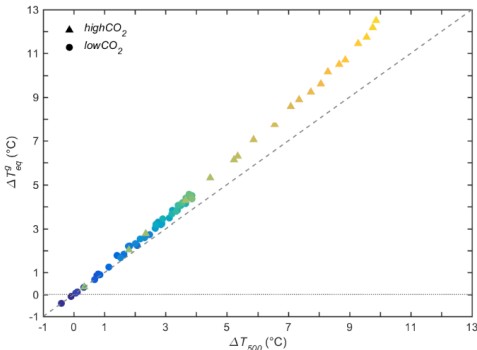

**Figure 6. Equilibrated global mean annual change in SAT ($\Delta T_{eq}{}^g$; ºC) estimated using the methodology of Gregory et al. (2004) against global mean annual change in SAT ($\Delta T_{500}$; ºC) at year 500 (average of final 50 years) for the *lowCO$_2$* (circles) and *highCO$_2$* (triangles) *modice* ensembles. The colours of the points indicate the CO$_2$ concentration of the experiment, from low (blue) to high (yellow). The 1:1 line (dashed) is included for reference. SAT is shown as an anomaly compared with the pre-industrial control simulation.**



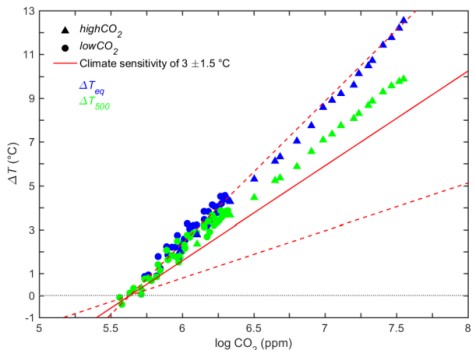

**Figure 7.** Equilibrated global mean annual change in SAT ($\Delta T_{eq}$; ºC; blue), estimated by applying the $\Delta T_{eq}{}^g/\Delta T_{500}$ ratio identified using the Gregory methodology to the GCM data, against atmospheric $CO_2$ (ppm) for the *lowCO₂* (circles) and *highCO₂* (triangles) *modice* ensembles. Also shown is $\Delta T_{500}$ (green), along with the idealized relationship between $\log(CO_2)$ and $\Delta T$ (red lines) for a climate sensitivity of 3ºC (solid), 1.5ºC (lower dashed) and 4.5ºC (upper dashed) (IPCC, 2013). SAT is shown as an anomaly compared with the pre-industrial control simulation.

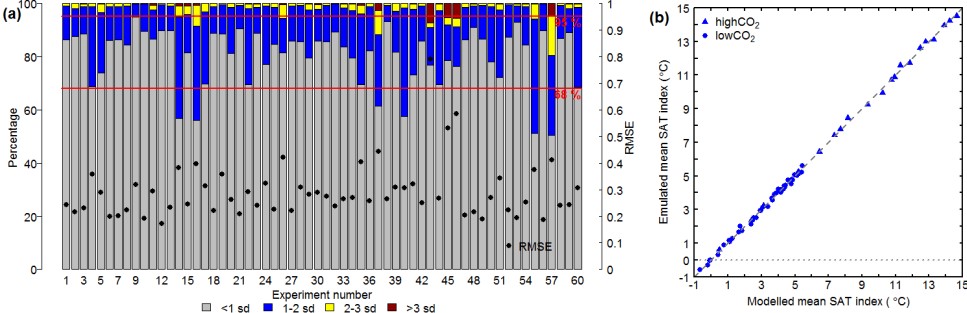

**Figure 8.** Evaluation of emulator performance. (a) Bars give the percentage of grid boxes for which the emulator predicts the SAT of the left-out experiment to within 1, 2, 3 and more than 3 standard deviations (sd). Also shown is the RMSE for the experiments (black circles). Red lines indicate 68% and 95%. (b) Mean annual SAT index (ºC) calculated by the emulator and the GCM for the *lowCO₂* (circles) and *highCO₂* (triangles) *modice* ensembles. The 1:1 line (dashed) is included for reference. Note: this is the mean value for the GCM output data grid assuming all grid boxes are of equal size, hence not taking into account variations in grid box area. SAT is shown as an anomaly compared with the pre-industrial control simulation.



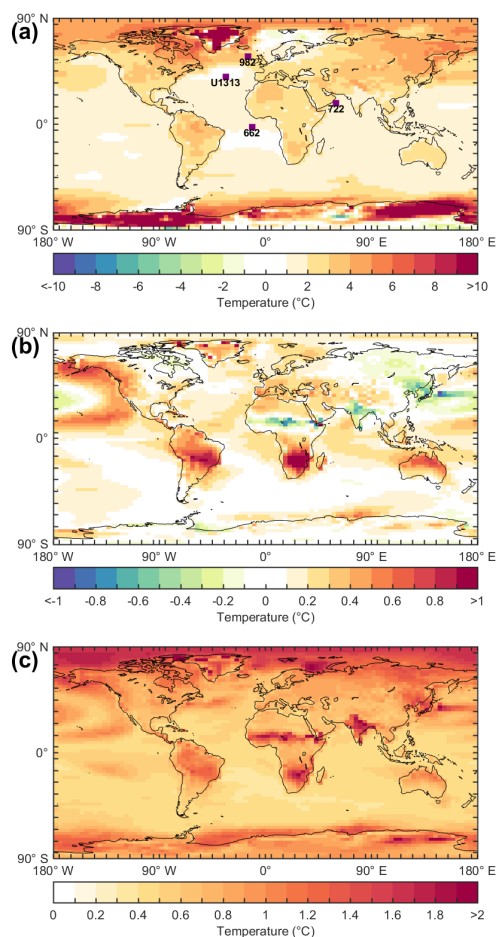

**Figure 9. Emulated mean annual SAT (ºC) for the 400 ppm $CO_2$ scenario, modelled using the *lowice* emulator. SAT is shown as an anomaly compared with the pre-industrial control simulation. (a) Mean annual SAT for modern-day orbital conditions. Also shown are the locations of the four ODP/IODP sites (purple squares): Site 982 (North Atlantic; (Lawrence et al., 2009)), Site U1313 (North Atlantic; (Naafs et al., 2010)), Site 722 (Arabian Sea; (Herbert et al., 2010)) and Site 662 (tropical Atlantic; (Herbert et al., 2010)). (b) Anomaly in mean annual SAT averaged over the period 3300-2800 kyr BP (late Pliocene) compared to that produced under modern-day orbital conditions (Fig. 9a). (c) Standard deviation of mean annual SAT for the period 3300-2800 kyr BP (late Pliocene), also taking into account the emulator posterior variance.**



**Figure 10. Data-model comparison of temperature for the period 3300-2800 kyr BP (late Pliocene). (a) Time series of orbital variations (Laskar et al., 2004), showing eccentricity (black) and precession (radians; blue) on the left axis, and obliquity (degrees; red) on the right axis. (b):(e) Time series of emulated grid box mean annual SAT (ºC; plain lines), modelled every 1 kyr, for three constant CO₂ scenarios; 280 ppm (black), 350 ppm (red) and 400 ppm (blue). Modelled using the *lowice* emulator. Error bands represent the emulated grid box posterior variance (1 standard deviation). Also shown is SST proxy data (ºC; dotted lines) calibrated using the method of Prahl et al. (1988) (maroon), and the method of Muller et al. (1998) (green). SSTs for four ODP/IODP sites are compared: Site 982 (North Atlantic; (Lawrence et al., 2009)), Site U1313 (North Atlantic; (Naafs et al., 2010)), Site 722 (Arabian Sea; (Herbert et al., 2010)) and Site 662 (tropical Atlantic; (Herbert et al., 2010)). SAT is shown as an anomaly compared with the pre-industrial control simulation, SST is shown as an anomaly compared with SST observations for the period 1870-1900 taken from the HadISST dataset (Rayner et al., 2003). Note the different vertical axis scales.**



**Figure 11. The wavelet power spectrum for 3300-2800 kyr BP (late Pliocene). Wavelet analysis was performed on emulated grid box mean annual SAT (ºC), modelled every 1 kyr using the *lowice* emulator, for constant $CO_2$ of 400 ppm (blue line in Fig. 10b to 10e). The data are normalized by the mean variance for the analysed SAT data ($\sigma^2$ = 0.14ºC). Four ODP/IODP sites are compared: (a) Site 982 (North Atlantic; (Lawrence et al., 2009)), (b) Site U1313 (North Atlantic; (Naafs et al., 2010)), (c) Site 722 (Arabian Sea; (Herbert et al., 2010)) and (d) Site 662 (tropical Atlantic; (Herbert et al., 2010)).**





**Figure 12. Data-model comparison of atmospheric CO₂ concentration (ppm) for the period 3300-2800 kyr BP (late Pliocene) for six ODP/IODP sites: Site 982 (North Atlantic), Site U1313 (North Atlantic), Site 722 (Arabian Sea), Site 999 (Caribbean), Site 662 (tropical Atlantic), and Site 1241 (east tropical Pacific). (a) Time series of atmospheric CO₂ concentration from selected proxy data records. Shown is CO₂ estimated from alkenone (squares) for Site 999 by Seki et al. (2010) (light blue), Badger et al. (2013) (dark blue) and for Site 1241 by Seki et al. (2010) (orange), and estimated from δ¹¹B (triangles) for Site 999 by Seki et al. (2010) based on modelled carbonate concentration ([CO₃²⁻]) (grey) and assuming modern total alkalinity (TA; pink), Bartoli et al. (2011) (dark green), Martinez-Boti et al. (2015) (red) and for Site 662 by Martinez-Boti et al. (2015) (purple). For the Seki et al. (2010) δ¹¹B records, error bars are ±25 ppm and the error band is the result of varying the modern TA by ±5%, whilst for Martinez-Boti et al. (2015) the error band represents the 95% confidence interval for a 10,000 member Monte Carlo analysis. (b):(e) Time series of atmospheric**

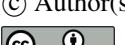



**CO$_2$ concentration estimated from SST proxy data (circles; Herbert et al. (2010) – Sites 662 and 722, Naafs et al. (2010)**
**– Site U1313, Lawrence et al. (2009) – Site 982) calibrated using the method of Prahl et al. (1988) (maroon), and the**
**method of Muller et al. (1998) (light green). CO$_2$ is calculated based on a linear relationship between emulated grid box**
**mean annual SAT (modelled using the *lowice* emulator) and CO$_2$, for three constant CO$_2$ scenarios of 280, 350 and 400**
**ppm. Error bands represent estimated atmospheric CO$_2$ concentration taking into account the emulated grid box**
**posterior variance (1 standard deviation). Where the error appears to be very low, this is generally an artefact of the**
**way that the data has been plotted. The pre-industrial CO$_2$ concentration of 280 ppm (grey dotted line) is included for**
**reference.**

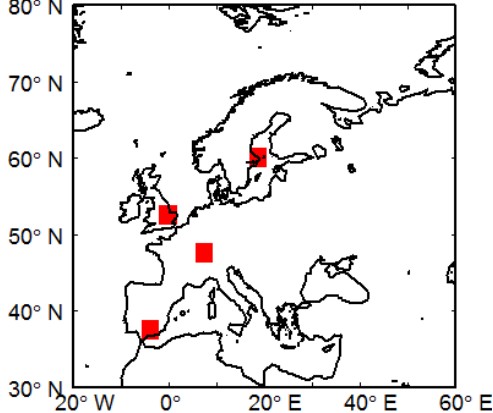

**Figure 13. Map of Europe highlighting the grid boxes that represent the four case study sites. From north to south:**
**Sweden, Central England, Switzerland and Spain.**



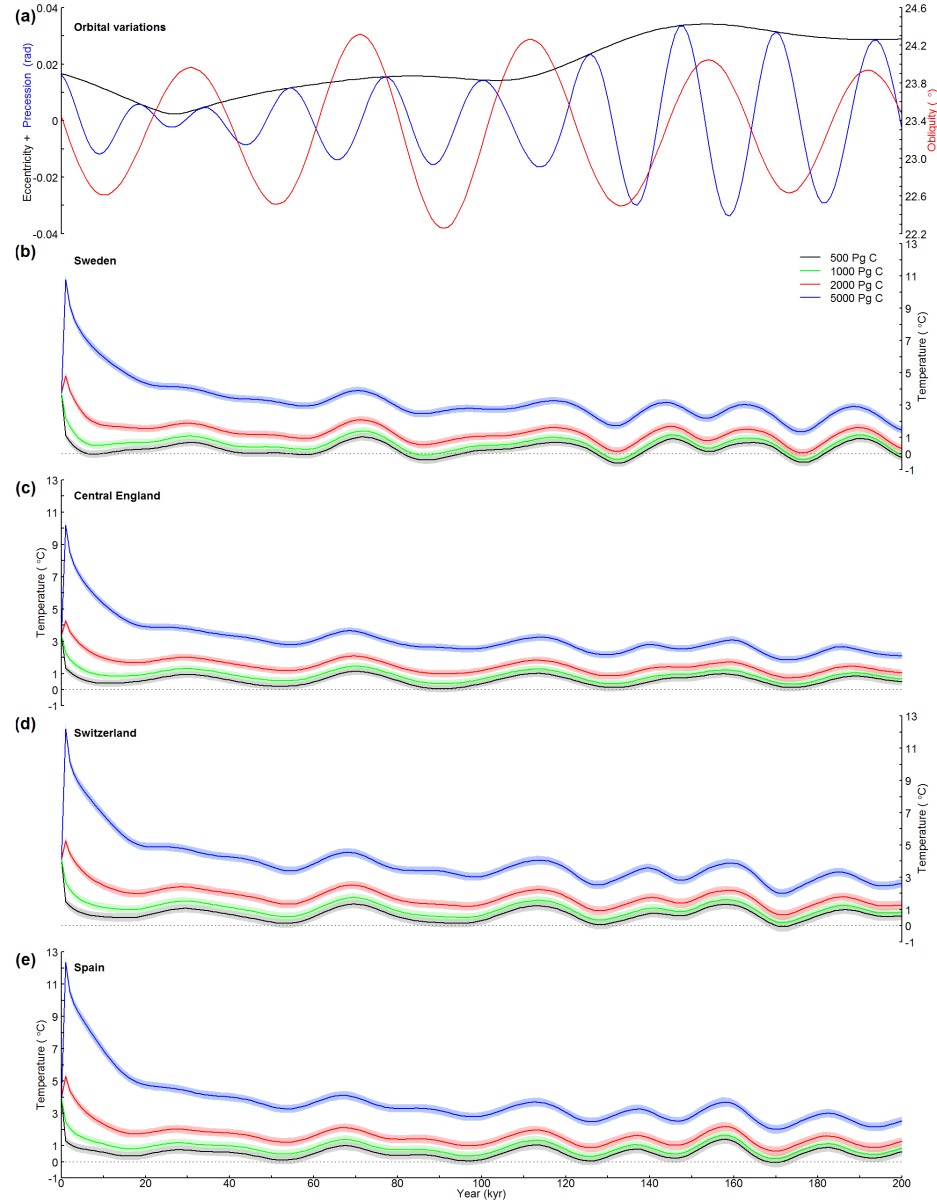

**Figure 14. Emulation of SAT for the next 200 kyr. (a) Time series of orbital variations (Laskar et al., 2004), showing eccentricity (black) and precession (radians; blue) on the left axis, and obliquity (degrees; red) on the right axis. (b): (e) Time series of emulated grid box mean annual SAT (ºC), modelled every 1 kyr, for four CO₂ emissions scenarios; 500 Pg C (black), 1000 Pg C (green), 2000 Pg C (red) and 5000 Pg C (blue). Modelled using the *modice* emulator. Error bands represent the emulated grid box posterior variance (1 standard deviation). Four sites are presented, representing grid boxes in Sweden, Central England, Switzerland and Spain. SAT is shown as an anomaly compared with the pre-industrial control simulation.**





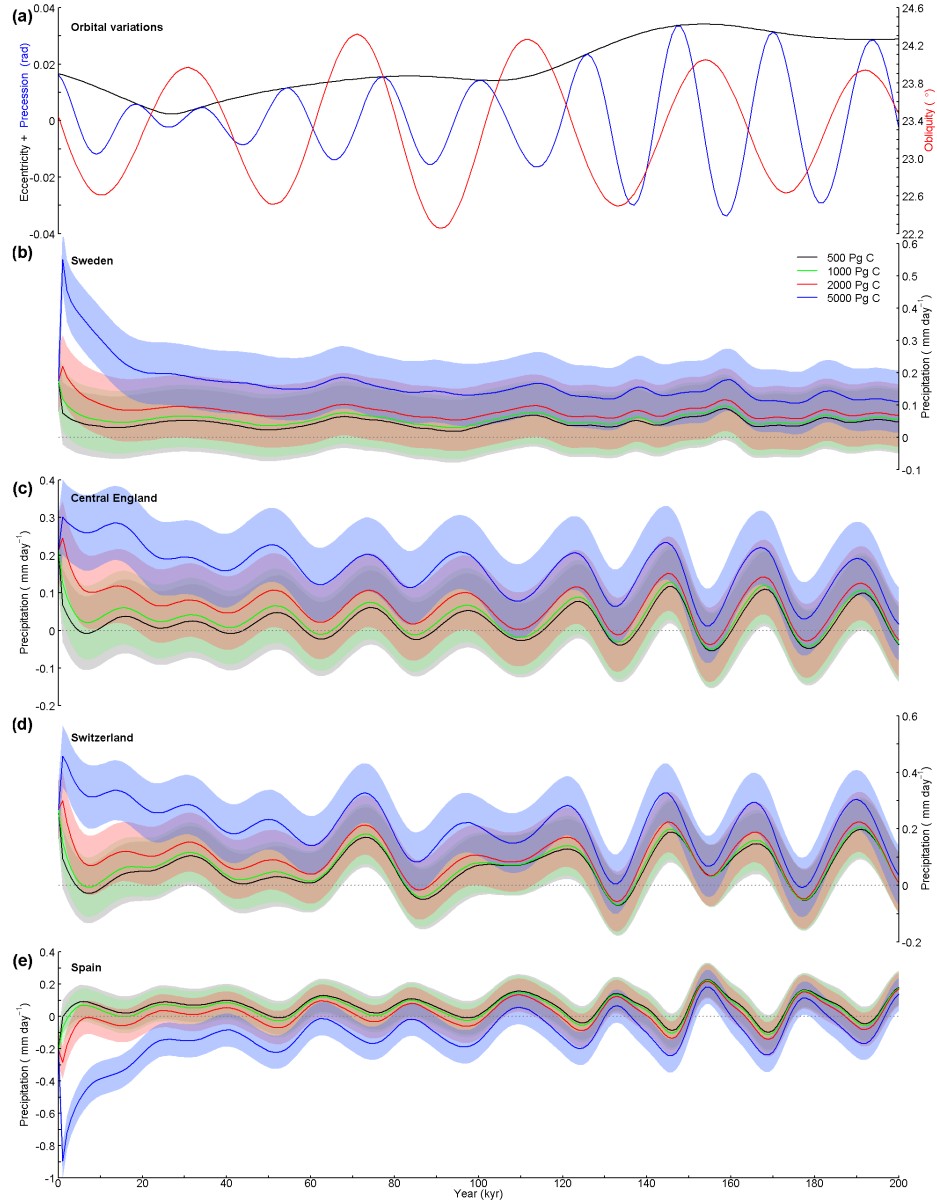

**Figure 15. Emulation of precipitation for the next 200 kyr. (a) Time series of orbital variations (Laskar et al., 2004), showing eccentricity (black) and precession (radians; blue) on the left axis, and obliquity (degrees; red) on the right axis. (b) : (e) Time series of emulated grid box mean annual precipitation (mm day⁻¹), modelled every 1 kyr, for four CO₂ emissions scenarios; 500 Pg C (black), 1000 Pg C (green), 2000 Pg C (red) and 5000 Pg C (blue). Modelled using the *modice* emulator. Error bands represent the emulated grid box posterior variance (1 standard deviation). Four sites are presented, representing grid boxes in Sweden, Central England, Switzerland and Spain. Precipitation is shown as an anomaly compared with the pre-industrial control simulation. Note the different vertical axis scales.**





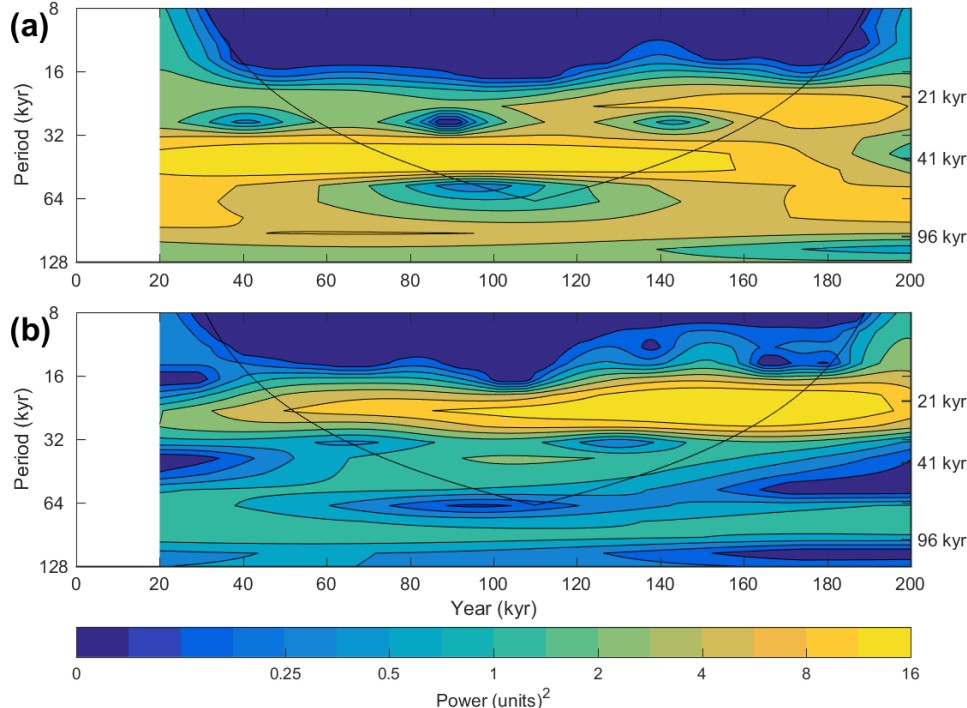

**Figure 16. The wavelet power spectrum for the next 200 kyr for the Central England grid box. Wavelet analysis was performed on data for 20 kyr AP onwards, for: (a) emulated grid box mean annual SAT (ºC; blue line in Fig. 14c), and (b) emulated grid box mean annual precipitation (mm day$^{-1}$; blue line in Fig. 15c). Both variables were modelled every 1 kyr using the *modice* emulator, for the 5000 Pg C emissions scenario. The data are normalized separately by: (a) the mean variance for the analysed SAT data ($\sigma 2 = 0.14$ºC), and (b) the variance for the analysed precipitation data ($\sigma^2 = 0.003$ºC).**