# Peer review of "Emulation of long-term changes in global climate: Application 1 to the late Pliocene and future 2"

_Climate of the Past, 2017_

## Referee Comment (RC1) · A. Ganopolski (Referee) · 6 Jun 2017

The manuscript by Lord et al. presents a new statistical emulator based on a large set of GCM simulations. The authors tested their methods against climate reconstructions for late Pliocene and then applied it to produce a set of climate change projections for the next 200,000 years for different CO2 emission scenarios. Advantage of proposed method that it allows one to obtained high resolution climate scenarios on very long time scales with very low computational cost. The manuscript is well-written and requires only minor revision.

General comments

1. I have no doubts that the authors clearly realize not only advantages but also important limitations of their methods. Some of these limitations are discussed in different parts of the paper. However, I believe it would be useful for potential users of the methods it would be useful to present a more critical discussion of applicability of the methods and its potential limitations.

Firstly, it should be stated very explicitly that the emulator is not applicable for simulations of transient climate cahnge on time scales shorter than several millennia . However, on such long time scales, two major climate forcing – CO2 and ice sheets – strongly interact with each other, that cannot be accounted for in the method presented in the manuscript. On the page 11 the authors wrote that they "are able to simulate global climate development over long periods of time (several million years), provided that atmospheric CO2 level for the period is known, . . . ice sheets do not change outside the range considered... and the topography and land-see mask are unchanged". I believe the authors are too optimistic concerning "several million years" - even a much shorter time interval for which all these conditions are met would be difficult to find in the recent past or in the near future. Clearly, this method is not applicable to Quaternary. For Pliocene, CO2 concentration is not known sufficient accuracy. However, it is very likely that during the late Pliocene CO2 concentration experienced significant fluctuation at different time scales. It is also likely that during Pliocene, the extent of northern hemisphere ice sheets varied beyond the range used in this study (e.g. Willeit et al., 2015). I cannot see how all these problems can be circumvented without use of a comprehensive Earth system model. The less important but still not negligible problem is that according to the PRISM4 reconstruction, land-sea mask and orography during the late Pliocene in some regions (primarily North America and Europe) differed considerably from the modern ones.

The situation is even more problematic for the future. It is not known how good is performance of existing carbon cycle models on such long time scales, but a reasonable agreement between results obtained with different models gives some hope. However, future simulations with the stand-alone carbon cycle models are only valid till the next

glacial inception. For the medium emission scenarios, the next glacial inception is immanent (of course making a brave assumption that humans will not influence climate after the end of fossil fuel era) before or soon after 100,000 AD. Beyond that time, the methodology described in the manuscript is not applicable any more. For the extreme Business-as-usual type scenarios (5000 GtC and more), the situation is even worse. Under such scenarios, most of the Greenland ice sheet will melt completely already within the next 1000 years and most of the Antarctic ice sheet will also melt eventually (e.g. Winkelmann et al., 2016). And, according to recent study by DeConto and Pollard (2016), this "eventually" may occur already within one or two millennia. Such rate of ice sheet melt would strongly affect the ocean circulation and stratification with unknown but long-term consequences. In addition, 70 meter sea level rise resulting from melting of existing ice sheets would strongly affect global land-sea mask and regional climates. In addition, submersion of the large part of northern Europe would also have serious implications for the geological storage of nuclear wastes in this area. As the result, the conditions required for applicability of the proposed method can be violated already after the first few thousand years.

Second, the emulator cannot be applied if the climate system possesses a strong non-linearity. AMOC shutdown is the most natural example. The authors mentioned non-linearity only once and assumed that "any non-linearities in the GCM response being absorbed by stochastic component of the Gaussian process" (p. 6). I am not sure I understand what this means. Please clarify.

By saying all that, I do not challenge the usefulness of the proposed methods. This method, if properly applied, can be used for different types of studies, like analysis of safety of long-term storage of nuclear wastes. However, for potential users the knowledge about limitations and potential caveats is crucial to prevent misuse of this method.

2. The model reveals strong response in annual mean temperature on precessional forcing. Since annual mean precessional component of orbital forcing is zero, I wonder

what causes such response. Is it really global or only regional phenomenon? May be it would be useful to add to the Fig. 4 annual SAT anomalies produced by other forcings: CO2, obliquity and precession (say difference between the maximum and minimum obliquity and difference between the "warm" and the "cold" orbits).

3. I found the attempt to reconstruct Pliocene CO2 from individual temperature records rather strange. These four temperature records are so poorly correlated with each other that it is hard to expect that any global factor (like CO2) can bring them in agreement with modeling results. As the result, all four CO2 "reconstructions" have very little in common. I wonder what one can learn from such exercise. Although I cannot be objective in this respect, but I do believe that using of stacked data (e.g. Willeit et al. (2015), Stap et al. (2016)) rather than individual records, is more appropriate approach to reconstruct past CO2 concentrations.

4. I would strongly suggest to not use expressions like "fossil fuel emission" or "anthropogenic fossil fuel emission". Unfortunately, this jargon is used in some publications related to energy and mitigation. However, I do not believe it is appropriate for climate modeling papers. In any case, burning of fossil fuel is the most important but not the only source of anthropogenic CO2. Land use and cement production also play a role in rising of atmospheric CO2 concentration.

Specific comments

L. 54 Which "system" is meant here?

L. 74 Typo. "precessional"

L. 110 I would change "modern day" to "Quaternary"

L. 128 "input configuration"?

L. 250 change "forcings" to "parameters"

L. 256 What about obliquity?

L. 265. This is not estimate of "remaining reserves". This is just "current estimate" of fossil fuel reserves which has a tendency to increase with time.

L. 277 I do not believe that 20 ppm $CO_2$ change during Holocene (which is primarily transient response to the deglaciation) has something to do with the natural $CO_2$ variability during Anthropocene.

L. 209 Emission cannot be removed

L. 298 $CO_2$ will not return to preindustrial level because glacial cycles will resume before this will happened. But even without glacial cycles, it is unlikely that preindustrial level of 280 ppm is the true equilibrium $CO_2$ concentration in the interglacial world. Even small disbalance between volcanic outgassing and weathering would cause significant $CO_2$ drift on time scale order of 100,000 years.

L. 353 Please specify initial conditions for model runs.

L. 367 Which positive feedback is meant here? I guess this is just an artifact of models with prescribed present day vertical ozone profile.

L. 502 Why "linear nature of the plot increases " confidence? In theory, this plot must not be necessarily linear.

L. 585 What is "SAT index"

L. 751 "Across the four sites. . ." This sentence is not clear

L. 758 What is the meaning of "emulated uncertainty" and how it was defined?

L. 763 What is meant under other "human activities"?

L. 776 "long atmospheric lifetime of fossil fuel emission"?

L. 776 Reference to the original Archer (2005) paper would be much more appropriate

L. 813 -820. The authors try to argue here that the fact that they cannot model ice sheet evolution is not very important for the future 200,000 years climate projections.

This is not true – see my general comments.

L. 899 "High latitude sites concentrations" Sounds like $CO_2$ concentration is different in different sites

Fig. 9. I guess Fig9a shows annual SAT difference due to $CO_2$ increase to 400 ppm. If so "modern annual SST" is misleading. What is shown in 9b is not clear to me.

---

## Short Comment (SC1) · 13 Jun 2017

We would like to thank Andrey Ganopolsky for his thoughtful and constructive review. We will provide a full response once all reviews are in, but we found it useful to clarify a potential source of misunderstanding about the emulator response without waiting further.

The emulator is a meta-model, a "model" of the GCM. In that sense, it expresses a prior judgement about the GCM behaviour before we start running the experiments, and this prior judgement conditions among other things the way we organise our experiment design.

In our case, as in most applications we have seen so far, the most important judgement is that the GCM response is *smooth*, but it does not need to be linear. Another important judgement is that the GCM internal variability is Gaussian. To illustrate this point we consider here an ideal "GCM", with only one parameter, of which the response would be:

$$f(x) = \texttt{atan}(2(x-5)), \tag{1}$$

with a Gaussian internal variability (that is, the deviation from $f(x)$ obtained when performing one run of limited length) of $0.2$.

Suppose ten experiments sampling $x$, as shown on Figure 1. The "true" response $f(x)$ is in blue. Now, we apply the Gaussian process emulation following the methodology used in our article, and produce the black dashed curve, with a zone of 1-$\sigma$ uncertainty represented by the gray zone.

In this case, the *meta-model* (Gaussian process) appears to be a good model of the true function, even though the latter is not linear. True, the prior mean of the Gaussian process is linear (term $h(x)\beta$ in equation (4)), but the *posterior* contains the term $t(x)A^{-1}(y - H\hat{\beta})$ (equation 7) which absorbs deviations from linearity.

We concede that the term "emulator" has been used by other authors to actually describe linear regression, but this is not the case here, as again shown by Figure 1.

It remains that, as in any model, a "meta-model" can be structurally wrong. The GCM response may not be smooth. This is what so-called validation diagnostics are supposed to detect. By inspecting the properties of leave-one-out experiments (Gaussianity of errors, independence with respect to explanatory variables. . . ) you comfort yourself that the meta-model adequately captures the GCM behaviour. It also gives you the possibility to explore potential explanations of diagnostics that may appear problematic, and it is at this stage that the prior hypotheses, such as smoothness, can be called into question.

In summary, the emulator must also be seen as an exploratory tool, in addition to its role in retrospective or prospective time-series modelling.

[Figure]

**Fig. 1.** Gaussian process fitting the function f(x) based on experiments represented by the red symbols.

---

## Referee Comment (RC2) · Anonymous Referee #2 · 21 Jun 2017

The authors of this manuscript are investigating long-term past and future climatic changes under the forcing of orbital parameters and prescribed CO2 scenarios. For this purpose they have developed an interesting emulation technique calibrated on GCM experiments. Overall, the manuscript is quite interesting, well written and represents a useful contribution. I believe that after some modifications following my comments below, it would be suitable for publication in Climate of the Past.

1 – My main concern is about the limitations of the emulation strategy. They are not sufficiently stressed in the manuscript. Indeed, the authors have performed a very good job in developing and implementing the emulator technique, and the manuscript explains in details the methodology. To some extent, this is "the best that can be done" based on GCM tools. But, obviously this is also probably not entirely sufficient... Overall, the fundamental hypothesis is that "climate" responds very smoothly (as explained in the paper) to external forcing. This also makes the even stronger assumption that long-term components of the Earth system, in particular the deep ocean, the carbon cycle and ice-sheets, have no dynamic role. Though this is indeed a fairly usual assumption when studying century-scale changes, this is unlikely to be adequate for 100-kyr to million-year studies. I think the authors should clearly state that their strategy cannot account for : (for instance) deep ocean changes (as experienced during the Quaternary during cold and but also warm periods), CO2 dynamics, ice sheet dynamics. The authors make the hypothesis that it might be suitable for warmer climates (thus the Pliocene and the future) while it is clearly inadequate for the Pleistocene. This might be true, but it is also likely a perspective problem: we know quite well that the Pleistocene climate results from complex interactions between ice-sheets, deep ocean and CO2; with much fewer data, we may (or may not) assume that the Pliocene is simpler...

2 – On Pliocene results. In line with the above comment, the hypothesis of rather small ice-sheet changes in the late Pliocene is not very well founded. The authors mention that their chosen time window does not include the M2 glaciation at 3300 kyrBP (line 614). This is not quite correct since they investigate the 3300-2800 kyrBP time window, which starts precisely with the M2 glaciation, as can be clearly seen on the data of Fig.10. The M2 glaciation is estimated to correspond to a sea-level fall between 40 and 65 m (Miller et al. 2012; Dwyer & Chandler, 2009). The following cold events (KM2 at G20) are not so well characterized, but should correspond to roughly half the size of M2 (20 to 40 m of sea level drop). On the other side, the G17, K1 or KM3 time periods experienced significantly reductions in ice volume with sea level rise estimated to be +25±10 m (Miller et al. 2012). Overall, ice-sheet changes are certainly much larger than assumed in the manuscript, and not bounded by the lowice/modice configurations.

3 – The corresponding calculation of pCO2 (§6.3) probably illustrates the failure of these assumptions. In any case, the four "reconstructions" shown on Fig.12 have little

in common, which certainly deserves some comments. The much higher variability seen in high-latitude data points to "polar" climatic processes not being accounted for by the emulator (like ice-sheets, incorrect sea-ice, . . .). Instead of presenting these curves as possible pCO2 reconstructions (something difficult to buy), I would rather use them to discuss the limitation of the overall strategy : if the model were perfect, the four curves should be identical. . . Most probably, the model-data strategy is furthermore inadequate: For instance, is it reasonable to use annual mean SAT to be compared with alkenone-based SST reconstructions ?

4 – On the future 200 ka results. I also have problems with the rather "conservative" assumption of small ice sheet changes. According to Pollard & DeConto (2016), the disappearance of WAIS (somewhat equivalent to lowice?) correspond to the rather mild RCP4.5 scenario, while an extended RCP8.5 results in more than 20 m of sea level rise for Antarctica alone. These ice-sheet changes might also impact the deep ocean circulation, something difficult to account with the emulator strategy.

5 – Lines 808 + following are discussing the limitations of the overall strategy for the next glacial inception, since there is no ice-sheet model component. I would also add that the carbon cycle is prescribed here, not interactive. In other words, the long-term smooth decrease of CO2 is based on the assumption that nothing unexpected will happen in the Earth carbon cycle, and that the "silicate weathering" mechanism (or hypothesis) is a robust one, something far from being fully understood.

6 – On the experimental design, it could be useful to explain why the ice-sheet size (lowice/modice) has not been included in the emulation procedure.

7 – The simulation of sea ice at high latitudes under high CO2 might be a problem, as explained in the text (lines 575-580). It could be useful to discuss rapidly how HadCM3 compares to other GCMs in terms of sea ice.

8 – Line 871. The comparison of model results with paleodata, or the projection of future impacts, is not so much a question of resolution. 1 - The GCM resolution is often

not sufficient. 2 - Very often, this requires additional modelling (proxy modeling, impact models, . . .)

9 – Fig.2: Simulations over 2000 ppm have been discarded (§3.4.1): the corresponding points should either be removed, or should be plotted with a different color. These plots are not "slices" but "projections".

10 – Fig.10: the comparison to data is poor. I believe just computing a correlation coefficient and/or explained variance ratio could be useful. See above comments on discussing the overall limitations.

References:

Miller KG et al. 2012 High tide of the warm Pliocene: implications of global sea level for Antarctic deglaciation. Geology 40, 407–410. (doi:10.1130/G32869.1) Dwyer, G.S., Chandler, M.A., 2009. Mid-Pliocene sea level and continental ice volume based on coupled benthic Mg/Ca palaeotemperatures and oxygen isotopes. Phil. Trans. R. Soc. A 367, 157–168.
 Pollard et DeConto. Contribution of Antarctica to past and future sea-level rise. Nature (2016) vol. 531 (7596) pp. 591-597

---

## Editor Comment (EC1) · L. Skinner (Editor) · 11 Jul 2017

Dear Natalie Lord and co-authors,

You will have seen that two reviews of your manuscript have now been provided. Both of these are generally supportive of publication and recommend as much. However, it is also clear that both reviewers feel that publication should be made contingent on at least one major revision and some other more detailed adjustments. The main weakness that is perceived by both reviewers is the need for a more explicit description and discussion of the limitations of the 'emulation' approach that has been adopted. This comment bears specifically on the ability of the emulator to 'predict' climate adjustments where significant ice sheet changes and/or irreversible transitions occur. I

think it is a reasonable proposition that the 1,000-1,000,000yr timescale is indeed one in which such changes are not only possible, but also likely to occur. Furthermore, such considerations are likely to be paramount for one of the stated motivations of this approach to long-term prediction: the need to foresee climatic changes, including e.g. glacial erosion and/or lithospheric loading, that may bear on the long-term safety of nuclear waste disposal sites. I do suspect that readers will wish to understand clearly the applicability (and its bounds) of a long-term prediction that does not admit of glacial advance, isostatic adjustment and 'irreversible' changes (non-linear climate adjustments that are not 'smooth'); however, you may have a robust response to this proposition.

I would therefore like to invite you to submit at your earliest convenience a full response to the reviewers' comments, as well as suitably revised manuscript that takes into account the reviewers' comments, including in particular the main comment regarding a clear description and discussion of the 'limitations' of the emulation approach that is adopted.

Yours sincerely, Luke Skinner

---

## Author Comment (AC1)

We thank the reviewers and editor for their useful comments, which have improved the manuscript. This document includes a response to all the Reviewer and Editor comments. This is then followed by a revised version of the manuscript in which all our proposed changes are clearly highlighted (including line numbers which are referenced by this document).

**Reviewer I – Andrey Ganopolski**

*General comments*

*1. ...I believe it would be useful for potential users of the methods it would be useful to present a more critical discussion of applicability of the methods and its potential limitations.*

**We agree, and have added a completely new section (Section 7; Conclusions are now Section 8) to clearly describe and discuss these limitations. A number of key limitations are also now included in the Abstract (lines 41-44).**

*Firstly, it should be stated very explicitly that the emulator is not applicable for simulations of transient climate change on time scales shorter than several millennia. However, on such long time scales, two major climate forcing – CO2 and ice sheets – strongly interact with each other, that cannot be accounted for in the method presented in the manuscript. On the page 11 the authors wrote that they "are able to simulate global climate development over long periods of time (several million years), provided that atmospheric CO2 level for the period is known, : : : ice sheets do not change outside the range considered... and the topography and land-see mask are unchanged". I believe the authors are too optimistic concerning "several million years" - even a much shorter time interval for which all these conditions are met would be difficult to find in the recent past or in the near future. Clearly, this method is not applicable to Quaternary.*

**We add a comment that the experiment design adopted here is not appropriate for simulations shorter than a few millennia, where complex models and transient simulations are most appropriate (lines 1006-1012). At the other extreme, "Several million years" has been changed to "several hundred thousand years or longer" (line 642) (see point below about the applicability of this emulator up to the next glacial inception, and new Section 7).**

*For Pliocene, CO2 concentration is not known sufficient accuracy. However, it is very likely that during the late Pliocene CO2 concentration experienced significant fluctuation at different time scales. It is also likely that during Pliocene, the extent of northern hemisphere ice sheets varied beyond the range used in this study (e.g. Willeit et al., 2015). I cannot see how all these problems can be circumvented without use of a comprehensive Earth system model. The less important but still not negligible problem is that according to the PRISM4 reconstruction, land-sea mask and orography during the late Pliocene in some regions (primarily North America and Europe) differed considerably from the modern ones.*

**Yes, we agree, and have made it clearer in the Pliocene section that (a) our approach is only appropriate for periods of the Pliocene with equivalent or less ice than modern, and (b) that we do not include palaeogeographic changes in our Pliocene simulations.**

*The situation is even more problematic for the future. It is not known how good is performance of existing carbon cycle models on such long time scales, but a reasonable agreement between results obtained with different models gives some hope. However, future simulations with the stand-alone carbon cycle models are only valid till the next glacial inception. For the medium emission scenarios, the next glacial inception is immanent (of course making a brave assumption that humans will not influence climate after the end of fossil fuel era) before or soon after 100,000 AD. Beyond that time, the methodology described in the manuscript is not applicable any more. For the extreme Business-as-usual type scenarios (5000 GtC and more), the situation is even worse. Under such scenarios, most of the Greenland ice sheet will melt completely already within the next 1000 years and most of the Antarctic ice sheet will also melt eventually (e.g. Winkelmann et al., 2016). And, according to recent study by DeConto and Pollard (2016), this "eventually" may occur already within one or two millennia. Such rate of ice sheet melt would strongly affect the ocean circulation and stratification with unknown but long-term consequences. In addition, 70 meter sea level rise resulting from melting of existing ice sheets would strongly affect global land-sea mask and regional climates. In addition, submersion of the large part of northern Europe would also have serious implications for the geological storage of nuclear wastes in this area. As the result, the conditions required for applicability of the proposed method can be violated already after the first few thousand years.*

**Thank you for raising these points. Whilst, as you say, we have mentioned some of the limitations at different points throughout the manuscript, it is better to have a section dedicated to describing the assumptions that the emulator is based on, its limitations and the conditions under which it may be applied. This includes the point made here about the applicability of the current emulator approach only up to the next glacial inception. We have added a new section (7; Conclusions are now Section 8) to clearly describe these limitations. We also explain that the emulator could be expanded to include glacial**

**states and therefore be applied to longer-term future (and the Quaternary), if the CO2 and ice volume were known (e.g. from a transient EMIC or conceptual model simulation).**

*Second, the emulator cannot be applied if the climate system possesses a strong nonlinearity. AMOC shutdown is the most natural example. The authors mentioned nonlinearity only once and assumed that "any non-linearities in the GCM response being absorbed by stochastic component of the Gaussian process" (p. 6). I am not sure I understand what this means. Please clarify.*

**Please see the response posted by Michel Crucifix in the Discussion.**

*2. The model reveals strong response in annual mean temperature on precessional forcing. Since annual mean precessional component of orbital forcing is zero, I wonder what causes such response. Is it really global or only regional phenomenon? May be it would be useful to add to the Fig. 4 annual SAT anomalies produced by other forcings: CO2, obliquity and precession (say difference between the maximum and minimum obliquity and difference between the "warm" and the "cold" orbits).*

**Figure 4 has been extended to show a larger range of forcings, as suggested. In addition to the SAT change due to reduced ice, plots are also now included showing the SAT change due to a doubling of $CO_2$, the difference between maximum and minimum obliquity and the difference between "warm" and "cold" orbital conditions. The following text has also been added to Section 3.5.1:**

**"Also shown in Fig. 4, for comparison, are mean annual SAT anomalies produced by the other forcings, including a doubling of $CO_2$, the difference between maximum and minimum obliquity and the difference between "warm" orbital conditions and "cold" orbital conditions. The warming caused by increased $CO_2$ is more widespread (Fig. 4b), with the largest warming occurring at high latitudes and for land regions, in agreement with typical future-climate simulations (IPCC, 2013, p. 1059). The temperature change due to obliquity and "warm" versus "cold" orbital conditions is less than that for either reduced ice (compared to pre-industrial) or increased $CO_2$. Changes in obliquity have the largest impact on temperatures in high latitude regions, since the exposure of these regions to the sun's radiation is most affected by changes in obliquity. Smaller temperature anomalies are observed over northern Africa and India and, since an increase in obliquity is indeed known to boost monsoon dynamics (e.g. Araya-Melo et al., 2015; Bosmans et al., 2015), changes in soil latent heat exchanges are therefore expected to contribute negatively to the temperature response. The comparison of "warm" versus "cold" orbital conditions, which highlights (annual mean) temperature changes primarily caused by precession, generally shows a warming trend, with the largest temperature changes occurring in monsoonal regions. Lower temperatures are observed in the Northern Hemisphere over northern Africa, India and, East Asia, whilst warmer temperatures occur in the Southern Hemisphere over South America, southern Africa and Australia. Figure 4 demonstrates that the temperature forcing caused by $CO_2$ affects mean annual temperatures on a global scale, whilst the forcing due to ice sheet and orbital changes affects mean annual temperatures in specific regions, having a limited impact on global mean temperatures. This is supported by the relatively high global mean SAT anomaly for the $2xCO_2$ scenario of $4.2°C$, compared with the lower SAT anomalies that result from the obliquity and precession forcing of $0.4°C$ each (see caption of Figure 4)."**

*3. I found the attempt to reconstruct Pliocene CO2 from individual temperature records rather strange. These four temperature records are so poorly correlated with each other that it is hard to expect that any global factor (like CO2) can bring them in agreement with modelling results. As the result, all four CO2 "reconstructions" have very little in common. I wonder what one can learn from such exercise. Although I cannot be objective in this respect, but I do believe that using of stacked data (e.g. Willeit et al. (2015), Stap et al. (2016)) rather than individual records, is more appropriate approach to reconstruct past CO2 concentrations.*

**Whilst we agree that the use of stacked benthic oxygen isotope data is appropriate in many circumstances, we do not believe that it would be suitable in this instance. This is because the GCM experiments that the emulator is calibrated on were relatively short (500 years), and hence benthic ocean temperatures would not yet have spun-up fully and reached steady state, particularly in the high $CO_2$ experiments. Therefore, it would not be appropriate to compare deep ocean temperatures from the proxy data to the modelled temperatures. We have added some text to the paper to explain this (lines 766-770).**

**We also believe that there are benefits to using individual temperature records. For example, if the model and emulator are correct, then the analysis shows that the temperature records are not consistent with each other, which may not be obvious by just comparing the records visually. We have added the following paragraph to Section 5.3 to describe the possible sources of the inconsistencies between the reconstructions at different sites, and what we think are most likely to be the cause:**

"There is substantial variation between our $CO_2$ estimates at different sites, and this may be attributed to a number of causes. It could be that there are errors in the GCM model used, in particular in its representation of the response of climate to $CO_2$ and/or orbital forcing. There could be inaccuracies associated with the SST data at one or more locations as, if the model was assumed to be correct, the estimated $CO_2$ should be similar across the four locations. The fact that they are not may indicate that the temperature records are not consistent with each other, which may not have been obvious by just comparing the records visually. This is one of the potential advantages to using individual temperature records rather than stacked records. It may also be that there is an issue with the dating of some of the proxy records; the data may be correct but there may be uncertainties/inaccuracies in the age models. Alternatively, the emulator may be wrong; for example, there may be non-linearities in the climate response simulated by the GCM that it is not capturing. Finally, there may be errors related to the modelled representation of the ice sheets, which are fixed at a constant configuration. In reality, of the possible sources of error that have been identified, the variations are less likely to be the result of errors in the emulator's estimates of the GCM output because validation diagnostics did not seem to suggest systematic failures. They are also less likely to be due to unrepresented changes in climate due to the ice sheets. Whilst some of the variation at the high latitude sites (982 and U1313) may be attributed to some regional climate processes not fully accounted for, e.g. involving the ice sheets and sea ice, two of the sites (722 and 662) are in tropical regions. Thus, SSTs at these sites would not be expected to be affected by changes in the ice sheets, and yet they show significantly different variations. Therefore, the inconsistencies are likely to be due to a combination of errors in the GCM model and inaccuracies in the SST data."

*4. I would strongly suggest to not use expressions like "fossil fuel emission" or "anthropogenic fossil fuel emission". Unfortunately, this jargon is used in some publications related to energy and mitigation. However, I do not believe it is appropriate for climate modelling papers. In any case, burning of fossil fuel is the most important but not the only source of anthropogenic CO2. Land use and cement production also play a role in rising of atmospheric CO2 concentration.*

**This is a good point. These instances have been changed to "anthropogenic CO2 emissions" or similar throughout.**

*L. 54 Which "system" is meant here?*

**Inserted "climate" (line 60).**

*L. 74 Typo. "precessional"*

**Done.**

*L. 110 I would change "modern day" to "Quaternary"*

**Done.**

*L. 128 "input configuration"?*

**Inserted "(i.e. any set of orbital and $CO_2$ conditions)" (line 136).**

*L. 250 change "forcings" to "parameters"*

**Done.**

*L. 256 What about obliquity?*

**We had missed it out of the sentence, so thank you for pointing it out. It has now been added.**

*L. 265. This is not estimate of "remaining reserves". This is just "current estimate" of fossil fuel reserves which has a tendency to increase with time.*

**Removed "remaining".**

*L. 277 I do not believe that 20 ppm CO2 change during Holocene (which is primarily transient response to the deglaciation) has something to do with the natural CO2 variability during Anthropocene.*

**The sentence has been reworded to make it clear that we present variations during the Holocene as an example of natural variations, rather than the change that we expect to occur in the future (lines 289-291).**

*L. 209 Emission cannot be removed*

**Replaced with "taken up".**

*L. 298 CO2 will not return to preindustrial level because glacial cycles will resume before this will happen. But even without glacial cycles, it is unlikely that preindustrial level of 280 ppm is the true equilibrium CO2 concentration in the interglacial world.*

*Even small disbalance between volcanic outgassing and weathering would cause significant CO2 drift on time scale order of 100,000 years.*

**This sentence has been modified to make these assumptions clear (lines 310-314).**

*L. 353 Please specify initial conditions for model runs.*

**Inserted "All experiments were initiated from a pre-industrial spin-up experiment, with an atmospheric $CO_2$ concentration of 280 ppmv, and pre-industrial ice sheet extents and orbital conditions." (lines 370-371).**

*L. 367 Which positive feedback is meant here? I guess this is just an artefact of models with prescribed present day vertical ozone profile.*

**This sentence has been reworded and extended (lines 384-389):**

**"This is the result of a runaway positive feedback in the GCM caused, at least in part, by the vertical distribution of ozone in the model being prescribed for modern-day climate conditions. Consequently, the ozone distribution is not able to respond to changes in climate, meaning that when increased mean global temperatures result in an increase in altitude of the tropopause and hence an extension of the troposphere, relatively high concentrations of ozone, which were previously located in the stratosphere, enter the troposphere, resulting in runaway warming."**

*L. 502 Why "linear nature of the plot increases" confidence? In theory, this plot must not be necessarily linear.*

**This sentence has been removed.**

*L. 585 What is "SAT index"*

**It is the globally averaged mean annual SAT for each experiment, but it has not been adjusted for grid box area, therefore we refer to it as a "SAT index". The caption for Figure 8 has been amended to clarify this.**

*L. 751 "Across the four sites…" This sentence is not clear*

**Sentence has been reworded (lines 844-846).**

*L. 758 What is the meaning of "emulated uncertainty" and how it was defined?*

**Inserted "(defined as 1 standard deviation of the emulated grid box posterior variance)" (line 712 line 844).**

*L. 763 What is meant under other "human activities"?*

**Sentence has been reworded to include combustion of fossil fuels, land-use change and cement production (lines 856-858.**

*L. 776 "long atmospheric lifetime of fossil fuel emission"?*

**Sentence has been reworded to "$CO_2$ emissions" (lines 871-873).**

*L. 776 Reference to the original Archer (2005) paper would be much more appropriate*

**This reference has been added (line 871).**

*L. 813 -820. The authors try to argue here that the fact that they cannot model ice sheet evolution is not very important for the future 200,000 years climate projections. This is not true – see my general comments.*

**The following sentence has been added to clarify that on these timescales the inability to model ice sheet evolution may be an issue (lines 916-918):**

**"As will be discussed in Sect. 7, however, the emulator was not designed and calibrated to predict changes in ice sheets. This is a limitation that should be addressed when modelling future climate on timescales of tens of thousands of years or more (depending on the $CO_2$ scenario(s) being modelled)."**

*L. 899 "High latitude sites concentrations" Sounds like CO2 concentration is different in different sites*

**This sentence has been reworded (lines 1088-1090):**

**"Our $CO_2$ concentrations derived from tropical ODP/IODP sites show relatively similar concentrations to $CO_2$ proxy records for the same period, although the concentrations derived from higher latitude sites are generally significantly higher than the proxy data."**

*Fig. 9. I guess Fig9a shows annual SAT difference due to CO2 increase to 400 ppm. If so "modern annual SST" is misleading. What is shown in 9b is not clear to me.*

**Yes, it is correct that Fig. 9a shows the annual SAT anomaly due to CO2 being increased to 400 ppm. The caption states that this is "mean annual SAT for modern-day orbital conditions", not "modern annual SST", so we think that this is clear. We agree that Fig. 9b was not really adding anything and have therefore removed it, and amended the main text accordingly.**

**Reviewer II**

*1 – My main concern is about the limitations of the emulation strategy. They are not sufficiently stressed in the manuscript. Indeed, the authors have performed a very good job in developing and implementing the emulator technique, and the manuscript explains in details the methodology. To some extent, this is "the best that can be done" based on GCM tools. But, obviously this is also probably not entirely sufficient… Over all, the fundamental hypothesis is that "climate" responds very smoothly (as explained in the paper) to external forcing. This also makes the even stronger assumption that long-term components of the Earth system, in particular the deep ocean, the carbon cycle and ice-sheets, have no dynamic role. Though this is indeed a fairly usual assumption when studying century-scale changes, this is unlikely to be adequate for 100-kyr to million-year studies. I think the authors should clearly state that their strategy cannot account for : (for instance) deep ocean changes (as experienced during the Quaternary during cold and but also warm periods), CO2 dynamics, ice sheet dynamics. The authors make the hypothesis that it might be suitable for warmer climates (thus the Pliocene and the future) while it is clearly inadequate for the Pleistocene. This might be true, but it is also likely a perspective problem: we know quite well that the Pleistocene climate results from complex interactions between ice-sheets, deep ocean and CO2; with much fewer data, we may (or may not) assume that the Pliocene is simpler…*

**Thank you for these helpful suggestions. Please see the response to comment (1) of Reviewer I (André Ganopolski). In particular, this new section includes a discussion of the fact that we do not carry out truly transient simulations, but a series of snapshots, and as such our methodology is inappropriate for examining deep ocean trends, and becomes compromised if deep ocean transient changes are important for controlling surface climate evolution.**

*2 – On Pliocene results. In line with the above comment, the hypothesis of rather small ice-sheet changes in the late Pliocene is not very well founded. The authors mention that their chosen time window does not include the M2 glaciation at 3300 kyr BP (line 614). This is not quite correct since they investigate the 3300-2800 kyr BP time window, which starts precisely with the M2 glaciation, as can be clearly seen on the data of Fig.10. The M2 glaciation is estimated to correspond to a sea-level fall between 40 and 65 m (Miller et al. 2012; Dwyer & Chandler, 2009). The following cold events (KM2 at G20) are not so well characterized, but should correspond to roughly half the size of M2 (20 to 40 m of sea level drop). On the other side, the G17, K1 or KM3 time periods experienced significantly reductions in ice volume with sea level rise estimated to be +25±10 m (Miller et al. 2012). Overall, ice-sheet changes are certainly much larger than assumed in the manuscript, and not bounded by the lowice/modice configurations.*

**The following text has been added (lines 665-669). Please also see the response to comment (1) of Reviewer I.**

**"represents the warm phase of climate (interglacial conditions), and does not include major glaciations (though the M2 cooling event may persist to the very start of the simulation at 3300 kyr BP, and the simulated period does include periods of likely glaciation, such as KM2 (~3100 kyr BP) and G20 (~3000 kyr BP)). The emulator would not be appropriate to periods of extensive glaciation and may not be well-matched to the periods of lesser glaciation included within the simulated interval."**

*3 – The corresponding calculation of pCO2 (§6.3) probably illustrates the failure of these assumptions. In any case, the four "reconstructions" shown on Fig.12 have little in common, which certainly deserves some comments. The much higher variability seen in high-latitude data points to "polar" climatic processes not being accounted for by the emulator (like ice-sheets, incorrect sea-ice, ...). Instead of presenting these curves as possible pCO2 reconstructions (something difficult to buy), I would rather use them to discuss the limitation of the overall strategy: if the model were perfect, the four curves should be identical... Most probably, the model-data strategy is furthermore inadequate: For instance, is it reasonable to use annual mean SAT to be compared with alkenone-based SST reconstructions?*

**Please see the response to comment (3) of Reviewer I, and new paragraph at the end of Section 5.3.**

*4 – On the future 200 ka results. I also have problems with the rather "conservative" assumption of small ice sheet changes. According to Pollard & DeConto (2016), the disappearance of WAIS (somewhat equivalent to lowice?) correspond to the rather mild RCP4.5 scenario, while an extended RCP8.5 results in more than 20 m of sea level rise for Antarctica alone. These ice-sheet changes might also impact the deep ocean circulation, something difficult to account with the emulator strategy.*

**These limitations have been discussed in a new section (7) describing the limitations of the methodology.**

*5 – Lines 808 + following are discussing the limitations of the overall strategy for the next glacial inception, since there is no ice-sheet model component. I would also add that the carbon cycle is prescribed here, not interactive. In other words, the long-term smooth decrease of CO2 is based on the assumption that nothing unexpected will happen in the Earth carbon cycle, and that the "silicate weathering" mechanism (or hypothesis) is a robust one, something far from being fully understood.*

**The following text has been added to this paragraph to highlight these assumptions (lines 919-924):**

**"Another caveat is that the carbon cycle in the emulator is also essentially prescribed, and thus not interactive. This means that the atmospheric $CO_2$ trajectory follows a smooth decline, as was projected using an impulse response function based on experiments using the *c*GENIE model (Lord et al., 2016), with long-term future climate being modelled as a series of snapshot simulations with the emulator. This smooth decline in $CO_2$ assumes that no non-linear or unexpected behaviour will be demonstrated by the long-term carbon cycle, and that the silicate weathering mechanism, which is associated with a substantial degree of uncertainty, is correct."**

*6 – On the experimental design, it could be useful to explain why the ice-sheet size (lowice/modice) has not been included in the emulation procedure.*

**The following text was added to the "Calibration and evaluation of the emulator" section (lines 563-566):**

**"This approach was adopted, rather than including the ice sheet extent as an active input parameter to the emulator, because only two ice sheet configurations have been simulated, which are not sufficient for an interpolation. One of the main benefits of including ice sheet extent as an active input parameter would be to emulate changing ice sheets over time, but this was beyond the scope of this study."**

*7 – The simulation of sea ice at high latitudes under high CO2 might be a problem, as explained in the text (lines 575-580). It could be useful to discuss rapidly how HadCM3 compares to other GCMs in terms of sea ice.*

**The section highlighted explains that the PCA, and therefore the emulator, may not be fully capturing high latitude variations, meaning that in the leave-one-out analysis some of the high $CO_2$ simulations include larger errors in these regions compared to the equivalent GCM simulation. It is true that there may also be underlying errors in the AOGCM representation of sea ice. These are discussed in Valdes et al (2017).**

*8 – Line 871. The comparison of model results with paleodata, or the projection of future impacts, is not so much a question of resolution. 1 - The GCM resolution is often not sufficient. 2 - Very often, this requires additional modelling (proxy modelling, impact models, ...)*

**The following sentence has been added to state this (lines 1060-1062):**

**"However, further downscaling of the data may also be necessary or beneficial, via further modelling such as proxy modelling, impact models or regional climate models, or via statistical downscaling techniques."**

*9 – Fig.2: Simulations over 2000 ppm have been discarded (§3.4.1): the corresponding points should either be removed, or should be plotted with a different colour. These plots are not "slices" but "projections".*

**Fig. 2a has been modified as suggested (colour changed). Replaced with "projections".**

*10 – Fig.10: the comparison to data is poor. I believe just computing a correlation coefficient and/or explained variance ratio could be useful. See above comments on discussing the overall limitations.*

**Correlation coefficients have been computed and some text to describe the results has been added to Section 5.1 (lines 694-702).**

**In addition, we have made a small number of minor changes:**

- **The affiliation of Charlotte O'Brien has been corrected**
- **The $CO_2$ reconstructions have been redone using a wider range of constant $CO_2$ scenarios (260, 300, 400, 500, 600, 700, and 800 ppmv) for the linear regression. Figure 12 has been updated with the new data.**
- **Figure 7 – $CO_2$ concentration has been added to the upper y axis**
- **Minor clarifications and rewordings throughout to improve clarity**
- **Ka/Ma has been changed to Myr/kyr where appropriate**

**Emulation of long-term changes in global climate: Application to the late Pliocene and future**

Natalie S. Lord[1,2], Michel Crucifix[3,4], Dan J. Lunt[1,2], Mike C. Thorne[5], Nabila Bounceur[3], Harry Dowsett[6], Charlotte L. O'Brien[6,7] and Andy Ridgwell[1,2,8]

[1]School of Geographical Sciences, University of Bristol, Bristol, BS8 1SS, UK.
[2]Cabot Institute, University of Bristol, Bristol, UK.
[3]Université catholique de Louvain, Georges Lemaître Centre for Earth and Climate Research, Earth and Life Institute, 1348 Louvain-la-Neuve, Belgium.
[4]Belgian National Fund of Scientific Research, Brussels, Belgium.
[5]Mike Thorne and Associates Limited, Quarry Cottage, Hamsterley, Bishop Auckland, Co. Durham, DL13 3NJ, UK.
[6]Eastern Geology and Paleoclimate Science Center, U. S. Geological Survey, Reston, VA 20192, USA.
[7]Department of Geology and Geophysics, Yale University, New Haven, CT 06511, USA.
[8]Department of Earth Sciences, University of California, Riverside, CA 92521, USA.

*Correspondence to:* Natalie S. Lord (Natalie.Lord@bristol.ac.uk)

**Abstract**

Multi-millennial transient simulations of climate changes have a range of important applications, such as for investigating key geologic events and transitions for which high resolution palaeoenvironmental proxy data are available, or for projecting the long-term impacts of future climate evolution on the performance of geological repositories for the disposal of radioactive wastes. However, due to the high computational requirements of current fully coupled General Circulation Models (GCMs), long-term simulations can generally only be performed with less complex models and/or at lower spatial resolution. In this study, we present novel long-term "continuous" projections of climate evolution based on the output from GCMs, via the use of a statistical emulator. The emulator is calibrated using ensembles of GCM simulations which have varying orbital configurations and atmospheric $CO_2$ concentrations and enables a variety of investigations of long-term climate change to be conducted which would not be possible with other modelling techniques at the same temporal and spatial scales. To illustrate the potential applications, we apply the emulator to the late Pliocene (by modelling surface air temperature (SAT)), comparing its results with palaeo-proxy data for a number of global sites, and to the next 200 thousand years (kyr) (by modelling SAT and precipitation). A range of $CO_2$ scenarios are prescribed for each period. During the late Pliocene, we find that emulated SAT varies on an approximately precessional timescale, with evidence of increased obliquity response at times. A comparison of atmospheric $CO_2$ concentration for this period, estimated using the proxy sea surface temperature (SST) data from different sites and emulator results , finds that relatively similar $CO_2$ concentrations are estimated based on sites  at lower latitudes, whereas higher latitude sites show larger discrepancies. In our second illustrative application, spanning the next 200 kyr into the future, we find that SAT oscillations appear to be primarily influenced by obliquity for the first ~120 kyr, whilst eccentricity is relatively low, after which precession plays a more dominant role. Conversely, variations in precipitation over the entire period demonstrate a strong precessional signal. Overall, we find that the emulator provides a useful and powerful tool for rapidly simulating the long-term evolution of climate, both past and future, due to its relatively high spatial resolution and relatively low computational cost. However, there are uncertainties associated with the approach used, including the inability of the emulator to capture deviations from a quasi-stationary response to the forcing, such as  transient adjustments of the deep ocean temperature and circulation, in addition to its limited range of fixed ice sheet configurations and its requirement for prescribed atmospheric $CO_2$ concentrations.

[revised manuscript text omitted]
 $\boldsymbol{y}$ is Gaussian, characterised by $\boldsymbol{y} \sim N(\boldsymbol{H\beta}, \sigma^2 \boldsymbol{A})$, with

$\boldsymbol{A}_{ij} = c(\boldsymbol{x}_i, \boldsymbol{x}_j)$.

Following the specification of the prior model above, a Bayesian approach is now used to update the prior distribution. The posterior estimate of the GCM output is described by:

$$m^*(\boldsymbol{x}) = \boldsymbol{h}(\boldsymbol{x})^{\mathrm{T}}\widehat{\boldsymbol{\beta}} + t(\boldsymbol{x})\boldsymbol{A}^{-1}(\boldsymbol{y} - \boldsymbol{H}\widehat{\boldsymbol{\beta}}), \tag{7}$$

$$V^*(\boldsymbol{x}, \boldsymbol{x}') = \sigma^2[c(\boldsymbol{x}, \boldsymbol{x}') - t(\boldsymbol{x})^T \boldsymbol{A}^{-1} t(\boldsymbol{x}') + \boldsymbol{P}(\boldsymbol{x})(\boldsymbol{H}^T \boldsymbol{A}^{-1} \boldsymbol{H})^{-1} \boldsymbol{P}(\boldsymbol{x}')^T], \tag{8}$$

where

$$\sigma^2 = (n - q - 2)^{-1}(\boldsymbol{y} - \boldsymbol{H}\widehat{\boldsymbol{\beta}})^T \boldsymbol{A}^{-1}(\boldsymbol{y} - \boldsymbol{H}\widehat{\boldsymbol{\beta}}), \tag{9}$$

$$\widehat{\boldsymbol{\beta}} = (\boldsymbol{H}^T \boldsymbol{A}^{-1} \boldsymbol{H})^{-1} \boldsymbol{H}^T \boldsymbol{A}^{-1} \boldsymbol{y}, \tag{10}$$

and $t(\boldsymbol{x})_i = c(\boldsymbol{x}, \boldsymbol{x}_i)$ and $\boldsymbol{P}(\boldsymbol{x}) = h(\boldsymbol{x})^T - t(\boldsymbol{x})^T \boldsymbol{A}^{-1} \boldsymbol{
[revised manuscript text omitted]
 result in periodic fluctuations in $CO_2$. For example, during the Holocene (11 kyr BP to

~1750 CE)  atmospheric $CO_2$ variing between 260 and 280 ppmv
~~kyr BP to ~1750 AD)~~ (Monnin et al., 2004). A value of 250 ppmv is therefore deemed to be appropriate to
account for these natural variations in an unglaciated world, in addition to possible uncertainties in the model
and hence is assumed as the value of the lower $CO_2$ limit in the ensemble.

The orbital and $CO_2$ parameter ranges that have been selected are also applicable to unglaciated periods during
the  late Pliocene, when atmospheric $CO_2$ was estimated to be higher than pre-industrial values (Martinez-
Boti et al., 2015; Raymo et al., 1996). In this study, we do not consider or attempt to simulate past or future
glacial episodes, which may be accompanied by larger continental ice sheets (see Sect. 7 for more discussion),
although the conditions required to initiate the next glaciation, and extending the ensemble of GCM simulations
to represent glacial states, are being investigated in a forthcoming study. The underlying assumption of
our ensemble is that it is suitable for simulating periods for which the $CO_2$ concentration is high enough to
prevent entry into a glacial state.

[revised manuscript text omitted]

For each ensemble, 3000 sample sets were created, with each set consisting of an $n$ by $p$ matrix, $X$, containing the four sampled input parameter values for each of the 40 experiments, and then the optimal sample set was selected as the final ensemble based on a number of criteria. Following Joseph and Hung (2008), we seek, in addition to the maxi-min criterion, to maximise $det(X^TX)$. Here, we will term this determinant the "orthogonality", because the columns of the design matrix will  approach orthogonality as this determinant is maximised (assuming that input factors are normalised). However, a limitation of the method of sampling the parameters $e\sin\varpi$ and $e\cos\varpi$, rather than eccentricity and longitude of perihelion directly, is that due to the nature of the $e\sin\varpi$ and $e\cos\varpi$ parameter space, the sampling process favours higher values of eccentricity over lower ones. This is not an issue for the longitude of perihelion, because when eccentricity is low the value of this parameter has little effect on insolation. However, the value of obliquity selected for a given eccentricity value could have a significant impact on climate, meaning that it is desirable to have a relatively large range of obliquity values for low (<0.01) and high (>0.05) eccentricity values, in order to sample the boundaries sufficiently. It was observed that the sample sets with the highest orthogonality had comparatively few, if any, values of low eccentricity, also meaning that a very limited number of obliquity values were sampled for low eccentricity. We therefore adopted the approach whereby all sample sets that demonstrated normalised orthogonality values that were more than 1 standard deviation above the mean orthogonality were selected. From these, the single sample set with the greatest range of obliquity values for low eccentricity, hence with maximal sampling coverage of the low eccentricity boundary, was selected as the final ensemble design. The input parameter values for the *highCO₂* and *lowCO₂* ensembles are given in Table 2, and the distributions in parameter space illustrated in Fig. 2.

**3.4 AOGCM simulations**

The two CO₂ ensembles were initially run with constant modern-day GIS and WAIS configurations (*modice*). All experiments were initiated from a pre-industrial spin-up experiment, with an atmospheric $CO_2$ concentration of 280 ppmv, and pre-industrial ice sheet extents and orbital conditions. Atmospheric $CO_2$ and the orbital parameters were kept constant throughout each simulation, and each experiment was run for a total of 500 model years. This simulation length allows the experiments with lower $CO_2$ to reach near-equilibrium at the surface. Experiments with higher $CO_2$ have not yet equilibrated by the end of this period; the significance of this is addressed in Sect. 3.6. A number of the very high $CO_2$ experiments caused the model to become unstable and the interpretation of these experiments is discussed in Sect. 3.4.1. A control simulation was also run for 500 years, with the atmospheric $CO_2$ concentration and the orbital parameters set at pre-industrial values. All climate variable results for the model, unless specified, are an average of the final 50 years of the simulation. Anomalies compared with the pre-industrial control (i.e. emulated minus pre-industrial) are discussed and used in the emulator, rather than absolute values, to account for biases in the control climate of the model.

**3.4.1 Very high CO₂ simulations**

As mentioned previously, experiments in the *highCO₂* ensemble with $CO_2$ concentrations of greater than 3100 ppmv become unstable. These experiments exhibit accelerating warming trends several hundred years into the simulation, which eventually cause the model to crash before completion. This is the result of a runaway positive feedback in the GCM caused, at least in part, by the vertical distribution of ozone in the model being prescribed for modern-day climate conditions. Consequently,  the ozone distribution is not  able to respond to changes in climate, meaning that  when increased mean global temperatures result in an increase in altitude of the tropopause and hence an extension of the troposphere,  relatively high concentrations of ozone, which were previously located in the stratosphere, enter the troposphere, resulting in runaway warming.

All other experiments ran for the full 500 years. However, those with a $CO_2$ concentration of 2000 ppmv or higher also exhibited accelerating warming trends before the end of the simulation. Consequently, only simulations with $CO_2$ concentrations of less than 2000 ppmv (equivalent to a total  $CO_2$ release of up to 6000 Pg C) are included in the rest of this study, meaning the methodology is not appropriate for $CO_2$ values greater than this. This equates to 20 experiments in total from the *highCO₂* ensemble, with $CO_2$ concentrations ranging from 303 to 1901 ppmv. All 40 of the *lowCO₂* experiments were used.

**3.5 Sensitivity to ice sheets**

In addition to running the two ensembles with modern-day GIS and WAIS configurations, we also investigated the climatic impact of reducing the sizes of the ice sheets. Many of the $CO_2$ values sampled, particularly in the *highCO₂* ensemble, are significantly higher than pre-industrial levels, and if the resulting climate were to persist for a long periods of time they it could result in significant melting of the continental ice sheets over timescales of $10^3$-$10^4$ years (Charbit et al., 2008; Stone et al., 2010; Winkelmann et al., 2015).

We therefore set up the *highCO$_2$* and *lowCO$_2$* ensembles with reduced GIS and WAIS extents (*lowice*), using the PRISM4 Pliocene reconstruction of the ice sheets (Dowsett et al., 2016). In this reconstruction, the GIS is limited to high elevations in the Eastern Greenland Mountains, and no ice is present over Western Antarctica.

Similar patterns of ice retreat have been simulated in response to future warming scenarios for the GIS (Greve,

2000; Huybrechts and de Wolde, 1999; Ridley et al., 2005; Stone et al., 2010) and WAIS (Huybrechts and de

Wolde, 1999; Winkelmann et al., 2015), equivalent to ~7 m (Ridley et al., 2005) and ~3 m (Bamber et al., 2009;

Feldmann and Levermann, 2015) of global sea level rise, respectively. Large regions of the East Antarctic ice sheet (EAIS) show minimal changes or slightly increased surface elevation, although there is substantial loss of ice in the Wilkes and Aurora subglacial basins (Haywood et al., 2016).

The same CO$_2$ and orbital parameter sample sets were used for both ice configuration ensembles to allow the impact of varying the ice- sheet extents on climate to be directly compared. Only the Greenland and Antarctic grid boxes were modified; the boundary conditions for all other grid boxes, as well as the land/sea mask, were the same as in the modern-day ice sheet simulations. For Greenland and Antarctica, the extent and orography of the ice sheets was updated with the PRISM4 data, as well as the orography of any grid boxes that are projected to be ice-free. Soil properties, land surface type and snow cover were also updated for these grid boxes. Figure 3

compares the orography for the *modice* and *lowice* ensembles, clearly showing the reduced extents for the ice sheets.

**3.5.1 Pattern scaling of reduced ice simulations**

It was expected that reducing the size of the continental ice sheets would have a relatively localised impact on climate (Lunt et al., 2004), and that the effect would be of a linear nature. Therefore, a subset of five simulations from the two ensembles were selected as reduced ice -sheet simulations (*lowCO$_2$* – experiments 8, 19 and 29;

*highCO$_2$* – experiments 21, and 34; see Table 2), covering a range of orbital and CO$_2$ values.

A comparison of the mean annual SAT anomaly for the five experiments showed that the largest temperature changes occur over Greenland and Antarctica, particularly in regions where there is ice in the *modice* ensemble but that are ice free in *lowice*. The spatial pattern of the change is also fairly similar across the simulations, suggesting that the response of climate to the extents of the ice sheets is largely independent of orbital variations or CO$_2$ concentration. The SAT anomaly for the five *lowice* experiments compared with their *modice*

equivalents was calculated, and then averaged across the experiments, shown in Fig. 4a. The largest SAT

anomalies occur locally to the GIS and Antarctic ice sheet (AIS), accompanied by smaller anomalies in some of the surrounding ocean regions (e.g. Barents and Ross Seas), with no significant changes in SAT elsewhere, in line with the results of Lunt et al. (2004); Toniazzo et al. (2004) and (Ridley et al., 2005). This SAT anomaly, caused by the reduced extents of the GIS and WAIS, was then applied (added) to the mean annual SAT anomaly data for all other *highCO$_2$* and *lowCO$_2$ modice* experiments, to generate the SAT data for two *lowice* ensembles.

Also shown in Fig. 4, for comparison, are mean annual SAT anomalies produced by the other forcings, including a doubling of $CO_2$, the difference between maximum and minimum obliquity and the difference between "warm" orbital conditions and "cold" orbital conditions. The warming caused by increased $CO_2$ is more widespread (Fig. 4b), with the largest warming occurring at high latitudes and for land regions, in agreement with typical future-climate simulations (IPCC, 2013, p. 1059). The temperature change due to obliquity and "warm" versus "cold" orbital conditions is less than that for either reduced ice (compared to pre- industrial) or increased $CO_2$. Changes in obliquity have the largest impact on temperatures in high latitude regions, since the exposure of these regions to the sun's radiation is most affected by changes in obliquity.

Smaller temperature anomalies are observed over northern Africa and India and, since an increase in obliquity is indeed known to boost monsoon dynamics (e.g. Araya-Melo et al., 2015; Bosmans et al., 2015), changes in soil latent heat exchanges are therefore expected to contribute negatively to the temperature response. The comparison of "warm" versus "cold" orbital conditions, which highlights (annual mean) temperature changes primarily caused by precession, generally shows a warming trend, with the largest temperature changes occurring in monsoonal regions. Lower temperatures are observed in the Northern Hemisphere over northern

Africa, India and, East Asia, whilst warmer temperatures occur in the Southern Hemisphere over South

America, southern Africa and Australia. Figure 4 demonstrates that the temperature forcing caused by $CO_2$

affects mean annual temperatures on a global scale, whilst the forcing due to ice sheet and orbital changes affects mean annual temperatures in specific regions, having a limited impact on global mean temperatures. This is supported by the relatively high global mean SAT anomaly for the 2x$CO_2$ scenario of 4.2°C, compared with the lower SAT anomalies that result from the obliquity and precession forcing of 0.4°C each (see caption of Fig.

4).

[revised manuscript text omitted]
 same scaling ratio was also applied to the precipitation anomaly data to estimate the equilibrated mean annual precipitation.

The equilibrated global mean annual SAT anomaly ($\Delta T_{eq}$) for the *highCO$_2$* and *lowCO$_2$ modice* ensembles is plotted against log($CO_2$) in Fig. 7, along with $\Delta T_{500}$ for reference. The linear nature of the plot increases our confidence that the Gregory methodology is suitable for our uses, given the logarithmic relationship between SAT and $CO_2$ concentration. Also plotted on Fig. 7 are a number of lines illustrating idealised relationships between $\Delta T_{eq}$ and $CO_2$ based on a range of climate sensitivities. The most recent IPCC report suggested that the likely range for equilibrium climate sensitivity is 1.5°C to 4.5°C (IPCC, 2013), hence sensitivities of 1.5°C, 3°C and 4.5°C have been plotted. The size of the correction required to calculate $\Delta T_{eq}$ from $\Delta T_{500}$ increases with increasing $CO_2$, and brings the final temperature estimates in line with the expected response (red lines), further increasing our confidence. The $\Delta T_{eq}$ estimated for the experiments generally follows the upper line, equivalent to an equilibrium climate sensitivity of 4.5°C, which is higher than a previous estimate of 3.3°C for HadCM3

(Williams et al., 2001). This difference may be due to our simulations being "fully equilibrated" following the
application of the Gregory plot methodology. In addition, Williams et al. (2001) used an older version of
HadCM3 and prescribed vegetation (MOSES1), whilst in this study interactive vegetation is used (MOSES2.1
with TRIFFID).

**4 Calibration and evaluation of the emulator**

By considering different contributions of modern and low ice, high and low $CO_2$, different number of PCs, and
different values for the correlation length hyperparameters, we generated an ensemble of emulators, in order to
test their relative performance. The *modice* and *lowice* ensembles were treated as independent data sets that
were used separately when calibrating the emulator, since ice extent is not defined explicitly as an input
parameter in the emulator code. This approach was adopted, rather than including the ice sheet extent as an
active input parameter to the emulator, because only two ice sheet configurations have been simulated, which
are not sufficient for an interpolation. One of the main benefits of including ice sheet extent as an active input
parameter would be to emulate changing ice sheets over time, but this was beyond the scope of this study. and
this methodology in its current form, as the glacial interglacial cycles are not considered. lLnog($
[revised manuscript text omitted]

approach is only appropriate for periods of the Pliocene with equivalent or less than modern ice sheet extents
(i.e. not glacial conditions), and that palaeogeographic changes for the Pliocene are not included here (
see Sect. 7 for further discussion ). We also tested the *modice* emulator
which, in agreement with the findings in Sect. 4, had a limited impact on the long-term evolution of global SSTs
outside the immediate region of the ice sheets themselves. Potential applications of the emulator for
palaeoclimate are described below.

**5.1 Time series data**

One application of the emulator is to produce a time series of the continuous evolution of climate for a particular
time period, as is illustrated here where climate is simulated at 1 kyr intervals over the period 3300 – 2800 kyr
BP. This period of the late Pliocene was selected because it has been extensively studied as part of a number of
projects (e.g. PRISM (Dowsett et al., 2016; Dowsett, 2007), PlioMIP (Haywood et al., 2010; Haywood et al.,

2016)), represents the warm phase of climate (interglacial conditions), and does not include major glaciations
(though  the M2 cooling event may persist to the very start of the simulation at 3300 kyr BP, and the
simulated period does include periods of likely glaciation, such as KM2 (~3100 kyr BP) and G20 (~3000 kyr
BP)). T the emulator would not be appropriate to periods of extensive glaciation and may not be well-
matched to the periods of lesser glaciation included within the simulated interval. Orbital data for each
1 kyr (Laskar et al., 2004) were provided as input to the calibrated emulator, along with three
representative $CO_2$ concentrations. Three $CO_2$ reference scenarios were initially emulated, with constant
concentrations of 280, 350 and 400 ppmv (although note that in reality, $CO_2$ varied during this period on orbital
timescales (Martinez-Boti et al., 2015)).

To illustrate the comparison of the emulator results to palaeo-proxy data, SST data for various locations were
compared with the emulated SAT for the equivalent grid box. Specifically, alkenone-derived palaeo-SST
estimates from four (Integrated) Ocean Drilling Program (IODP/ODP) sites were used: ODP Site 982 (North
Atlantic; (Lawrence et al., 2009)), IODP Site U1313 (North Atlantic; (Naafs et al., 2010)), ODP Site 722
(Arabian Sea; (Herbert et al., 2010)) and ODP Site 662 (tropical Atlantic; (Herbert et al., 2010)). The locations
of the sites are shown in Fig. 9a and detailed in Table 4. These Pliocene datasets were selected because they are
all of sufficiently high resolution (≤4 kyr) for the impacts of individual orbital cycles on climate to be captured,
whilst covering a range of locations and climatic conditions. Alkenone data are shown converted to SST using
two commonly applied calibrations: Prahl et al. (1988) and Muller et al. (1998). All temperatures are presented
as an anomaly compared with pre-industrial. The emulator results are compared with the SAT for the relevant
grid box in the pre-industrial control experiment, whilst the proxy data are compared with SST observations for
the relevant location taken from the HadISST dataset (Rayner et al., 2003). Observations are annual means and
are averaged over the period 1870-1900.

Table 4 presents the mean SAT anomaly (compared with pre-industrial) for the modelled period as estimated
by the emulator  for the 280 ppmv
scenario
for each of the four grid boxes. The
mean increases with increasing $CO_2$, by ~1°C at low latitudes to 2-3°C at high latitudes for atmospheric $CO_2$ of
400 ppmv. Figure 10 illustrates the evolution of annual mean temperature variations through the late Pliocene as
calculated using the various methods. For the equatorial and Arabian Sea sites (662 and 722), the SAT and SST
estimates are relatively similar to each other in terms of the general estimated temperature, particularly for the
higher $CO_2$ scenarios of 350 and 400 ppmv. However, the comparison of timings and variations between the
SAT and SST data is fairly poor, and there was not found to be a significant correlation between the emulated
and proxy data temperatures at these sites when correlation coefficients were calculated. In fact, Site 982 was
the only location for which significant (negative) correlations were found for a confidence interval of 95%,
although the correlation coefficient is still relatively low. These correlation coefficients were -0.22 (p-value
0.004) for the Prahl et al. (1988) proxy SST data compared with the emulated SAT for the 280 ppmv scenario,
and -0.2 (p-value 0.007) for the same SST data compared with the emulated SAT for the 350 ppmv scenario.

The Muller et al. (1998) SST data demonstrated correlation coefficients that were essentially identical to those above when compared with the same emulated SATs.

[revised manuscript text omitted]

and 722), Naafs et al. (2010) (Site U1313) and Lawrence et al. (2009) (Site 982). Individual records of SST, rather than stacked benthic oxygen isotope data, were used because the GCM experiments that the emulator is calibrated on were only run for 500 years, meaning that deep ocean conditions would not yet have spun-up sufficiently, particularly in the experiments with high $CO_2$. Thus, it would not be appropriate to compare deep ocean temperatures from the experiments with those from the proxy data.

A linear regression is performed on the emulated grid box mean annual SAT data versus prescribed atmospheric

$CO_2$ concentration, for the three constant $CO_2$ scenarios of ranging from 2680, 350 and 400 ppmv up to 800

[revised manuscript text omitted]

Up until ~20 kyr AP, the behaviour of the climate is primarily driven by the high levels of $CO_2$ in the atmosphere caused by as a result of fossil fuel anthropogenic $CO_2$ emissions from a range of sources, including combustion of fossil fuels, and other human activities land-use change and cement production. However, after this time, changes in orbital conditions begin to exert a relatively greater influence on climate, as the periodic fluctuations in SAT at all locations appear to be paced by the orbital cycles, which are shown in Fig. 14a.

The timing and relative amplitudes of the oscillations in future SAT are in good agreement with a number of previous studies. Paillard (2006) applied the conceptual model of Paillard and Parrenin (2004), previously mentioned in Sect. 5, to the next 1 Ma. The development of atmospheric $CO_2$ over the next 200 kyr, simulated by the model following emissions of 450 to 5000 Pg C and accounting for natural variations, shows a similar pattern of response to that of SAT presented here. Estimates of global mean temperature in Archer and

Ganopolski (2005), derived by scaling changes in modelled ice volume to temperature, before applying anthropogenic $CO_2$ temperature forcing for a number of emissions scenarios, also demonstrate fluctuations in global mean annual SAT (not shown) of a similar timing and relative scale. The influence of declining $CO_2$ is still evident after 20 kyr, particularly for the higher emissions scenarios, in the slightly negative gradient of the general evolution of SAT. This is due to the long atmospheric lifetime of fossil fuel $CO_2$ emissions (Archer,

2005), and is also demonstrated in other studies (Archer and Ganopolski, 2005; Archer et al., 2009; Lord et al.,

2016; Paillard, 2006). The impact of excess atmospheric $CO_2$ on the long-term evolution of SAT appears to be fairly linear, with only minor differences between the scenarios and sites, discounting the overall offset of SAT

for different total emissions.

One of the key uncertainties associated with future climate change, which is of particular relevance to radioactive waste repositories located at high northern latitudes, is the timing of the next glacial inception. This is expected to occur during a period of relatively low incoming solar radiation at high northern latitudes, which, for the next 100 kyr, occurs at 0 kyr, 54 kyr and 100 kyr. A number of studies have investigated the possible timing of the next glaciation under pre-industrial atmospheric $CO_2$ concentrations (280 ppmv), finding that it is unlikely to occur until after 50 kyr AP (Archer and Ganopolski, 2005; Berger and Loutre, 2002; Paillard, 2001).

When fossil fuel anthropogenic $CO_2$ emissions are taken into account, the current interglacial is likely to last significantly longer, until ~130 kyr AP following emissions of 1000 Pg C and beyond 500 kyr AP for emissions of 5000 Pg C (Archer and Ganopolski, 2005). A recent study by Ganopolski et al. (2016) using the CLIMBER-2

model found that emissions of 1000 Pg C significantly reduced the probability of a glaciation in the next 100

kyr, and that a glacial inception within the next 100 kyr is very unlikely for $CO_2$ emissions of 1500 Pg C or higher.

Our $CO_2$ emissions scenarios, modelled using the response function of Lord et al. (2016), suggest that atmospheric $CO_2$ will not have returned to pre-industrial levels by 100 ka kyr AP, equalling 298 and 400 ppmv for the 500 and 5000 Pg C emissions scenarios, respectively. We calculated the critical summer insolation threshold at 65° N using the logarithmic relationship identified between maximum summer insolation at 65° N and atmospheric $CO_2$ by Ganopolski et al. (2016). The evolution of atmospheric $CO_2$ concentration over the course of our emissions scenarios suggests that, for emissions of 1000 Pg C or less, Northern Hemisphere summer insolation will next fall below the critical insolation threshold in approximately 50 ka, and in ~100  kyr for emissions of 2000 Pg C. For the highest emissions scenario of 5000 Pg C, the threshold is not passed for considerably longer, until ~160 ka. However, the uncertainty of the critical insolation value is ±4 W m$^{-2}$ (1 standard deviation), and often the difference between summer insolation at 65° N and the insolation threshold is less than this, potentially impacting whether the threshold has in fact been passed and therefore whether glacial inception is likely. For example, for the 1000 Pg C scenario, whilst insolation first falls below the critical threshold at ~50 kyr, it does not fall below by more than the uncertainty value until ~130 ka.

A limitation of our study relates to the continental ice sheets in HadCM3 being prescribed rather than responsive to changes in climate. A consequence of this is that an increase in the extent or thickness of the ice sheets, and hence the onset of glaciation, cannot be explicitly projected, but this also means that a regime shift of the ice sheets to one of negative mass balance, which may be expected to occur under high $CO_2$ emissions scenarios (Ridley et al., 2005; Stone et al., 2010; Swingedouw et al., 2008; Winkelmann et al., 2015), cannot be modelled. However, the results of the sensitivity analysis to ice sheets described in Sect. 3.5., for which a number of simulations were run again with reduced GIS and WAIS extents, suggest that the reduction in continental ice results in relatively localised increases in SAT in regions that are ice free, in addition to some regional cooling at high latitudes. Consequently, this does not act as a significant restriction on the glaciation timings put forward in this study considering their radioactive waste disposal application; given that the earliest timing of the next glaciation is of significant interest, smaller continental ice sheets and therefore higher local SATs would likely inhibit the build-up of snow and ice, delaying glacial inception further. As such, the estimates presented here should be viewed as conservative. As will be discussed in Sect. 7, however, the emulator was not designed and calibrated to predict changes in ice sheets. This is a limitation that should be addressed when modelling future climate on timescales of tens of thousands of years or more (depending on the $CO_2$ scenario(s) being modelled). Another caveat is that the carbon cycle in the emulator is also essentially prescribed, and thus not interactive. This means that the atmospheric $CO_2$ trajectory follows a smooth decline, as was projected using an impulse response function based on experiments using the $c$GENIE model (Lord et al., 2016), with long-term future climate being modelled as a series of snapshot simulations with the emulator. This smooth decline in $CO_2$ assumes that no non-linear or unexpected behaviour will be demonstrated by the long-term carbon cycle, and that the silicate weathering mechanism, which is associated with a substantial degree of uncertainty, is correct.

The emulator can also be used to project the evolution of a range of other climate variables, providing that they were modelled as part of the initial GCM ensembles. Figure 15 illustrates the development of mean annual precipitation and emulated uncertainty over the next 200 kyr at the four sites. The maximum increase in precipitation is between $0.3 \pm 0.1$ mm day$^{-1}$ (Switzerland grid box) and $0.6 \pm 0.1$ mm day$^{-1}$ (Sweden grid box) in the 500 Pg C and 5000 Pg C scenarios, respectively. Precipitation increases with increasing atmospheric $CO_2$ at all sites apart from the Spain grid box, where it decreases by up to $0.9 \pm 0.1$ mm day$^{-1}$. Regional differences in the sign of changes in precipitation, including an increase at high latitudes and a decrease in the Mediterranean, are consistent with modelling results included in the International Panel on Climate Change (IPCC) Fifth Assessment Report, for simulations forced with the Representative Concentration Pathway (RCP) 8.5 scenario (Collins et al., 2013). In contrast to SAT, precipitation appears to be more closely influenced by precession, illustrated by its periodicity of slightly less than 25 kyr. There appears to be; an increase in the intensity of precipitation fluctuations from approximately 140 kyr onwards, suggest implying that the modulation of precession by eccentricity also has an impact, as expected.

**6.2 Orbital variability and spectral analysis**

The impact of orbital forcing was assessed by performing a spectral wavelet analysis on the SAT and precipitation time series data produced by the emulator for the Central England grid box for the 5000 Pg C emissions scenario, represented by blue lines in Fig. 14c and 15c, respectively. As for the late Pliocene, the wavelet software of Torrence and Compo (1998) was utilized. The analysis was performed on the data for 20-200 kyr AP, because the climate response up until ~20 kyr AP is dominated by the impact of elevated atmospheric $CO_2$ concentrations, which masks the orbital signal and affects the results of the wavelet analysis.

For future SAT, Fig. 16a suggests that, up until ~160 kyr, the obliquity cycle acts as the dominant influence, resulting in temperature oscillations with a periodicity of approximately 41 kyr. This is confirmed by Fig. 14c, which shows that the major peaks in SAT generally coincide with periods of high obliquity. Over this period, precession has a far more limited influence, likely due to eccentricity being relatively low until ~110 kyr (Fig. 14a). However, from ~120 kyr AP onwards, concurrently with increasing eccentricity, precession becomes a more significant forcing on climate, resulting in SAT peaks approximately every 21 kyr. In contrast, precession appears to be the dominant forcing on precipitation for the Central England grid box for the entire 20-200 kyr AP period (Fig. 15c and 16b). This signal is particularly strong after ~120 kyr AP, due to higher eccentricity.

**7 Limitations**

There are a number of limitations associated with the methodology outlined above, emulator, particularly relating to the assumptions that it is based on and its application to different periods of time. Although these have mostly been discussed briefly in the preceding sections, here we summarise them together.

- AFirstly, as noted previously, the carbon cycle in the emulator is not coupled to the climate, essentially fixed, since the atmospheric $CO_2$ concentration is prescribed. It The methodology thus assumes that there will be no unexpected non-linearities in the carbon cycle, and that changes in climate that are different from those in cGENIE do not feed back to the carbon cycle. This may be of particular importance when simulating future climates, when the natural carbon cycle is expected to be significantly perturbed due to ongoing anthropogenic emissions of $CO_2$, in a way that may not be fully represented in cGENIE. There is also uncertainty surrounding the dynamics of the carbon cycle over long periods of time, such as the role of the silicate weathering mechanism, although the observation that different carbon cycle models generally produce fairly similar results increases our confidence (Archer et al., 2009).

- TSecondly, the ice sheets in the emulator are also fixed, at either modern-day or reduced extents, although expanding the range of ice sheets that can be modelled is currently being undertaken in ongoing research. This means that care needs to be taken when simulating very long periods of time. For example, neither Quaternary nor futurethe glacial-interglacial cycles cannot be accounted forcapturedsimulated using the current version of the emulator, which are known to have occurred in the past (e.g. Petit et al., 1999) and are expected to continue in the future. Furthermore, even dDuring the Pliocene, it is likely that the extent of ice sheets in the Northern Hemisphere varied beyond the range simulated in this study (Willeit et al., 2015), and the emulator in its current form cannot represent this.

- In the context of the PlicoenePliocene, tThe land-sea mask and orography used in the simulation of Pliocene climate are also fixed and appropriate to modern-day conditions, whereas the PRISM4 reconstruction of paleogeographyPliocene ice sheets suggests that therey may have been considerably different in some regions, for example the region of the Hudson Bay is thought to have been land in the Pliocene (Dowsett et al., 2016).

- Due to both the carbon cycle and ice sheets being prescribed, interactions between these components of the climate system can also not be simulated. These include natural changes in $CO_2$ which have been found to accompany past glacial-interglacial cycles, with glacial periods over the last 800 kyr exhibiting $CO_2$ concentrations of approximately 180 to 200 ppmv (Petit et al., 1999), whereas interglacial periods demonstrated concentrations of 240 to 290 ppmv (Luthi et al., 2008). Changes in the ice sheets in response to atmospheric $CO_2$ can also not be modelled, such as the likely future melting of the GIS and AIS in response to anthropogenic $CO_2$ emissions. Various studies have modelled the response of the ice sheets to future climate warming, finding that the ice sheets may experience significantly increased melt. In fact, for scenarios with high $CO_2$ emissions (>~5000 Pg C), it has been suggested that the GIS and AIS may be almost entirely melted within the next few thousand years (e.g. DeConto and Pollard, 2016; Huybrechts et al., 2011; Winkelmann et al., 2015), which would cause significant changes in deep ocean circulation and ocean stratification. These ocean changes cannot be captured by the current version of the methodologyemulator and, whilst their impacts on global and regional climate are uncertain, they are expected to be long-term. The melting of the ice sheets would also cause significant increases in global sea level, of approximately 70 m if both the GIS and AIS melted, which would strongly affect the global land-sea mask and regional climates, and which cannot be represented using the current methodology. This sea level rise would also have serious implications for radioactive waste repositories located in relatively low-lying coastal regions that are vulnerable to sea level rise, such as in northern Europe.
- Since the emulator models climate via a series of snapshots rather than a truly transient simulation, it is not able to capture deviations frorm a stationary responsetrends in deep ocean conditions. As a consequence, the methodology becomes inappropriate if such transient changes in the deep ocean are found to be important for controlling surface climate evolution.

- The emulator presented in this study is only suitable for modelling transient climate changes on timescales of several millennia or longer, as a number of shorter-term processes in the climate and carbon cycle are not represented. These include internal variability in the climate system, such as interannual variability, North Atlantic Oscillation (NAO), and El Niño – Southern Oscillation (ENSO), as well as radiative forcing occurring on shorter timescales (e.g. volcanic activity), and terrestrial carbon cycle processes. On these timescales, transient simulations run using  complex models such as GCMs or EMICS are most appropriate.

As a consequence of these limitations, care needs to be taken when applying the emulator to ensure that its application is appropriate. For example,  when considering future climate, the way in which future carbon dioxide concentration have been modelled, and the ice sheet configurations modelled, mean that this methodology is only applicable on timescales up until the next glacial inception. After this, atmospheric $CO_2$ would be expected to change in response to the initiation of glacial conditions, accompanied by the expansion of the ice sheets, decreasing sea level, and the climatic changes that would results from these changes. A number of studies have modelled the possible timing of the next glacial inception, finding that for $CO_2$ scenarios with medium emissions the current interglacial period may end in approximately 130 kyr (Archer and Ganopolski, 2005; Ganopolski et al., 2016). However, for high emissions of 5000 Pg C, glacial inception may be delayed for more than 500 kyr (Archer and Ganopolski, 2005). A study by Brandefelt et al. (2013) estimated that for permafrost development to occur at Forsmark, Sweden during the insolation minima at 17 and 54 kyr AP, atmospheric $CO_2$ concentrations of ~210 ppmv or less and ~250 ppmv or less would be required, respectively. In light of the long atmospheric lifetime of $CO_2$ emissions that has been discussed, low concentrations such as these are unlikely in the next few tens of thousands of years; however, they cannot be entirely excluded. In order to account for this limitation, the emulator could be extended to include glacial states, meaning that it could be applied to future climate on a longer timescale, as well as to the Quaternary, if the evolution of $CO_2$ and ice volume were known (e.g. from a transient EMIC of conceptual model simulation). Thus, when emulating long-term climate, careful consideration should be given to what assumptions are being made and whether the methodology is appropriate for the conditions being modelled.

Bearing in mind these limitations,  the methodology described in this paper is to be a useful and powerful tool for simulating long-term past and future climatic changes, as well as for exploring the dynamics and sensitivities of the climate system.

**Summary and Conclusions**

In this study, we present long-term continuous projections of future climate evolution at the spatial resolution of a GCM, via the use of a statistical emulator. The emulator was calibrated on two ensembles of simulations with varied orbital and atmospheric $CO_2$ conditions and modern day continental ice sheet extents, produced using the HadCM3 climate model. The method presented by Gregory et al. (2004) to calculate the steady-state global temperature change for a simulation, by regressing the net radiative flux at the top of the atmosphere against the change in global SAT, was utilised to calculate the equilibrated SAT data for these ensembles, as it was not feasible to run the experiments to equilibrium due to the associated time and computer resources needed. A number of simulations testing the sensitivity of SAT to the extent of the GIS and WAIS suggest that the response of SAT is fairly linear regardless of orbit, and that the largest changes are generally local to regions that are ice free. The mean SAT anomaly identified across these experiments was then applied to the equilibrated SAT results of the modern-day ice sheet extent ensembles, to generate two equivalent ensembles with reduced ice sheets.

Output data from the modern-day and reduced ice sheet ensembles were then used to calibrate separate emulators, which were optimised and then validated using a leave-one-out approach, resulting in satisfactory performance results. We discuss a number of useful applications of the emulator, which may not be possible using other modelling approaches at the same temporal and spatial resolution. Firstly, a particular benefit of the emulator is that it can be used to produce time series of climatic variables that cover long periods of time (i.e. several thousand years or more) at a GCM resolution, accompanied by an estimation of the uncertainty in the form of the posterior variance. This would not be feasible using GCMs due to the significant time and computational requirements involved. The global grid coverage of the data also means that the evolution of a climate variable at a particular grid box can be examined, allowing for comparisons to data at a regional or local scale, such as palaeo-proxy data, or for the evolution of climate at a specific site to be studied. However, further downscaling of the data may also be necessary or beneficial, via further modelling such as proxy modelling, impact models or regional climate models, or via statistical downscaling techniques. 
[revised manuscript text omitted]

Bosmans, J. H. C., Drijfhout, S. S., Tuenter, E., Hilgen, F. J., and Lourens, L. J.: Response of the North
African summer monsoon to precession and obliquity forcings in the EC-Earth GCM, Clim Dynam, 44,
279-297, doi: 10.1007/s00382-014-2260-z, 2015.

[revised manuscript text omitted]

Pepin, L., Ritz, C., Saltzman, E., and Stievenard, M.: Climate and atmospheric history of the past
420,000 years from the Vostok ice core, Antarctica, Nature, 399, 429-436, doi: 10.1038/20859, 1999.
Pope, V. D., Gallani, M. L., Rowntree, P. R., and Stratton, R. A.: The impact of new physical
parametrizations in the Hadley Centre climate model: HadAM3, Clim Dynam, 16, 123-146, doi:
10.1007/s003820050009, 2000.
Prahl, F. G., Muehlhausen, L. A., and Zahnle, D. L.: Further evaluation of long-chain alkenones as
indicators of paleoceanographic conditions, Geochim Cosmochim Ac, 52, 2303-2310, doi:
10.1016/0016-7037(88)90132-9, 1988.
Prell, W. L. and Kutzbach, J. E.: Monsoon Variability over the Past 150,000 Years, J Geophys Res-
Atmos, 92, 8411-8425, doi: 10.1029/JD092iD07p08411, 1987.
Prescott, C. L., Haywood, A. M., Dolan, A. M., Hunter, S. J., Pope, J. O., and Pickering, S. J.: Assessing
orbitally-forced interglacial climate variability during the mid-Pliocene Warm Period, Earth Planet Sc
Lett, 400, 261-271, doi: 10.1016/j.epsl.2014.05.030, 2014.
Raymo, M. E., Grant, B., Horowitz, M., and Rau, G. H.: Mid-Pliocene warmth: Stronger greenhouse
and stronger conveyor, Mar Micropaleontol, 27, 313-326, doi: 10.1016/0377-8398(95)00048-8,
1996.
Rayner, N. A., Parker, D. E., Horton, E. B., Folland, C. K., Alexander, L. V., Rowell, D. P., Kent, E. C., and
Kaplan, A.: Global analyses of sea surface temperature, sea ice, and night marine air temperature
since the late nineteenth century, J Geophys Res-Atmos, 108, 4407, doi: 10.1029/2002jd002670,
2003.
Ridley, J. K., Huybrechts, P., Gregory, J. M., and Lowe, J. A.: Elimination of the Greenland ice sheet in
a high $CO_2$ climate, J Climate, 18, 3409-3427, doi: 10.1175/Jcli3482.1, 2005.
Rogner, H. H.: An assessment of world hydrocarbon resources, Annu Rev Energ Env, 22, 217-262,
doi: 10.1146/annurev.energy.22.1.217, 1997.
Sacks, J., Welch, W. J., Mitchell, T. J., and Wynn, H. P.: Design and analysis of computer experiments,
Statistical Science, 4, 409-423, doi: 10.1214/ss/1177012413, 1989.
Seki, O., Foster, G. L., Schmidt, D. N., Mackensen, A., Kawamura, K., and Pancost, R. D.: Alkenone and
boron-based Pliocene $pCO_2$ records, Earth Planet Sc Lett, 292, 201-211, doi:
10.1016/j.epsl.2010.01.037, 2010.
SKB: Long-term safety for the final repository for spent nuclear fuel at Forsmark. Main report of the
SR-Site project, Svensk Kärnbränslehantering AB, Stockholm, Sweden, SKB Report TR-11-01.
Available from: www.skb.com/publication/2345580, 2011.
Stap, L. B., van de Wal, R. S. W., de Boer, B., Bintanja, R., and Lourens, L. J.: Interaction of ice sheets
and climate during the past 800 000 years, Clim Past, 10, 2135-2152, doi: 10.5194/cp-10-2135-2014,
2014.
Stone, E. J., Lunt, D. J., Rutt, I. C., and Hanna, E.: Investigating the sensitivity of numerical model
simulations of the modern state of the Greenland ice-sheet and its future response to climate
change, Cryosphere, 4, 397-417, doi: 10.5194/tc-4-397-2010, 2010.
Swingedouw, D., Fichefet, T., Huybrechts, P., Goosse, H., Driesschaert, E., and Loutre, M. F.: Antarctic
ice-sheet melting provides negative feedbacks on future climate warming, Geophys Res Lett, 35, doi:
10.1029/2008gl034410, 2008.
Texier, D., Degnan, P., Loutre, M. F., Paillard, D., and Thorne, M. C.: Modelling sequential BIOsphere
systems under CLIMate change for radioactive waste disposal. Project BIOCLIM, Las Vegas, Nevada,
30 March – 2 April 2003, 2003.
Toniazzo, T., Gregory, J. M., and Huybrechts, P.: Climatic impact of a Greenland deglaciation and its
possible irreversibility, J Climate, 17, 21-33, doi: 10.1175/1520-
0442(2004)017<0021:Cioagd>2.0.Co;2, 2004.
Torrence, C. and Compo, G. P.: A practical guide to wavelet analysis, B Am Meteorol Soc, 79, 61-78,
doi: 10.1175/1520-0477(1998)079<0061:Apgtwa>2.0.Co;2, 1998.

Tuenter, E., Weber, S. L., Hilgen, F. J., and Lourens, L. J.: The response of the African summer monsoon to remote and local forcing due to precession and obliquity, Global Planet Change, 36, 219-235, doi: 10.1016/S0921-8181(02)00196-0, 2003.

Valdes, P. J., Armstrong, E., Badger, M. P. S., Bradshaw, C. D., Bragg, F., Davies-Barnard, T., Day, J. J., Farnsworth, A., Hopcroft, P. O., Kennedy, A. T., Lord, N. S., Lunt, D. J., Marzocchi, A., Parry, L. M., Roberts, W. H. G., Stone, E. J., Tourte, G. J. L., and Williams, J. H. T.: The BRIDGE HadCM3 family of climate models: HadCM3@Bristol v1.0, Geosci Model Dev, doi: 10.5194/gmd-2017-16, 2017. doi: 10.5194/gmd-2017-16, 2017.

Wilkinson, R. D. (Ed.): Bayesian calibration of expensive multivariate computer experiments, John Wiley & Sons, Ltd, 2010.

Willeit, M., Ganopolski, A., Calov, R., Robinson, A., and Maslin, M.: The role of $CO_2$ decline for the onset of Northern Hemisphere glaciation, Quaternary Sci Rev, 119, 22-34, doi: 10.1016/j.quascirev.2015.04.015, 2015.

Williams, K. D., Senior, C. A., and Mitchell, J. F. B.: Transient climate change in the Hadley Centre models: The role of physical processes, J Climate, 14, 2659-2674, doi: 10.1175/1520-0442(2001)014<2659:Tccith>2.0.Co;2, 2001.

Williams, K. D., Ingram, W. J., and Gregory, J. M.: Time variation of effective climate sensitivity in GCMs, J Climate, 21, 5076-5090, doi: 10.1175/2008jcli2371.1, 2008.

Winkelmann, R., Levermann, A., Ridgwell, A., and Caldeira, K.: Combustion of available fossil-fuel resources sufficient to eliminate the Antarctic ice sheet, Science Advances, 1, e1500589, doi: 10.1126/sciadv.1500589, 2015.

Winton, M., Takahashi, K., and Held, I. M.: Importance of ocean heat uptake efficacy to transient climate change, J Climate, 23, 2333-2344, doi: 10.1175/2009jcli3139.1, 2010.

Yokoyama, Y., Lambeck, K., De Deckker, P., Johnston, P., and Fifield, L. K.: Timing of the Last Glacial Maximum from observed sea-level minima, Nature, 406, 713-716, doi: 10.1038/35021035, 2000.

**Table 1. Ensembles setup: sampling ranges for input parameters (obliquity, $e\sin\varpi$, $e\cos\varpi$ and $CO_2$) for the *highCO₂***
**and *lowCO₂* ensembles.**

| Ensemble | Time covered from present day (AP) | Parameter | Sampling range | |
|---|---|---|---|---|
| | | | Minimum | Maximum |
| *highCO₂* | 110 kyr | $\varepsilon$ (°) | 22.3 | 24.3 |
| | | $e\sin\varpi$ | -0.016 | 0.016 |
| | | $e\cos\varpi$ | -0.016 | 0.015 |
| | | $CO_2$ (ppmv) | 280 | 3600 |
| *lowCO₂* | 1 Ma | $\varepsilon$ (°) | 22.2 | 24.4 |
| | | $e\sin\varpi$ | -0.055 | 0.055 |
| | | $e\cos\varpi$ | -0.055 | 0.055 |
| | | $CO_2$ (ppmv) | 250 | 560 |

**Table 2. Experiment setup: Orbital parameters (obliquity, eccentricity and longitude of perihelion) and atmospheric**
**$CO_2$ concentration for simulations in the *highCO₂* and *lowCO₂* ensembles. All experiments in both ensembles were**
**run with modern ice sheet (*modice*) configurations. Experiments shown in bold were also run with reduced ice sheet**
**(*lowice*) configurations. The experiment number is given, and the experiment name is constructed using the ice sheet**
**configuration, the ensemble name and the experiment number, for example: modice_lowCO2_1.**

| Ensemble | # | $\varepsilon$ (°) | $e$ - | $\varpi$ (°) | $CO_2$ (ppmv) | Ensemble | # | $\varepsilon$ (°) | $e$ - | $\varpi$ (°) | $CO_2$ (ppmv) |
|---|---|---|---|---|---|---|---|---|---|---|---|
| *highCO₂* | 1 | 23.53 | 0.0093 | 240.3 | 3348.2 | *lowCO₂* | 1 | 22.99 | 0.0481 | 320.1 | 375.7 |
| | 2 | 24.24 | 0.0135 | 212.6 | 2159.3 | | 2 | 23.02 | 0.0323 | 63.7 | 516.9 |
| | 3 | 22.38 | 0.0110 | 260.0 | 1645.0 | | 3 | 22.81 | 0.0481 | 334.2 | 470.4 |
| | 4 | 24.07 | 0.0044 | 101.8 | 800.8 | | 4 | 24.03 | 0.0537 | 84.9 | 390.3 |
| | 5 | 23.07 | 0.0203 | 313.0 | 1999.9 | | 5 | 23.09 | 0.0294 | 293.8 | 325.3 |
| | 6 | 24.03 | 0.0087 | 184.9 | 3049.0 | | 6 | 23.58 | 0.0098 | 325.1 | 337.5 |
| | 7 | 22.53 | 0.0163 | 162.0 | 900.9 | | 7 | 23.72 | 0.0133 | 74.3 | 489.2 |
| | 8 | 23.57 | 0.0158 | 21.0 | 1746.3 | | **8** | **24.17** | **0.0066** | **174.1** | **346.0** |
| | 9 | 23.34 | 0.0131 | 113.5 | 996.8 | | 9 | 23.82 | 0.0400 | 48.2 | 260.6 |
| | 10 | 23.37 | 0.0198 | 220.2 | 3139.3 | | 10 | 23.39 | 0.0412 | 53.8 | 409.5 |
| | 11 | 22.73 | 0.0187 | 236.1 | 1081.9 | | 11 | 22.89 | 0.0531 | 115.2 | 436.6 |
| | 12 | 22.63 | 0.0121 | 184.8 | 2451.5 | | 12 | 23.34 | 0.0281 | 133.9 | 504.4 |
| | 13 | 22.41 | 0.0131 | 192.8 | 3372.4 | | 13 | 22.65 | 0.0473 | 102.6 | 555.6 |
| | 14 | 22.78 | 0.0137 | 299.3 | 448.2 | | 14 | 23.20 | 0.0368 | 180.9 | 385.1 |
| | 15 | 22.97 | 0.0111 | 14.1 | 1225.7 | | 15 | 23.96 | 0.0232 | 40.0 | 403.4 |
| | 16 | 22.90 | 0.0087 | 62.2 | 1841.9 | | 16 | 24.27 | 0.0460 | 298.1 | 341.1 |
| | 17 | 23.63 | 0.0151 | 200.6 | 1151.6 | | 17 | 22.35 | 0.0391 | 265.9 | 522.1 |
| | 18 | 23.77 | 0.0134 | 78.7 | 2101.7 | | 18 | 23.91 | 0.0361 | 343.2 | 318.6 |
| | 19 | 23.73 | 0.0159 | 323.7 | 1526.6 | | **19** | **22.33** | **0.0484** | **324.2** | **264.5** |
| | 20 | 24.29 | 0.0082 | 164.6 | 2890.4 | | 20 | 22.94 | 0.0350 | 268.7 | 540.8 |

| | | | | | | | | | |
|---|---|---|---|---|---|---|---|---|---|
| **21** | **22.31** | **0.0038** | **299.1** | **1389.5** | 21 | 22.68 | 0.0323 | 332.4 | 531.5 |
| 22 | 23.42 | 0.0117 | 122.5 | 397.3 | 22 | 24.28 | 0.0387 | 118.7 | 446.7 |
| 23 | 24.00 | 0.0101 | 206.6 | 303.4 | 23 | 23.60 | 0.0484 | 282.0 | 310.5 |
| 24 | 22.48 | 0.0146 | 294.9 | 2845.7 | 24 | 24.19 | 0.0337 | 346.3 | 548.3 |
| 25 | 22.57 | 0.0067 | 81.2 | 1341.2 | 25 | 24.14 | 0.0423 | 11.6 | 425.4 |
| 26 | 22.93 | 0.0171 | 114.4 | 3516.0 | 26 | 22.20 | 0.0035 | 85.2 | 303.0 |
| 27 | 24.13 | 0.0143 | 257.3 | 2951.8 | 27 | 22.78 | 0.0070 | 212.1 | 480.4 |
| 28 | 23.00 | 0.0062 | 272.2 | 2274.6 | 28 | 22.72 | 0.0526 | 239.9 | 280.0 |
| 29 | 23.95 | 0.0103 | 114.7 | 564.7 | **29** | **23.65** | **0.0543** | **30.3** | **362.0** |
| 30 | 23.17 | 0.0169 | 56.7 | 1900.9 | 30 | 23.24 | 0.0351 | 200.4 | 411.9 |
| 31 | 23.70 | 0.0122 | 1.4 | 773.0 | 31 | 23.87 | 0.0276 | 156.5 | 287.5 |
| 32 | 23.24 | 0.0021 | 310.2 | 2582.1 | 32 | 22.25 | 0.0499 | 208.9 | 365.3 |
| 33 | 22.81 | 0.0121 | 66.3 | 2386.5 | 33 | 22.54 | 0.0510 | 103.4 | 471.1 |
| **34** | **24.18** | **0.0145** | **36.6** | **668.2** | 34 | 22.58 | 0.0404 | 292.2 | 544.5 |
| 35 | 23.82 | 0.0075 | 10.8 | 2244.8 | 35 | 22.87 | 0.0530 | 20.9 | 498.2 |
| 36 | 23.14 | 0.0141 | 314.1 | 3588.9 | 36 | 23.53 | 0.0414 | 147.0 | 507.0 |
| 37 | 23.49 | 0.0121 | 101.5 | 2760.4 | 37 | 22.39 | 0.0165 | 149.1 | 393.9 |
| 38 | 22.66 | 0.0162 | 69.5 | 2623.9 | 38 | 22.43 | 0.0537 | 175.0 | 484.8 |
| 39 | 23.28 | 0.0146 | 207.5 | 1484.8 | 39 | 24.38 | 0.0482 | 342.9 | 418.3 |
| 40 | 23.89 | 0.0092 | 21.1 | 3188.8 | 40 | 23.76 | 0.0504 | 127.0 | 528.1 |

**Table 3. Parameter values estimated from Gregory plots for the 2x and 4x pre-industrial $CO_2$ simulations. Shown are**
**the effective radiative forcing ($F$; W m$^{-2}$) and the climate feedback parameter ($\alpha$; W m$^{-2}$ °C$^{-1}$) for years 1-20 and years**
**21-100. The uncertainties are the standard error from the linear regression.**

| Simulation | | $F$ | | $\alpha$ | |
|---|---|---|---|---|---|
| | | (W m$^{-2}$) | | (W m$^{-2}$ °C$^{-1}$) | |
| | | yr 1-20 | yr 21-100 | yr 1-20 | yr 21-100 |
| $2xCO_2$ | *modice_lowCO2_13* | $4.24 \pm 0.4$ | - | $-1.30 \pm 0.2$ | $-0.68 \pm 0.05$ |
| $4xCO_2$ | *modice_highCO2_17* | $6.88 \pm 0.3$ | - | $-0.99 \pm 0.1$ | $-0.56 \pm 0.02$ |

**Table 4. Mean temperature anomalies and uncertainties (1 standard deviation) for the period 3300-2800 kyr BP estimated by the emulator and alkenone proxy data for the four ODP/IODP sites.**

| ODP/IODP Site | Location | | | Emulated SAT anomaly (ºC) | | | Proxy data SST anomaly (ºC) | |
|---|---|---|---|---|---|---|---|---|
| | | Lat | Lon | 280 ppmv | 350 ppmv | 400 ppmv | Prahl et al. (1988) | Muller et al. (1998) |
| 982[1] | North Atlantic | 57.5º N | 15.9º W | 0.6 ±0.4 | 2.4 ±0.3 | 3.3 ±0.3 | 5.4 | 5.7 |
| U1313[2] | North Atlantic | 41.0º N | 33.0º W | -0.8 ±0.3 | 0.0 ±0.2 | 0.8 ±0.2 | 1.6 | 2.0 |
| 722[3] | Arabian Sea | 16.6º N | 59.8º E | 0.0 ±0.2 | 1.0 ±0.2 | 1.7 ±0.2 | 1.0 | 1.7 |
| 662[3] | Tropical Atlantic | 1.4º S | 11.7º W | 0.2 ±0.2 | 0.9 ±0.2 | 1.3 ±0.2 | 1.3 | 1.9 |

[1]Lawrence et al. (2009); [2]Naafs et al. (2010); [3]Herbert et al. (2010).

[Figure]

**Figure 1. Time series of atmospheric $CO_2$ concentration (ppmv) for the next 200 kyr following logistic $CO_2$ emissions of 10,000 PgC, run using the cGENIE model (Lord et al., 2016). Also shown are the upper and lower $CO_2$ limits of the *highCO2* (red dashed lines) and *lowCO2* (green dashed lines) ensembles. The pre-industrial $CO_2$ concentration of 280 ppmv (horizontal grey dotted line), and the 110 kyr cut-off for the highCO2 ensemble (vertical grey dotted line) are included for reference.**

[Figure]

**Figure 2. Distribution of 40 experiments produced by Latin hypercube sampling, displayed as two-dimensional**
**projections through four-dimensional space. (a)** *highCO₂* **ensemble, (b)** *lowCO₂* **ensemble. The variables are**
**eccentricity (*e*), longitude of perihelion (*ϖ*; degrees), obliquity (*ε*; degrees), and atmospheric CO₂ concentration**
**(ppmv). A pre-industrial control simulation is shown in red. In the *highCO₂* ensemble, experiments with CO₂**
**concentrations of more than 2000 ppmv, shown in grey, were excluded from the emulator.**

[Figure]

**Figure 3. Orography (m) in the two ice sheet configuration ensembles. (a)** *modice* **ensemble, (b)** *lowice* **ensemble.**
**Differences only occur over Greenland and Antarctica.**

[Figure]

**Figure 4. Mean annual SAT (ºC) anomalies produced by the various climate forcings. (a)The *lowice* experiments compared with their *modice* equivalents, averaged across the five *lowice* experiments. (b)  :(d) Idealized experiments performed using the *modice* emulator. All orbital and $CO_2$ conditions are set to pre-industrial values unless specified: (b) 2x pre-industrial $CO_2$, (c) maximum obliquity compared to minimum obliquity, (d) "warm" orbital conditions (high eccentricity, NH summer at perihelion) compared to "cold" orbital conditions (low eccentricity, NH summer at aphelion).  The different forcings result in global mean SAT anomalies of: (b) 4.2°C, (c) 0.4°C, and (d) 0.4°C.**

[revised manuscript text omitted]